# Coral reefs benefit from reduced land–sea impacts under ocean warming

Jamison M. Gove[1,17 ✉], Gareth J. Williams[2,17 ✉], Joey Lecky[3], Eric Brown[4], Eric Conklin[5], Chelsie Counsell[6], Gerald Davis[3], Mary K. Donovan[7,8], Kim Falinski[5], Lindsey Kramer[9], Kelly Kozar[10], Ning Li[11], Jeffrey A. Maynard[12], Amanda McCutcheon[10], Sheila A. McKenna[10], Brian J. Neilson[13], Aryan Safaie[14], Christopher Teague[13], Robert Whittier[15] & Gregory P. Asner[7,16]

Coral reef ecosystems are being fundamentally restructured by local human impacts and climate-driven marine heatwaves that trigger mass coral bleaching and mortality[1]. Reducing local impacts can increase reef resistance to and recovery from bleaching[2]. However, resource managers lack clear advice on targeted actions that best support coral reefs under climate change[3] and sector-based governance means most land- and sea-based management efforts remain siloed[4]. Here we combine surveys of reef change with a unique 20-year time series of land–sea human impacts that encompassed an unprecedented marine heatwave in Hawai'i. Reefs with increased herbivorous fish populations and reduced land-based impacts, such as wastewater pollution and urban runoff, had positive coral cover trajectories predisturbance. These reefs also experienced a modest reduction in coral mortality following severe heat stress compared to reefs with reduced fish populations and enhanced land-based impacts. Scenario modelling indicated that simultaneously reducing land–sea human impacts results in a three- to sixfold greater probability of a reef having high reef-builder cover four years postdisturbance than if either occurred in isolation. International efforts to protect 30% of Earth's land and ocean ecosystems by 2030 are underway[5]. Our results reveal that integrated land–sea management could help achieve coastal ocean conservation goals and provide coral reefs with the best opportunity to persist in our changing climate.

Coastal areas contain some of the most biologically diverse and productive marine ecosystems on Earth[6]. But with four times the population density living within 20 km of the ocean compared to the rest of the world[7], direct human impacts on local scales are fundamentally restructuring these important marine communities[8]. Coastal areas are also affected by stronger and more frequent disturbances fuelled by human-induced climate change[9]. These human stressors are especially acute on tropical coral reefs where up to 90% of the local population live along the shoreline[10]. Land-based stressors, such as wastewater pollution, combine with sea-based stressors, such as overfishing, to disrupt natural ecological feedbacks on reefs[11]. Corals are further stressed by prolonged periods of anomalously warm ocean temperatures, known as marine heatwaves[12], that can cause mass coral bleaching[13] and mortality and fundamentally transform reef assemblages[14,15].

Reducing human impacts on local scales to maintain ecosystem integrity has been the guiding model of coral reef conservation for decades[3]. Its importance was established in the indigenous stewardship of island ecosystems, which used a decentralized and integrated resource management strategy that extended from the mountains to the sea[16,17]. By contrast, contemporary centralized governance means most terrestrial and ocean management efforts remain siloed[4,17,18]. As a result, whereas local resource managers have aspired to an integrated land–sea approach[19], evidence of its efficacy above either approach in isolation remains wanting and difficult to test. Detecting conservation benefits in highly dynamic ecosystems is challenging[20], but recent studies have identified salient connections between local conditions and coral reef resistance to and recovery potential following mass bleaching[2,11,21–23]. Managers therefore require unambiguous targets for the combination of land–sea human impacts they should mitigate to support coral reef persistence under climate change. Hampering these efforts are a lack of spatially resolved data on local drivers of coral reef ecosystems over time. Researchers are often forced to use proxies

[1]Pacific Islands Fisheries Science Center, National Oceanic and Atmospheric Administration (NOAA), Honolulu, HI, USA. [2]School of Ocean Sciences, Bangor University, Menai Bridge, Anglesey, UK. [3]Pacific Islands Regional Office, National Oceanic and Atmospheric Administration, Honolulu, HI, USA. [4]National Park of American Samoa, Pago Pago, American Samoa, USA. [5]The Nature Conservancy, Honolulu, HI, USA. [6]Cooperative Institute for Marine and Atmospheric Research, Honolulu, HI, USA. [7]Center for Global Discovery and Conservation Science, Arizona State University, Hilo, HI, USA. [8]School of Geographical Sciences and Urban Planning, Arizona State University, Tempe, AZ, USA. [9]Hawai'i Wildlife Fund, Kealakekua, HI, USA. [10]National Park Service, Pacific Island Network Inventory and Monitoring, Hawai'i National Park, HI, USA. [11]Department of Ocean and Resources Engineering, University of Hawai'i at Mānoa, Honolulu, HI, USA. [12]SymbioSeas, Carolina Beach, NC, USA. [13]Hawai'i Division of Aquatic Resources, Honolulu, HI, USA. [14]Graduate School of Oceanography, University of Rhode Island, Narragansett, RI, USA. [15]Hawai'i Department of Health, Honolulu, HI, USA. [16]School of Ocean Futures, Arizona State University, Hilo, HI, USA. [17]These authors contributed equally: Jamison M. Gove, Gareth J. Williams. ✉e-mail: jamison.gove@noaa.gov; g.j.williams@bangor.ac.uk

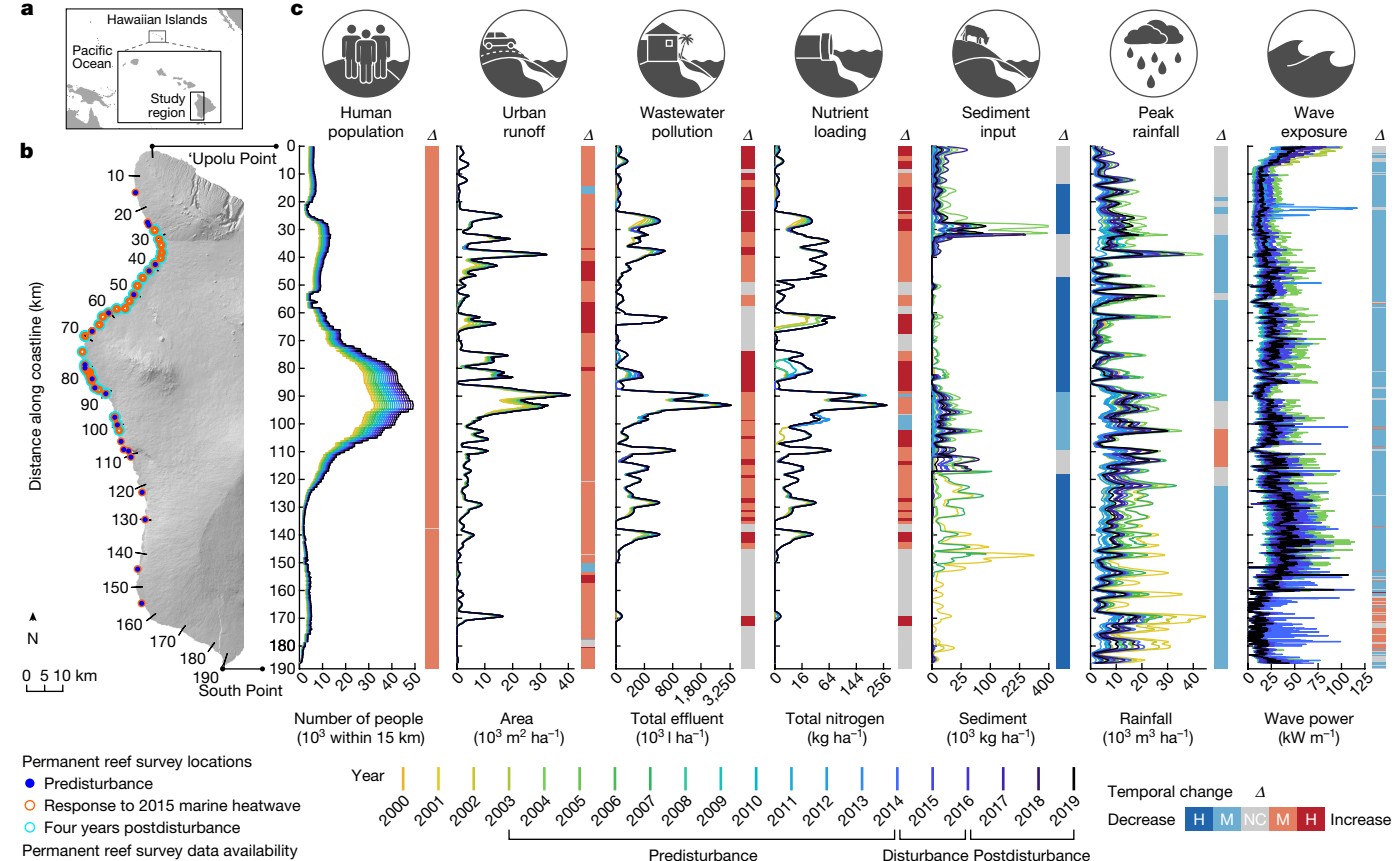

**Fig. 1 | Select local land–sea human impacts and environmental factors on coral reefs in our study region in Hawai'i. a**, Geographic location of the Hawaiian Islands. **b**, Study region with reef surveys shown for the following: reef trajectories predisturbance (*n* = 23; Fig. 2), coral response to the 2015 marine heatwave (*n* = 80; Fig. 3) and coral reefs four years postdisturbance (*n* = 55; Fig. 4). **c**, Spatial distribution in annual, high-resolution (100 m) data on local human impacts and environmental factors from 2000 to 2019 (coloured lines). The *y* axis represents distance along the coastline in kilometres from north to south along the study region in **b**. Vertical bar represents the change over time (*Δ*) for each 100 m section along the coast. A change over time is high (H, *Δ* ≥ 50%), moderate (M, 0 > *Δ* < 50%) or there is no change (NC, grey), with

blue hues indicating decreases and red hues indicating increases. Change is based on the mean difference between the first 5 years (2000–2004) and the most recent 5 years (2015–2019) in the time series. This accounted for year-to-year variability in the episodic nature of factors such as wave exposure, rainfall and sediment input. A subset of factors is shown in **c** owing to space constraints. Additional factors (not shown) include annual rainfall, phytoplankton biomass, ocean temperature (mean and variability), heat stress, irradiance, fishing gear restrictions, depth and metrics of fish biomass. The distribution, change over time and variability of all factors are shown in Supplementary Fig. 1. See Extended Data Table 1 and Supplementary Information for detailed information on local land–sea human impacts and environmental factors.

such as population density[24,25] and reef accessibility[26], or composite indices such as 'water quality'[11] that can be affected by anything from deforestation[27] to aquaculture[28]. Such proxies do not identify the policy levers local resource managers can pull and are less likely to result in management actions or successful conservation outcomes.

Here we present a unique 20-year time series of land–sea human impacts and environmental factors known to affect coral reef ecosystem processes across our study region in the Hawaiian Islands (Fig. 1a). Human factors include urban runoff, wastewater pollution, nutrient loading, sediment input and local restrictions on types of fishing gear. Environmental factors include peak and annual rainfall, wave exposure, variability in ocean temperatures and heat stress, irradiance and phytoplankton biomass. We also incorporate multiple fish biomass metrics that represent the critical role reef fish play in maintaining coral reef ecosystem function[29–31] (see Extended Data Table 1 for a full list of factors). We combined this dataset with recurring, permanently marked and site-specific underwater survey data on coral reef benthic communities (Fig. 1b). Our study reefs spanned large spatiotemporal gradients in land–sea human impacts and environmental factors (Fig. 1c) that are comparable to coral reef ecosystems globally (Extended Data Fig. 1), and which experienced the most severe marine heatwave on record in the

Hawaiian Islands (Extended Data Fig. 2). We quantified drivers of coral reef benthic change at the scale of individual reefs over 12 years before disturbance (2003–2014), during and immediately following the marine heatwave (2014–2016) and four years postdisturbance (2016–2019). Our findings show that simultaneously mitigating local human impacts on both land and sea supports positive coral cover trajectories in the absence of periodic acute disturbance, reduces coral loss during a marine heatwave and promotes coral reef persistence following severe heat stress.

## Reef trajectories predisturbance

Coral cover among reefs surveyed in 2003 was 36.9 ± 2.3% (mean ± s.e.; *n* = 23) and changed by less than 3% in the subsequent years leading up to the 2015 marine heatwave (Fig. 2a). However, coral cover trajectories on individual reefs varied considerably over this time period: 44% of reefs showed a positive trajectory (that is, increased coral cover), 35% of reefs showed a negative trajectory (that is, decreased coral cover) and the remaining reefs showed no change (Fig. 2b). To the best of our knowledge, no acute disturbance occurred that can explain these divergent trajectories. Yet, we did find distinct differences in local conditions between positive and negative trajectory reefs in the years before and

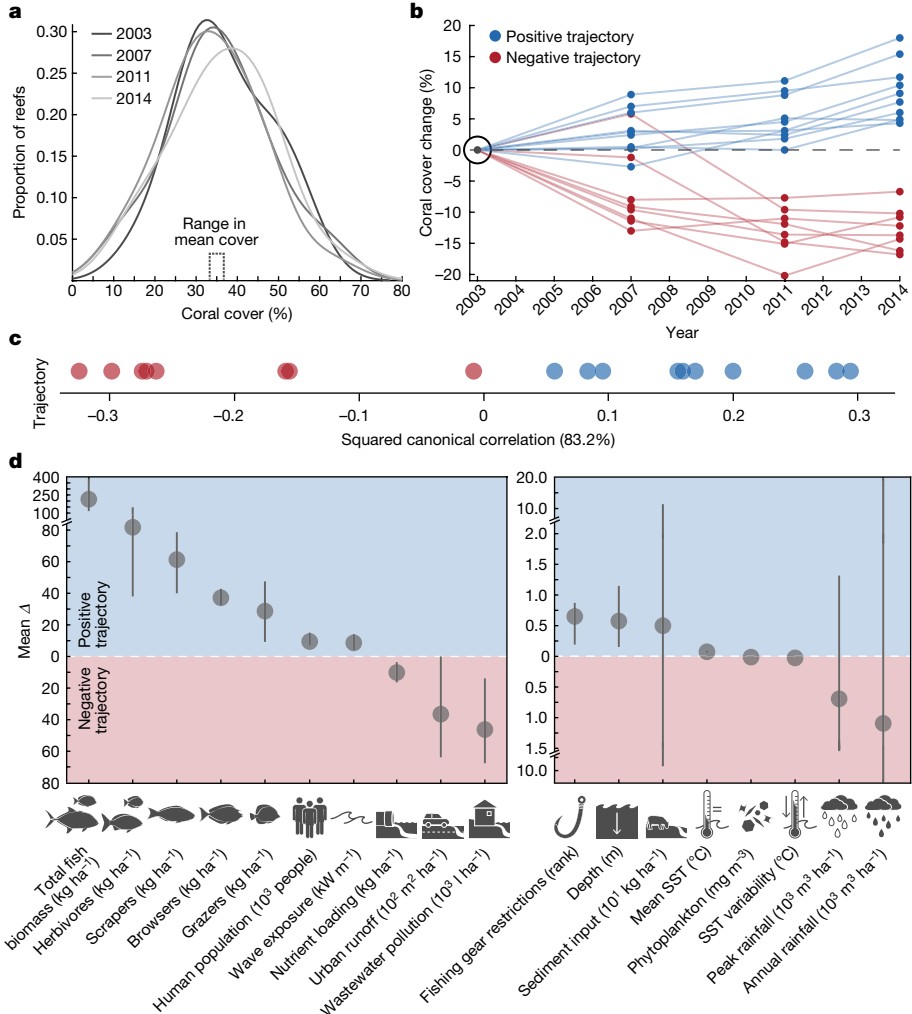

**Fig. 2 | Reef trajectories predisturbance and associated local land–sea human impacts and environmental factors. a**, Coral cover distributions among surveyed reefs between 2003 and 2014 (*n* = 23). **b**, Coral cover trajectories of individual reefs. A reef was considered on a positive trajectory (blue; *n* = 10) or negative trajectory (red; *n* = 8) if coral cover between 2003 and 2014 changed by more than 3%. This cut-off was based on mean coral cover range among all 23 reefs for the 12-year predisturbance period (range 2.8%; min 34.1%; max 36.9%). Reefs with no coral cover change (within ±3%) are not shown. **c**, Difference in local conditions between positive versus negative trajectory reefs (PERMANOVA, pseudo-$F_{1,17}$ = 3.38, *P* = 0.001) visualized along a single multivariate axis (capturing the multidimensional and correlated nature of the data, Supplementary Fig. 2) using a canonical analysis of principal coordinates (*n* = same as in **b**). Allocation success equalled 90 and 87.5% for positive and negative trajectory reefs, respectively (more than 50% indicates an

increasingly more distinct set of conditions than expected by chance alone). **d**, Mean difference (dots) in drop-one jackknife values with upper and lower bars representing the respective maximum and minimum differences in local human impacts and environmental factors between positive and negative trajectory reefs (*n* = same as in **b**). Blue and red shaded regions indicate factors that were greater on reefs that had positive and negative trajectories, respectively. Zero line represents equal values. See Extended Data Fig. 3 for the percentage difference in local conditions between positive and negative trajectory reefs. We included all local human impacts and environmental factors in **d** to provide a general comparison of local conditions between reefs with divergent trajectories. See Fig. 1b for reef locations and Supplementary Fig. 3 for predictor variable distributions. See Methods, Extended Data Table 1 and Supplementary Information for detailed information on local land–sea human impacts and environmental factors.

inclusive of this time frame (Fig. 2c). For example, the average biomass of all fishes, all herbivorous fishes and groups of herbivorous fishes that fill important ecological roles such as scrapers, grazers and browsers[30] were 24–113% (29–214 kg ha⁻¹) greater on reefs with positive trajectories compared to those with negative trajectories (Fig. 2d and Extended Data Fig. 3). These patterns probably reflect positive feedbacks, whereby increasing coral cover promotes habitat suitability for reef fishes, with herbivorous fishes then facilitating coral growth by reducing competitive exclusion by fleshy algae[32]. By contrast, wastewater pollution, nutrient loading and urban runoff were 46–80% greater on reefs with negative trajectories compared to those with positive trajectories. Despite these land-based human stressors being comparatively higher on reefs with negative trajectories, reefs with positive trajectories had 63% greater human population density (the number of people within

a 15 km radius). This finding supports the notion that human population density is a poor indicator of human-driven land–sea impacts at local scales[33]. We observed minimal differences between positive and negative trajectory reefs relative to fishing gear restrictions, depth, sediment input, ocean temperatures, phytoplankton biomass and rainfall. Wave exposure was slightly higher (8.6 kW m⁻¹) on reefs with positive trajectories, but the difference is minor because the entire study region is generally protected from large wave events[34].

## Coral response to the marine heatwave

In 2015, the Hawaiian Islands experienced the strongest marine heatwave on record over the past 120 years (Extended Data Fig. 2). Ocean temperatures across our study region were 2.2 °C above normal and

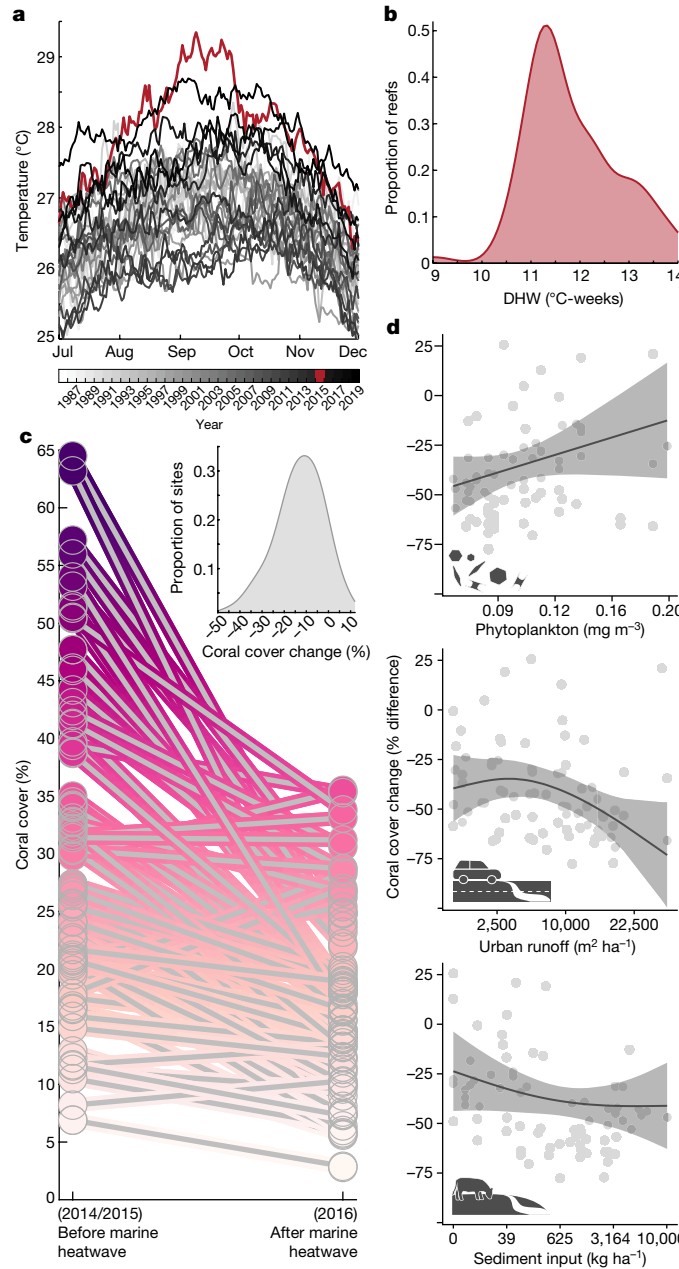

**Fig. 3 | Local land–sea human impacts and environmental factors that modified coral response to the 2015 marine heatwave. a**, Historical (1986–2019) SSTs during the seasonal peak (July–December) averaged across the study region; 2015 marine heatwave shown in red. **b**, Maximum DHW exposure in 2015, a common heat stress metric, among surveyed reefs. All reefs exceeded the eight DHW threshold expected to produce severe and widespread coral bleaching and mortality. **c**, Coral cover before (2014–2015) and one year following (2016) the marine heatwave among surveyed reefs (*n* = 80, Fig. 1b). The inset represents the distribution of absolute coral cover change. **d**, The GAMM results ($R^2$ = 0.79) showing key factors explaining coral response to the marine heatwave. Change accounts for starting condition, defined as: percentage difference = $((A_{a,i} - A_{b,i})/A_{b,i}) \times 100$, where $A_b$ and $A_a$ are the mean coral cover values at each reef in 2014 or 2015, and 2016, respectively (Methods and Supplementary Fig. 4). Positive and negative relationships reduce or increase coral loss, respectively. Shaded regions represent 80% confidence intervals. Factors with the strongest model averaged slopes are shown. Total fish biomass and scraper biomass were also important factors in our models but had weak slopes (representing less than 5% change; Extended Data Fig. 4). Relative importance of factors among all models (that is, sum of AICc model weights across all models containing each factor) were: sediment input (0.99), scraper biomass (0.99), total fish biomass (0.90), urban runoff (0.60), phytoplankton biomass (0.38), wastewater pollution (0.28), peak rainfall (0.20), nutrient loading (0.19), grazer biomass (0.16), DHW (0.08), wave power (0.07), depth (0.06) and fishing gear restrictions (0.05). See Extended Data Table 1 for full list of factors included in the analysis, including those removed that were highly correlated (*r* > 0.7, see Methods and Supplementary Fig. 5). See Supplementary Fig. 6 for predictor variable distributions.

Coral bleaching involves the breakdown of the mutualistic relationship between the coral animal and its algal endosymbionts[36]. A prolonged breakdown in this relationship often results in coral starvation and death, as much of the energetic demands of corals are met by the photosynthetic activity of its endosymbionts[36]. We found that reefs with the highest levels of water column phytoplankton biomass (that is, chlorophyll-*a*) during the marine heatwave showed reduced coral mortality (Fig. 3d). Productivity increases nearshore to tropical islands such as Hawai'i[37] and is further concentrated by small-scale ocean processes that attract dense aggregations of plankton[38]. The increase in nutritional subsidies to the coral animal may have helped to reduce coral starvation during the heatwave or provided higher energetic reserves that promoted their recovery[39]. In other regions (for example, Great Barrier Reef), high levels of chlorophyll-*a* are an indicator of poor water quality that drives negative outcomes for corals[40]. Here, chlorophyll-*a* was uncorrelated to land-based human impacts (Supplementary Fig. 6) and probably reflective of natural gradients in energetic subsidies that facilitated coral survival. Working towards locally relevant management strategies requires understanding how human impacts superimpose on natural biophysical drivers, such as phytoplankton biomass[24], to influence reef ecosystem response to acute disturbance.

Coastal runoff can deliver a broad spectrum of land-based contaminants that degrade nearshore water quality, with cascading effects on coral health[41]. We found that reefs exposed to the lowest levels of urban runoff, and to a lesser extent sediment input, experienced a modest reduction in coral mortality from the marine heatwave (Fig. 3d). Urban runoff often contains heavy metals and petrochemicals that cause coral tissue death[42] and sediment input can impede the photosynthetic capacity of corals and reduce growth by burying coral colonies[41]. Together, these stressors can undermine the natural defence abilities of corals and increase the likelihood of mortality from heat stress[40]. Although turbid waters may shade corals from excessive sunlight that can exacerbate coral bleaching, high levels of heat stress can override any protective benefits decreased light may provide[43]. Existing but underused local and national policies such as the Clean Water Act in the United States provide actionable pathways for marine management

peaked at 29.4 °C (Fig. 3a). Degree heating weeks (DHWs), a widely used heat stress metric for coral reefs, averaged 12 DHWs among surveyed reefs (Fig. 3b), far exceeding the eight DHW threshold expected to cause severe and widespread coral bleaching and mortality[35]. Reef surveys performed one year following the marine heatwave showed that nearly one-quarter of reefs (19 out of 80) lost more than 20% coral cover whereas the hardest-hit reef lost 49% (Fig. 3c). But not all reefs experienced such catastrophic change. Coral cover remained unchanged or increased on 18% (14 out of 80) of reefs surveyed. This divergent ecological response was unexpected given that all reefs were exposed to similarly extreme levels of heat stress (Fig. 3b).

Interactions between heat stress and local conditions such as a high abundance of competitive macroalgae can exacerbate coral bleaching and mortality[22]. However, we lack a detailed understanding of the land- and sea-based factors that mediate coral response to marine heatwaves. Using a generalized additive mixed-modelling framework, we identified the land–sea factors that best explained variations in coral cover change (accounting for starting cover) among reefs one year after the 2015 marine heatwave in Hawai'i (Fig. 3d and Extended Data Table 2).

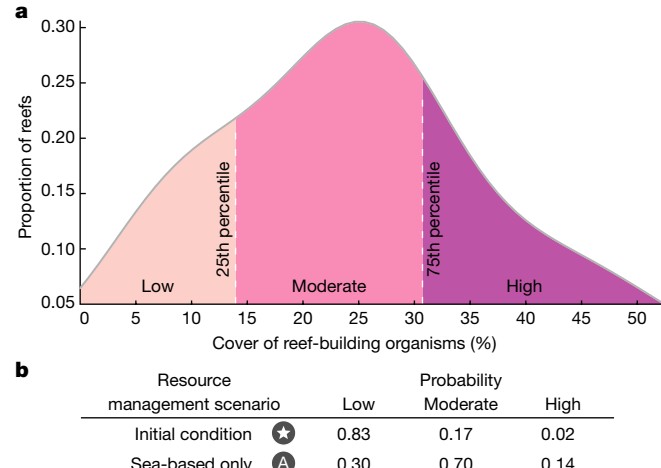

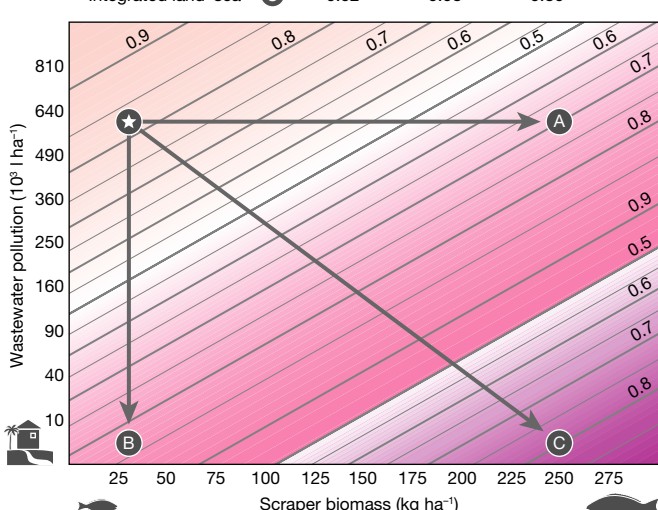

**Fig. 4 | Local management scenarios that support coral reef persistence four years postdisturbance. a**, Percentage cover of reef-building organisms (hard coral + crustose coralline algae) among reefs surveyed (*n* = 55) in 2019, four years following the marine heatwave. Colours represent low (≤25th percentile), moderate (>25th and <75th percentile) or high (≥75th percentile) cover. **b**, Probability of low, moderate or high cover of reef-builders shown in relation to variations in scraper biomass and wastewater pollution. Example scenarios show that simultaneously decreasing wastewater pollution and increasing scraper biomass results in a far greater probability of high reef-builder cover (scenario 'C') than achieving either management scenario in isolation (scenarios 'A' and 'B'). The upper (250 kg ha⁻¹) and lower (30 kg ha⁻¹) management scenarios for scraper biomass represented the 92nd and 36th percentiles, respectively. We specifically chose 250 kg ha⁻¹ as it approximates the long-term mean (2003–2019; *n* = 17) scraper biomass in Kealakekua Bay, a marine protected area in our study region where no fishing has been allowed since 1969 (Supplementary Fig. 11). Similarly, the upper (600,000 l ha⁻¹) and lower (2,500 l h⁻¹) management scenarios chosen for wastewater pollution represented the 95th and 36th percentiles of the 2019 distribution, respectively (Supplementary Fig. 12). Probability values and lines were derived from the top model from ordinal logistic regression modelling (Extended Data Table 3, Methods and Supplementary Information). Colours for low, moderate and high in **b** are the same as those in **a**. See Extended Data Table 1 for full list of local land–sea human impacts and environmental factors included in the analysis, including those removed that were highly correlated (*r* > 0.7, Methods and Supplementary Fig. 8). See Supplementary Fig. 9 for predictor variable distributions.

interventions of land-based stressors[44]. Management strategies that leverage such policies to help mitigate coastal runoff, particularly in urban areas, may support increased coral survival during severe marine heatwaves.

We also found that total fish biomass and scraper biomass were important factors in our models (Extended Data Table 2). Healthy fish populations provide numerous reef-scale ecosystem functions[29], including some species releasing beneficial nutrient subsidies that increase coral thermal tolerance[45]. Scrapers remove fast-growing algal turfs that could otherwise outcompete and overgrow stress-compromised corals[30]. By comparison to phytoplankton biomass and coastal runoff, the slopes of the relationships between total fish biomass and scraper biomass with heat-driven coral loss were weak (Extended Data Fig. 4). Intense marine heatwaves can cause severe coral mortality even on highly protected, uninhabited reefs with intact fish populations[46], suggesting that extreme heat stress may simply overwhelm the functional roles of reef fish over short time scales. However, abundant fish populations, in particular herbivores, can support coral reef recovery potential following disturbance[2]. Understanding whether this positive relationship holds across gradients in land-based impacts is key for supporting targeted fisheries management in coastal marine ecosystems.

## Coral reefs four years postdisturbance

The dominant reef-builders in tropical coral reef ecosystems are hard corals and crustose coralline algae[25]. Crustose coralline algae are encrusting calcifying algae that fuse the reef framework together and promote coral growth by serving as a successional prerequisite for coral recruitment and suppressing competitive fleshy algae[25]. Given that coral cover can take a decade or more to recover to prebleaching levels[47], assessing the total cover of reef-building organisms (hard coral + crustose coralline algae) is more indicative of coral reef recovery potential following disturbance. Our surveys four years following the 2015 marine heatwave found that reef-builder cover ranged from 3.4 to 51.9% (mean of 24.3% ± 1.7 s.e.; *n* = 55; Fig. 4a). Critically, there were different reefs with high (more than or equal to the 75th percentile) and low (less than or equal to the 25th percentile) reef-builder cover before and after the marine heatwave. Nearly two-thirds of reefs with high reef-builder cover in 2019 did not support such levels of cover before the marine heatwave. Similarly, we observed a more than 40% change in the location of reefs with low reef-builder cover between 2015 and 2019. This reshuffling of reefs in terms of relative reef-builder cover suggested differential coral reef persistence in the years following severe heat stress.

We used an ordinal logistic regression framework to identify the local land–sea human impacts and environmental factors that best supported coral reef persistence in the years following the 2015 marine heatwave. Decreased wastewater pollution and increased scraper biomass were the most important and significant (*P* < 0.05) in predicting whether a reef had relatively higher reef-builder cover four years postdisturbance (Extended Data Table 3). Pollution from human waste affects coastal marine ecosystems globally[48] and is especially harmful to corals from untreated sources, such as septic tanks and cesspools, which are both common in Hawai'i[49]. Consequently, high concentrations of toxins and pathogens leach into coastal waters that increase coral disease, reduce coral growth and reproduction, and increase coral susceptibility to bleaching[42]. These negative impacts on coral persistence are therefore much reduced in areas of decreased wastewater pollution. Scrapers reveal bare substrate as they feed and facilitate the settlement, growth and survival of crustose coralline algae and corals following acute disturbance[30]. Beyond these top-down effects on benthic condition, bottom-up effects of improved habitat quality could be contributing to the positive relationship we observed between scraper biomass and higher reef-builder cover. Parrotfish are the dominant scrapers in Hawai'i, and typically have home ranges of

less than 1 km (ref. 50). Furthermore, our scraper biomass estimates were derived from multiple observations across several time points following the marine heatwave, rather than a single snapshot estimate. Such strong site-based fidelity, combined with our recurring surveys, suggests that resident scrapers played a key role in promoting higher reef-builder cover rather than the association driven purely by an influx of individuals seeking more favourable habitat postdisturbance.

Sea-based management efforts are often disconnected from those occurring on land[17,18]. We generated management scenarios of how varying scraper biomass (sea-based management) and wastewater pollution (land-based management) influenced the probability of being in a low, moderate (more than the 25th and less than the 75th percentile) or high reef-builder cover category. Our findings indicate that an integrated management approach can result in a positive synergistic outcome for coral reefs (Fig. 4b). For example, four years following the marine heatwave, a reef across our study region with low scraper biomass (for example, 30 kg ha$^{-1}$) and relatively high wastewater pollution (for example, 600,000 l ha$^{-1}$) is most likely to have low reef-builder cover (83% probability) (Fig. 4b, 'initial condition'). Where scraper biomass is higher (for example, 250 kg ha$^{-1}$) but wastewater pollution remains high, there is a 70% probability of moderate reef-builder cover (scenario A). Conversely, where wastewater pollution is lower (for example, 2,500 l ha$^{-1}$), but scraper biomass remains low, there is an 83% probability of moderate reef-builder cover (scenario B). However, if both land and sea management scenarios occur, there is an 80% probability of high reef-builder cover (scenario C). Combining land and sea management resulted in a three- to sixfold increase in the probability of high reef-builder cover four years following severe heat stress than if land or sea were managed in isolation.

## Conclusion

Here we show that simultaneously mitigating local land- and sea-based human impacts promotes coral reef persistence before, during and in the years following a historically unprecedented marine heatwave in Hawai'i. Our unique spatially and temporally resolved data highlighted the specific impacts that best correlated with coral reef persistence in each of these temporal periods. For example, the biomass of all reef-fish groups was associated with positive reef trajectories over the 12 years leading up to the marine heatwave. By contrast, scraper biomass was the only fish group associated with positive outcomes for reefs four years following severe heat stress. This suggests that reef fish play essential functional roles at different points in time and that particular feeding and behaviours are probably critical for reef persistence following acute distrubance[30]. Similarly, land-based impacts consistently emerged as driving negative coral reef outcomes, but the combination of stressors changed depending on the observational time window in question. Highly resolved data on the local human impacts that drive reef ecosystem trajectories over time are unlikely to be available in most regions. However, our overarching finding that integrated land–sea management benefits coral reefs under ocean warming, is applicable to populated reefs globally.

The local human impacts we identify here represent the direct or proximate drivers of reef condition in our study. These in turn are dictated by an array of distal socioeconomic and cultural factors such as human migration and urbanization, finance, trade and tourism that indirectly affect how people interact with coral reefs[1,51]. Distal human drivers also underpin climate change that is driving severe marine heatwaves that trigger mass coral bleaching at global scales. Increases in future ocean temperatures and the frequency and severity of coral bleaching events[52] could simply overwhelm the positive effects of local management actions on coral reefs. However, there is substantial variation in the projected rates of ocean warming within and among countries under reduced emissions scenarios[52]. Actions that support coral reef persistence locally alongside global reductions

in greenhouse gas emissions may buy reefs more time to adapt and persist into the future. Contemporary governance must therefore shift towards an integrated approach to align management strategies with reef ecosystem processes and the coincident multiscale human drivers that affect them[1].

An ambitious effort is underway to protect 30% of Earth's land and sea areas by 2030 as part of the recently adopted Kunming-Montreal Global Biodiversity Framework[5]. The motivation behind the '30 by 30' is to support ecological resilience, conserve biodiversity and preserve ecosystem services that underpin human well-being[53]. The 30 by 30 has broad participation and is being incorporated into conservation efforts by nations globally. However, our results reveal that sea-based management alone is insufficient to mitigate the full spectrum of local human effects on coastal ecosystems such as coral reefs. These efforts must therefore explicitly couple the respective 30% land–sea targets to realize coastal ocean conservation goals. But in most coastal geographies, 30% protection is impractical and unethical given the high proportion of people that live near and depend on these ecosystems[54]. Instead, mitigating land-based impacts such as wastewater pollution must occur together with fisheries governance for successful conservation outcomes, akin to long-standing indigenous stewardship practices of island ecosystems[16]. Only by adopting coupled land–sea policy measures, alongside global emissions reductions, will coral reef ecosystems and the human communities they support have the best opportunity for persistence in our changing climate.

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

## Methods

### Study site
Hawaiʻi Island (19.55° N, 155.66° W) is the southeastern most island of the Hawaiian Archipelago, located in the northern central Pacific (Fig. 1). The western section has roughly 200 km of coastline predominantly oriented north to south. The coastline contains the longest contiguous reef ecosystem in the main Hawaiian Islands[55] and large gradients in human population, local land–sea impacts and environmental factors that are comparable to reef ecosystems globally (Extended Data Fig. 1). The region represents an ideal study location for resolving the land–sea human impacts driving reef ecosystem change and coral trajectories following acute climate-driven disturbance.

### Reef surveys
Full details related to sampling design, site selection and survey frequency for benthic and reef-fish data collection across our study region are in the Supplementary Information. In brief, underwater visual surveys of benthic assemblages were collated from three monitoring programmes for the following years (number of reefs surveyed are in parentheses): 2003 (23), 2007 (23), 2011 (23), 2014 (40), 2015 (40), 2016 (80), 2017 (80), 2018 (15) and 2019 (55). All benthic surveys used permanently marked pins to ensure the same area of reef was surveyed over time. High-resolution photographs were collected by using photoquadrats at 1 m intervals along 25 m belt-transects ($n = 26$ photographs per transect). Between 30 and 50 random points were overlaid on each photograph and the benthic component under each point was identified to the lowest possible taxonomic level. Percentage cover of the major functional groups at each reef were used in this analysis, namely hard coral and crustose coralline algae. Surveys of reef-fish assemblages were performed along the same permanently marked 25 m transects concurrently with benthic surveys. However, reef-fish surveys were performed more frequently (one to six times per year from 2003 to 2019) than benthic surveys, depending on the reef location and monitoring programme performing the surveys. In all surveys, fishes were identified to species, sized and enumerated. To account for differences among programmes in how researchers surveyed reef fish, counts were calibrated using species and method specific adjustments[56].

### Local land–sea human impacts and environmental factors
**Fish biomass.** The biomass of fishes at a given reef was measured as total fish biomass, herbivore fish biomass and the biomass of browsers, grazers and scrapers[56]. Total fish biomass is an indicator of the overall state of the fish assemblage[57] and is reduced in areas that have increased fishing pressure[58,59]. In Hawaiʻi, non-commercial nearshore fisheries dominate, with people fishing for recreational, subsistence and cultural purposes[60,61]. However, the dominant harvesting modes and magnitude of fishing activities are largely unknown at spatial or temporal scales relevant to this study[62]. As such, we include total fish biomass in part to represent fishing effort on reefs but recognize its shortcomings in capturing reef- and species-specific differences in fishing pressure across our study region. We also include herbivores and subdivisions by feeding guilds that represent important indicators of resilience on coral reefs[30,63,64]. Browsers are defined as herbivores that feed on macroalgae and associated epiphytic material, and are important for reducing the cover of larger, more established macroalgae. Grazers are herbivores that feed largely on small algal turfs, helping to prevent their succession into larger macroalgae, and scrapers are herbivores that closely crop the substrate and open up new space to promote the settlement, growth and survival of crustose coralline algae and corals[30].

We followed established methods for calculating fish biomass[56]. The biomass of individual fishes was estimated using the allometric length–weight conversion: $W = a\text{TL}^b$, where parameters $a$ and $b$ are species-specific constants, TL is total length (cm) and $W$ is weight (g).

Length–weight fitting parameters were obtained from a comprehensive assessment of Hawaiʻi specific parameters[56] and FishBase[65]. Fish species were excluded from fish biomass calculations according to life history characteristics that are not well captured with visual surveys, including cryptic benthic species, nocturnal species, pelagic schooling species and manta rays.

**Human population.** We quantified human population density using NASA Gridded Population of the World v.4 (ref. 66). The dataset is available at 1 km resolution at 5-year intervals. Linear interpolation was used to fill in the missing years and produce annual time steps of human population within 15 km of each 100 m grid cell across our study region (Supplementary Fig. 12).

**Wastewater pollution.** We calculated wastewater effluent (l ha$^{-1}$ yr$^{-1}$) and nitrogen input (kg ha$^{-1}$ yr$^{-1}$) from onsite sewage disposal systems (for example, cesspools and septic tanks) and injection wells (collectively OSDS) in coastal waters at 100 m resolution. Only OSDS located within a modelled one-year groundwater travel time of the coast were included in the analysis and nutrients from OSDS were assumed to flow to the nearest point on the shoreline. Wastewater effluent and nutrient input were estimated on the basis of ref. 67 and discharge rates and nutrient loading according to ref. 68. A Gaussian decay function was used to estimate dispersal offshore, approaching zero at 2 km (Supplementary Figs. 13–15). This same dispersal function was also used for nutrient input, urban runoff, sediment input and rainfall, which are each described below.

**Nutrient input.** We calculated nutrient input (kg ha$^{-1}$ yr$^{-1}$) at 100 m resolution as the combination of total nitrogen from OSDS (Wastewater pollution section above) and golf courses. The total golf course area per watershed was derived from NOAA Coastal Change Analysis Program (CCAP) land-use and land-cover data and Landsat cloud-free composite images created with Google Earth Engine. The golf course area was multiplied by an annual nitrogen application rate of 585 kg ha$^{-1}$ (refs. 69,70) and then by a leaching rate of 32%[71–73] to estimate nitrogen that either runs off or reaches the groundwater. We also imposed a reduction in nitrogen that reached the ocean on the basis of distance inland and used subwatershed catchment data[74] to estimate nutrient transport from golf courses to the coastline (Supplementary Figs. 16–18).

**Urban runoff.** We quantified the total area of impervious surfaces (that is, paved roads, parking lots, sidewalks and roofs) within 10 km of the coastline at 100 m resolution for each year from 2000 to 2017 (Supplementary Figs. 19 and 20). Data were extracted from NOAA CCAP land-use land-cover data from 1992, 2001, 2005 and 2010. We also digitized 2017 impervious surface cover from a single cloud-free Landsat 8 image (courtesy of the United States Geological Survey, USGS) (15 m resolution pan-sharpened). Years in between data availability were filled in by linear interpolation.

**Rainfall.** We quantified annual rainfall (m$^3$ ha$^{-1}$) and peak rainfall (maximum 3-day rainfall total, m$^3$ ha$^{-1}$) at 100 m resolution. Daily rainfall data were generated following refs. 75,76. Rainfall from each rain station was used to derive interpolated surfaces at annual time steps using Empirical Bayesian Kriging in ArcGIS. Subwatershed catchment data[74] were clipped to 0–10 km from the coast and used to calculate rainfall per drainage area (Supplementary Figs. 21 and 22).

**Sediment input.** The Integrated Valuation of Ecosystem Services and Tradeoffs sediment delivery model was used to derive long-term annual average sediment input (kg ha$^{-1}$) reaching the coast[77–80] at 100 m resolution. We then modulated the long-term annual average sediment over time by watershed on the basis of discharge calculated from peak rainfall data (Rainfall section above). Discharge by watershed was

calculated following ref. 81. Sediment load was assumed to scale with discharge according to a approximate ratings curve following ref. 82 (Supplementary Figs. 23 and 24).

**Fishing gear restrictions.** We created a categorical value for local fishing gear restrictions using regulation information and marine managed area boundary designations updated from ref. 80. All regulations were evaluated for prohibition of gear categories in relation to fishing for reef finfish species over time: line fishing, lay nets, spear fishing and aquarium collection. Ranked fishing gear categories are as follows: (1) full no-take, (2) no lay net, spear or aquarium, (3) no lay net or aquarium, (4) no lay net, (5) no aquarium and (6) open to all gear types (Supplementary Table 1 and Supplementary Fig. 25).

**Sea surface temperature and heat stress.** The mean and variability (that is, standard deviation) in summertime sea surface temperature (SST) was calculated over a 90-day window centred on the maximum value of a 7-day moving window average for each SST pixel (Supplementary Fig. 26). Mean regional temperature (Fig. 3a) was calculated by taking the 7-day running mean of daily values and then averaging across all coastal pixels within our study region. Heat stress on reefs during the 2015 marine heatwave was assessed using DHW[35], a widely used metric by coral reef scientists across the world. All data were NOAA's Coral Reef Watch v.3.1, available daily at 5 km resolution[35].

**Phytoplankton biomass and irradiance.** We used satellite derived chlorophyll-$a$ (mg m$^{-3}$; a proxy for phytoplankton biomass) and irradiance (E m$^{-2}$ d$^{-1}$) from two sources. The long-term mean (2002–2013) in 8-day, 4 km data were obtained from ref. 80 and shown in Fig. 2d and Extended Data Fig. 3. All subsequent analysis used the visible-infrared imaging/radiometer suite, which has high spatial (750 m) and temporal (daily) resolution data starting in 2014 (provided by NOAA's Coral Reef Watch). All data were quality controlled and masked to account for cloud cover (Supplementary Information) and optically shallow waters following ref. 83 (Supplementary Fig. 27).

**Wave exposure.** Wave power (kW m$^{-1}$) combines wave height and period and provides a more representative metric of wave exposure than wave height alone[84]. A series of nested grids (from global to 50 m) using WAVEWATCH III[85] and Simulating Waves Nearshore[86] were used to quantify wave transformation over the reef environment at 50 m, at hourly intervals across our study region from ref. 87 and updated for this study. Annual data were then generated for each 50 m grid cell by taking the mean of the top 97.5% in daily maximum wave power (Supplementary Fig. 28).

**Depth.** Depth of the reef floor (m) was measured using diver depth gauges during the in-water reef surveys.

## Statistical analyses

**Coral reef trajectories predisturbance.** We quantified the change in coral cover at 23 reefs from 2003 to 2014. A reef was considered to have a positive trajectory or negative trajectory if coral cover from the 2003 survey to the 2014 survey increased or decreased by greater than 3%, respectively (Fig. 2b). This cut-off was based on the range in mean coral cover among all 23 reefs across the 12-year period (range 2.8%; minimum 34.1%; maximum of 36.9%). We then quantified local human impacts and environmental factors at each reef as follows: fish biomass metrics were from the mean of all annual surveys for each year from 2003 to 2014; human population, wastewater pollution, nutrient loading, urban runoff, annual rainfall, peak rainfall, SST mean and SST variability from the mean of all data from 2000 to 2014. Phytoplankton biomass and irradiance were from the maximum monthly climatology from 2002 to 2013. Sediment and wave exposure came from the mean of the top five events from each year spanning

2000–2014. Fishing gear restrictions were from marine managed area designation at the onset of reef surveys and the depth came from in-water diver-assessed values.

The difference in local human impacts and environmental factors between positive and negative trajectory reefs were then calculated as the difference in the mean drop-one jackknife values for each impact or factor[88]. Upper and lower bars in Fig. 2d represent the respective maximum and minimum differences in drop-one jackknife values between positive and negative trajectory reefs. Before calculating the drop-one jackknife values, we identified and removed outliers that fell outside a threshold of ±2 standard deviations of the median. We formally tested for a difference in the local conditions of positive versus negative trajectory reefs using a multivariate permutational analysis of variance (PERMANOVA)[89] based on a Euclidean distance similarity matrix, type III (partial) sums-of-squares and unrestricted permutations of the normalized data. We visualized the results in Fig. 2c using a constrained analysis of principal coordinates[90] and calculated the cross-validation allocation success (a measure of group distinctness) from the leave-one-out procedure of the constrained analysis of principal coordinates analysis.

**Coral response to the 2015 marine heatwave.** Our goal was to assess the local land–sea human impacts and environmental factors that best explained changes in coral cover as a consequence of the 2015 marine heatwave. Any potential to observe change, however, could be influenced by variations in starting condition. Reefs with higher initial cover (such as those on positive coral cover trajectories predisturbance, Fig. 2b) had greater scope for loss and vice versa[91] (Extended Data Fig. 5). To account for this and ensure comparability across reefs (Supplementary Fig. 4) we calculated coral cover change following ref. 92 as:

$$\%\text{difference}\Delta = [(A_{a,i} - A_{b,i})/A_{b,i}] \times 100$$

where $A_b$ and $A_a$ are the mean coral cover values at each reef in 2014 or 2015, and 2016, respectively.

We then calculated the following predictors based on current literature and our hypotheses of the principal factors that drive changes in coral cover owing to severe heat stress (Extended Data Table 1). Fish biomass metrics included the mean of fish data that were coupled with benthic surveys: 2014 ($n = 40$) or 2015 ($n = 40$) and 2016 ($n = 80$); human population, wastewater pollution, nutrient loading, urban runoff, annual rainfall, peak rainfall and wave exposure were taken from the mean of all data from 2012 to 2016, sediment was measured from the mean of the top three events from 2006 to 2016; SST mean and SST variability were taken from the mean from 2000 to 2014; DHW was the maximum value for 2015; phytoplankton biomass and irradiance was the mean from June to November 2015, representing the time inclusive of the marine heatwave; fishing gear restrictions was the marine managed area designation before the marine heatwave (2014 or 2015, depending on the reef surveyed) and depth came from in-water diver-assessed values.

We then tested for correlations between coral loss and our suite of predictor variables using a generalized additive mixed-effects modelling (GAMM) framework[24] with the gamm4 (ref. 93) package for R (www.r-project.org) v.4.0.2. Before model fitting, we identified the presence of outliers in our predictor variables as any point that fell outside a threshold of ±2 standard deviations of the median. We then applied an additional step to retain any point above this threshold that was within 25% of the maximum predictor value below the threshold. This ensured that no data points were unnecessarily discarded from our formal model-fitting process because of applying an arbitrary threshold cut-off for data inclusion. The following predictors were square-root transformed to down-weight the influence of values at the extreme ends of their distributions: all fish biomass metrics, wastewater

pollution, urban runoff, nutrient loading, phytoplankton biomass and peak rainfall. A fourth-root transformation was applied to sediment.

To reduce model overfitting, Pearson's correlation coefficients were calculated among all predictors (Supplementary Fig. 5), removing one of each pair of highly correlated ($r > 0.7$) predictors. To further strive for model parsimony, we a priori excluded human population density from the model-fitting process as it was a poor indicator of human-driven land-to-sea impacts on local scales (Figs. 1c and 2d and Extended Data Fig. 3). We also excluded browser biomass as they represented less than 10% on average of total herbivore biomass across all reefs before, during and postdisturbance. This resulted in the following predictors included in the models (correlated predictors in parentheses were removed): total fish biomass, biomass of scrapers, biomass of grazers (total herbivore biomass), DHW (SST mean and variability), wastewater pollution, nutrient input, urban runoff, sediment input and peak rainfall (annual rainfall correlated with both), wave power, phytoplankton biomass (irradiance), fishing gear restrictions and depth. The decision of which correlated predictors to retain was based on a hypothesis-driven approach, in part whether the given predictor had the potential to directly (for example, sediment input) rather than indirectly (for example, annual rainfall driving sediment input) affect heat-driven coral loss.

We incorporated a random spatial factor to account for the possible influence of a change in an underlying variable along the coastline not quantified in this study. This was done by breaking the coastline up into discrete 10 km sections running north to south. Section size was determined using hierarchical clustering based on pairwise Euclidean distances between reefs and identifying an inflection point in the intragroup variance[24] (Supplementary Fig. 7). We fitted GAMMs for all possible candidate models (unique combinations of the predictor variables) using the UGamm wrapper function, in combination with the dredge function in the MuMIn package[94]. Nonlinear smoothness in the models was determined using penalized cubic regression splines, with the number of knots (limited to four to reduce overfitting) spread evenly throughout each covariate. All possible candidate models were computed (unique combinations of the predictor variables) but limiting the total number of predictors in any given candidate model to five to reduce overfitting. We used Akaike's information criterion with a bias correction for small sample sizes[95] (AICc) for model comparison and all models within $\Delta$AICc $\leq 2$ of the top model ($\Delta$AICc $= 0$) are presented in Extended Data Table 2. To visualize the effect of predictor terms on coral cover change, we averaged the coefficients from the top models (that is, $\Delta$AICc $\leq 2$) to generate a predicted dataset and set all other predictor terms to their median value. Finally, we calculated a measure of predictor variable relative importance within each candidate model by calculating the sum of AICc model weights for each predictor (that is, the sum of model weights across all models containing each predictor; Fig. 3).

**Coral reefs four years postdisturbance.** Our goal was to assess the local land–sea human impacts and environmental factors that best explained variations in the cover of reef-building organisms four years following the marine heatwave. The cover of reef-building organisms for reefs surveyed in 2019 ($n = 55$) were parsed into three categories on the basis of the following percentiles: low, less than or equal to the 25th; moderate, more than 25th and less than 75th; and high, more than or equal to the 75th. We then performed ordinal logistic regression[96] to determine the probability of a given reef having high, moderate or low cover of reef-building organisms on the basis of the prevailing local human impacts and environmental factors (that is, predictor variables; Extended Data Table 1). Logit models are multivariate extensions of generalized linear regression models that provide parameter estimates by means of maximum likelihood estimation (MLE) to model the relative log odds of observing a reef-builder cover category or less versus observing the remaining higher categories:

$$\ln\left(\frac{P(y_i \leq j)}{P(y_i > j)}\right) = C_j + B_1 z_{i1} + \cdots + B_k z_{ik}$$

Here, $i$ indexes each of $N$ observations, with categories $y_i$, and the left-hand side of the equation is the logit of the probability of a reef-builder category of $j$ or lower, for $j = 1$ (high) or 2 (moderate). Reefs with low reef-builder cover contributed to the regression through calculation of the log odds. Each $C_j$ is an MLE-computed model intercept, and each $B_k$ is the MLE coefficient corresponding to the standardized independent variable $z_{ik}$, for $k = 1$ through $n$, where $n$ is the variable number of predictors used in a given candidate model, hence the ellipsis (...). A fundamental component of this model is the assumption of proportional odds, or parallel regression, which indicates that $B_k$ values are independent of the logit level $j$. The validity of this parallel regression assumption was ascertained using Brant's Wald test[97], as well as a likelihood ratio test ($\alpha = 0.05$).

We then calculated the following predictors based on current literature and our hypotheses of the principal factors that drive changes in reef-builder cover across space and time following a major thermal disturbance: fish biomass metrics, wastewater pollution, nutrient loading, urban runoff, annual rainfall, peak rainfall, wave exposure, phytoplankton biomass and irradiance: the mean of all data from 2016 to 2019; sediment was measured as the mean of top three events over the 2006–2019 time period; SST mean and SST variability: mean of all data from 2000 to 2018. Note that 2019 was excluded in SST mean and SST variability owing to the marine heatwave that affected Hawai'i[21], but occurred after our 2019 fish and benthic surveys; fishing gear restrictions involved the marine managed area designation in 2016 and depth was assessed by in-water diver-assessed values.

We used the same process as in the GAMM analysis to remove outliers in our predictor variables (above). We then square-root transformed the following predictors to down-weight the influence of values at the extreme ends of their distributions: total fish biomass, wastewater pollution, sediment input and nutrient loading. Pearson's correlation coefficients were calculated among all predictors (Supplementary Fig. 8), removing highly correlated ($r > 0.7$) predictors. For the reasons outlined in our GAMM analysis and for continuity, we a priori excluded human population density and the biomass of browsers from the model-fitting process. This resulted in the following predictors included in the models (correlated predictors in parentheses were removed): total fish biomass, biomass of scrapers, biomass of grazers (total herbivore biomass), wastewater pollution, nutrient input, sediment input, urban runoff (phytoplankton biomass), wave exposure, fishing gear restrictions and depth. The decision of which correlated predictors to retain followed the same logic as our GAMM analysis. The mean and variability in SST were excluded given the negligible range of values among reefs (0.1 and 0.025 °C, respectively). All possible candidate models were computed while limiting the total number of predictors in any given candidate model to four (to reduce overfitting and to account for the lower response variable replication compared to our GAMM analysis). Models were computed using the multinomial logistic regression function mnrfit in MATLAB. We again used AICc for model comparison and all models within $\Delta$AICc $\leq 2$ of the top model ($\Delta$AICc $= 0$) are presented in Extended Data Table 3. McFadden's pseudo-$R^2$ was computed for the highest ranked models and ranged from 0.21 to 0.22. Unlike traditional $R^2$ values, McFadden's pseudo-$R^2$ of more than 0.2 represents an excellent fit[98]. Models within $\Delta$AICc $\leq 2$ of model 1 in Extended Data Table 3 demonstrated comparable levels of goodness of fit and parsimony[99,100]. Many of the parameter coefficients within these models were sensitive to the underlying variability in the data and their estimates did not differ significantly from zero ($P < 0.05$). The top model contained parameters with covariate estimates significantly different from zero, namely scraper biomass and wastewater pollution. Using model 1, we examined changes in the probability of a

given reef having high (more than or equal to the 75th percentile), moderate (more than the 25th and less than the 75th percentile) or low (less than or equal to the 25th percentile) reef-builder cover (Fig. 4a) on the basis of variations in these two land–sea predictors (Fig. 4b). Probability curves for high, moderate and low were calculated on the basis of changing scraper biomass and wastewater pollution and holding all other predictors at their mean.

**Resource management scenarios.** The resource management scenarios presented in Fig. 4b were selected on the basis of the following rationale. We chose 250 kg ha$^{-1}$ as the management target for scraper biomass as this value approximates the long-term mean (2003–2019; $n$ = 17) biomass of scrapers within Kealakekua Bay, a marine protected area where no fishing has been allowed since 1969 (Supplementary Fig. 10). Kealakekua Bay is also exposed to numerous land-based stressors, including high levels of wastewater pollution (258,000 l h$^{-1}$ in 2019). As such, our value of 250 kg ha$^{-1}$ represents an estimate of scraper biomass on a reef with strong fisheries protection but with land-based stressors present. In addition, we compared our upper (250 kg ha$^{-1}$) and lower (30 kg ha$^{-1}$) scraper biomass values to the distribution of scraper biomass among all reefs ($n$ = 80) in 2019, the most recent time point in which all reefs were surveyed within the same year (Supplementary Fig. 10). The upper and lower limits represent the 92nd and 36th percentiles, respectively. For wastewater pollution, we used our 2019, 100 m grid cell values that fell along the 10 m isobath (same as Fig. 1c) but constrained the latitudinal extent to be consistent with the northern- and southern-most locations of the 2019 reef surveys. This approach provided far greater replication and a more representative assessment of wastewater pollution along the coastline for which to assess our management scenarios. The upper (600,000 l ha$^{-1}$) and lower (2,500 l h$^{-1}$) values chosen for wastewater pollution represented the 95th and 36th percentiles of the 2019 distribution, respectively (Supplementary Fig. 11).

### Reporting summary

Further information on research design is available in the Nature Portfolio Reporting Summary linked to this article.

## Data availability

All data that support the findings of this study are available at https://github.com/jamisongove/Coral-Reef-Persistence. Reef fish length–weight parameters were obtained from FishBase (https://fishbase.org) and ref. 56, human population data from NASA Gridded Population of the World v.4 (https://sedac.ciesin.columbia.edu/data/set/gpw-v4-population-count-rev11), land-use and land-cover data from the NOAA Coastal Change Analysis Program (https://www.coast.noaa.gov/htdata/raster1/landcover/bulkdownload/), soils data from USDA Gridded Soil Survey Geographic Database (gSSURGO; https://www.nrcs.usda.gov/resources/data-and-reports/gridded-soil-survey-geographic-gssurgo-database), subwatershed catchment data from USGS Stream Stats (https://water.usgs.gov/GIS/metadata/usgswrd/XML/ds680_archydrohucs.xml)[74], watershed and digital elevation model data from USGS National Hydrography Dataset (https://www.usgs.gov/national-hydrography/national-hydrography-dataset), rainfall data from refs. 75,76, Landsat 8 satellite image from USGS (https://earthexplorer.usgs.gov/), Landsat 7 and 8 cloud-free composites derived using Google Earth Engine (https://earthengine.google.com/), individual wastewater systems for Hawai'i from refs. 101,102, marine managed area designation from ref. 80 and downloadable from the State of Hawai'i (https://planning.hawaii.gov/gis), fishing regulations from the State of Hawai'i (https://dlnr.hawaii.gov/dar/fishing/fishing-regulations/), SST and DHW data from NOAA Coral Reef Watch (https://coralreefwatch.noaa.gov/product/5km), ocean colour (chlorophyll-*a* and irradiance) data from NOAA Coral Reef Watch (https://coralreefwatch.noaa.gov/product/oc/index.php) and ref. 80. See Methods and Supplementary Information for more detailed information on the data used to support the findings of this study.

## Code availability

Statistical analyses were performed using the software packages R (www.r-project.org) v.4.0.2 (using libraries gamm4, MuMIn, foreach, doMC, ggplot2, gmt, tidyverse, zoo and lubridate)[103], MATLAB (www.mathworks.com) using v.2021a (using Statistics and Machine Learning toolbox), ArcGIS Desktop (www.esri.com) v.10.6 with Advanced licensing and extensions Spatial Analyst and Geostatistical Analyst, Integrated Valuation of Ecosystem Services and Tradeoffs Sediment Delivery Ratio model (https://naturalcapitalproject.stanford.edu/software/invest) and the PERMANOVA+ (ref. 89) add-on for Primer v.7 (ref. 104). Code is available for download at https://github.com/jamisongove/Coral-Reef-Persistence.

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

**Acknowledgements** We thank A. Dillon of Aline Designs for graphics support, the numerous divers, boat drivers and support staff from the Hawai'i Division of Aquatic Resources, National Park Service and The Nature Conservancy for logistical and data collection support, individuals from the Golf Course Superintendents Association of America, M. Johnson and R. Doog for their input in determining the golf course nitrogen application rates, R. Longman for contributing updated rainfall data, E. Darling for providing data layers from ref. 57, P. Neubauer and Y. Eynaud for analytical advice, W. Walsh for being an early supporter of this effort and J. Link and J. Samhouri for their review of an early version of the manuscript. This research was supported by NOAA's Integrated Ecosystem Assessment Program (contribution no. 2022_7), NOAA's Fisheries and The Environment Program, NOAA's Pacific Islands Fisheries Science Center, and grants from the NOAA Coral Reef Conservation Program (grant no. NA16NOS4820059) and National Marine Fisheries Service, Office of Habitat Conservation (grant nos. EA133F17SE1203 and NA17NMF4630301) and the Harold K.L. Castle Foundation.

**Author contributions** J.M.G., G.J.W. and J.L. conceived the study. J.M.G., G.J.W. and J.L. developed and implemented the analyses. J.M.G. and G.J.W. led the writing of the manuscript with G.P.A. and J.L. All other authors made substantive contributions to the manuscript and contributed data that were central to this effort.

**Competing interests** The authors declare no competing interests.

**Additional information**

**Correspondence and requests for materials** should be addressed to Jamison M. Gove or Gareth J. Williams.

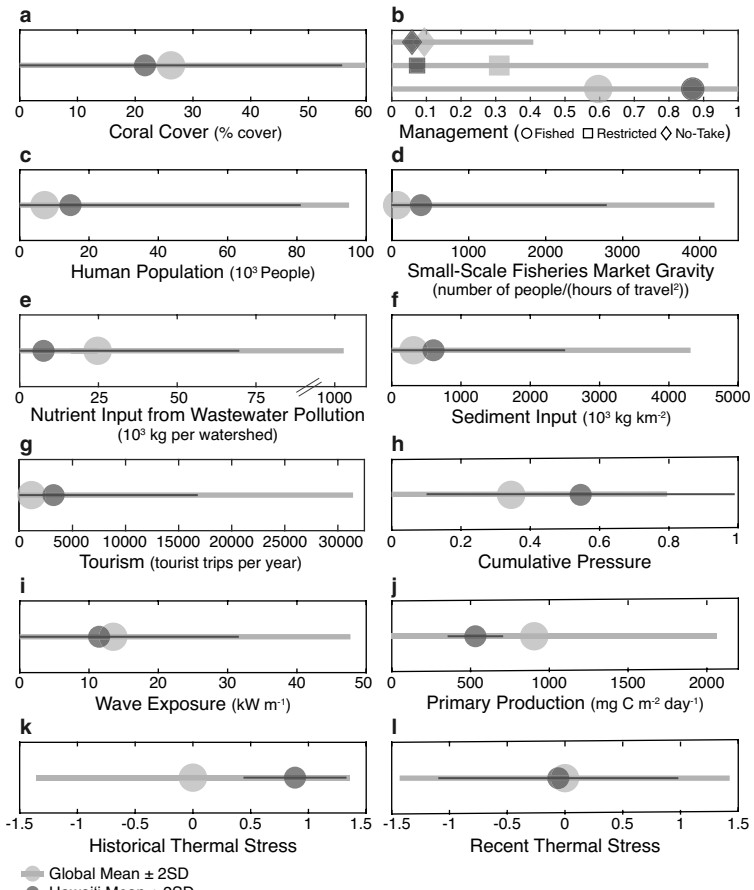

**Extended Data Fig. 1 | Comparison of human, environmental, and climate factors for reefs in Hawai'i with coral reef ecosystems globally.** Dots represent global (light grey) and Hawai'i (dark grey) mean values. Error bars represent the mean ± 2 standard deviation (SD). Factors presented are: **a**, Coral cover (per cent hard coral; global n = 2,584; Hawai'i n = 137); **b**, Proportional reef area by country that is open (fished), gear restricted (restricted), or fully restricted (no-take) to fishing (sample number same as in **a**); **c**, Human population within 5 km in 2018 (global n = 54,596; Hawai'i n = 199); **d**, Small-scale fisheries market gravity (number of people/(hours of travel)$^2$ represents human use and fishing pressure related to the size and accessibility of coral reefs to nearby human settlements and markets[59] (n = same as in **c**); **e**, Annual input of nitrogen (10$^3$ kg) per watershed from wastewater pollution on coral reefs (global n = 38,033; Hawai'i n = 324); **f**, Sediment input (10$^3$ kg km$^{-2}$) to coral reefs (n = same as in **c**); **g**, Annual number of tourist visits driven by coral reefs

combining on-reef (e.g., recreational diving and snorkelling) and reef-adjacent (e.g., provision of calm waters, sand beaches, views, and seafood) aspects (sample number same as in **c**); **h**, Cumulative pressure score from stressors to coral reefs per 5 km reef containing pixel (unitless); **i**, Mean wave energy, or wave power (kW m$^{-1}$), from 1979–2009 (n = same as in **a**); **j**, Mean primary productivity (mg C m$^{-2}$ day$^{-1}$) between 2003–2013 (n = same as in **a**); **k-l**, Unitless metric of (**k**) historical (1985–2017) and (**l**) recent (2014–2017) thermal stress on coral reefs, whereby positive values represent more desirable (i.e., less thermal stress) over the respective time frames[105] (n = same as in **c**). The mean for reefs in Hawai'i falls within 2 SD of the global mean for all factors. Data are from the following sources: **a,b,i,j** from[106] **e** from[48]; **c,d,f,g,h,k**, from[107]. Factors presented here were obtained from global datasets and will differ from those presented within our present study owing to the methodological differences as well as differences in their spatiotemporal extent and resolution.

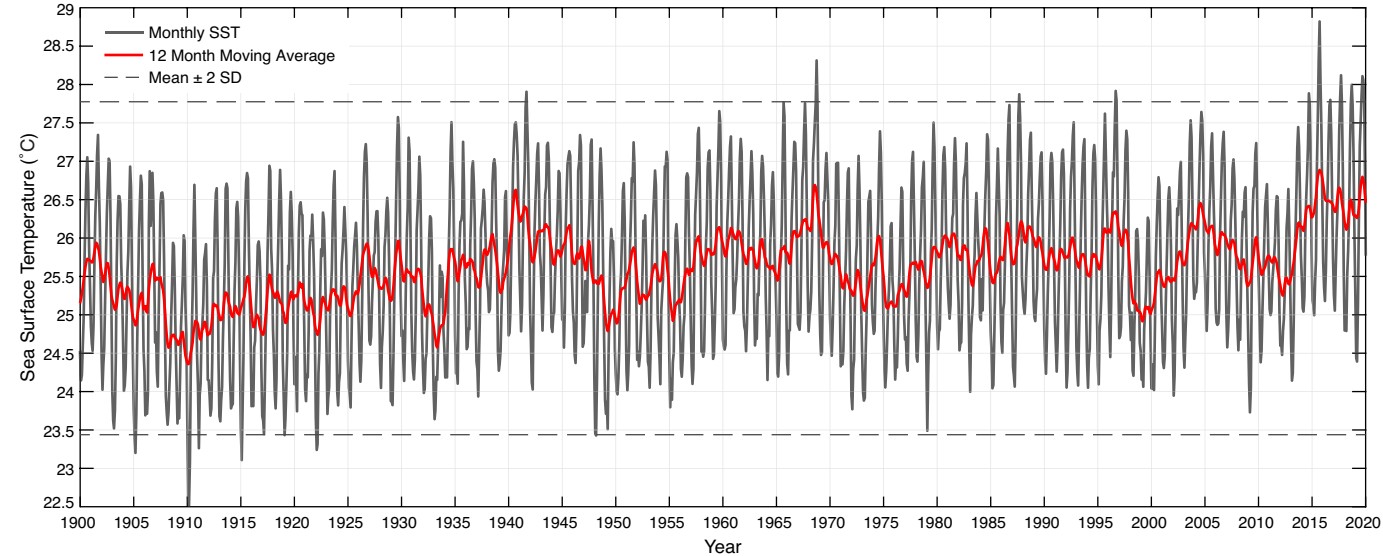

**Extended Data Fig. 2 | Long-term ocean temperature record for Hawaiʻi.** Monthly sea surface temperature (SST) for the main Hawaiian Islands from 1900–2020. Dashed lines represent ± 2 standard deviations (SD) above and below the long-term mean. Red line is the 12-month moving average. The 2015 marine heatwave is represented by the highest SST values over the 120-year time series. Data are from NOAA's Extended Reconstructed SST v5 (https://www.ncei.noaa.gov/products/land-based-station/noaa-global-temp) and values shown are the 90th percentile of monthly SST from within the vicinity of the main Hawaiian Islands (18.5 to 22.5°N; −160.5 to −154.5°W).

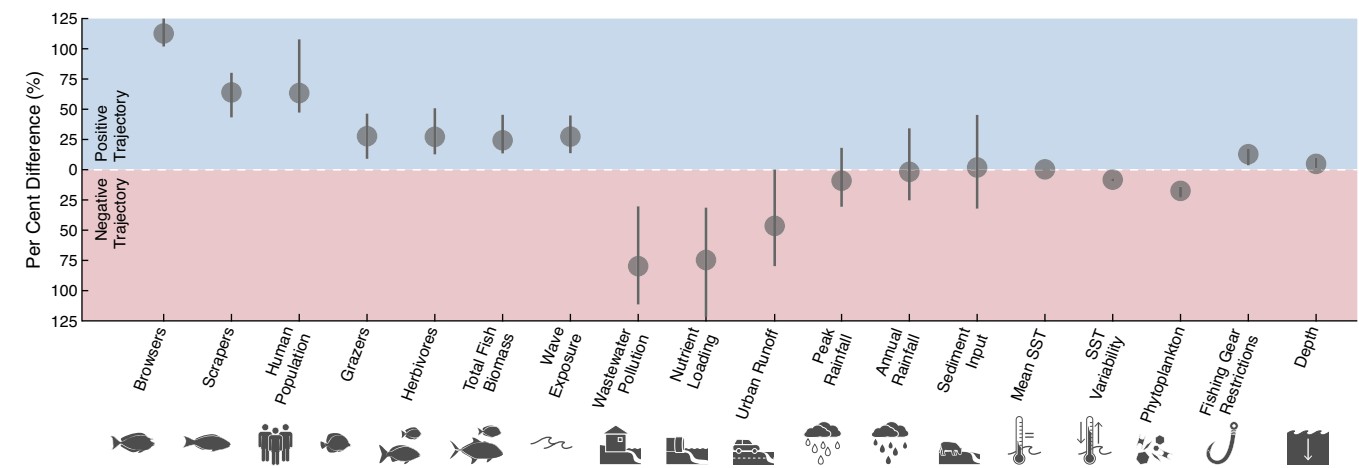

**Extended Data Fig. 3 | Per cent difference in mean drop-one jackknife values of local human impacts and environmental factors between positive and negative trajectory reefs.** The per cent difference ((V1−V2)/[(V1+V2)/2]; dots) was quantified by taking the ratio of the mean in drop-one jackknife values between positive (*n* =10) and negative (*n* = 8) trajectory reefs (*sensu*)[88]. Upper and lower bars represent the respective maximum and minimum per cent differences. Blue and red shaded regions indicate factors that were greater on reefs that had positive and negative trajectories, respectively. Zero line represents equal values. Outliers that fell outside a threshold of ±2 standard deviations of the median were removed prior to analysis. See Fig. 1b for reef locations and Fig. 2d for mean absolute differences in factor values. See Methods, Extended Data Table 1, and Supplementary Information for detailed information on local land-sea human impacts and environmental factors.

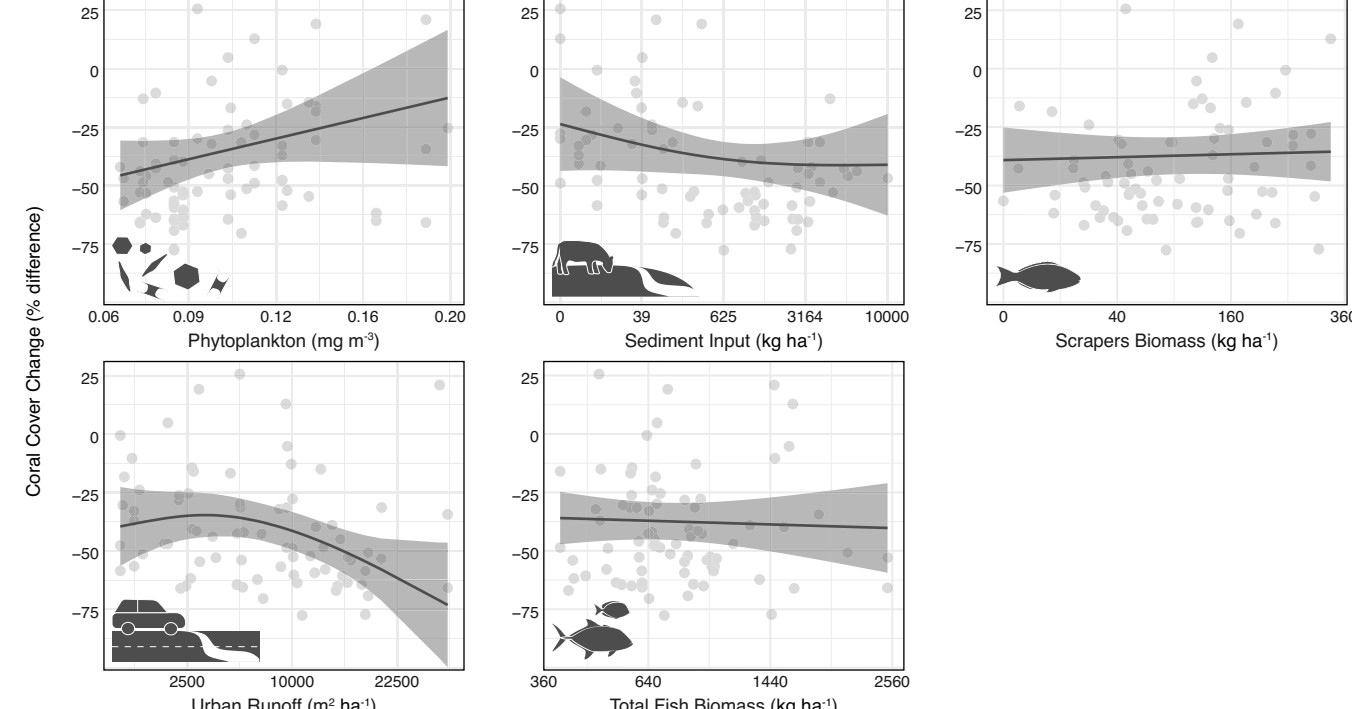

**Extended Data Fig. 4 | Generalized Additive Mixed Model (GAMM) results (R² = 0.79) showing key local land-sea human impacts and environmental factors that modified coral response to the 2015 marine heatwave.** Positive and negative relationships reduce or increase coral loss, respectively. Because changes in coral cover following disturbance can be affected by variations in starting condition (reefs with higher initial cover have greater scope for loss, and vice versa)[91] we modelled relative coral cover change following ref. 92 to ensure comparability across reefs (see Methods). Median values with shaded region representing the 80% confidence interval. The relative importance of factors among all models (i.e., sum of AICc model weights across all models containing each factor) was as follows: sediment input (0.99), scraper biomass (0.99), total fish biomass (0.90), urban runoff (0.60), phytoplankton biomass (0.38), wastewater pollution (0.28), peak rainfall (0.20), nutrient loading (0.19), grazer biomass (0.16), DHW (0.08), wave power (0.07), depth (0.06), and fishing gear restrictions (0.05). See Extended Data Table 1 for full list of local land-sea human impacts and environmental factors included in the analysis, including those removed that were highly correlated (Fig. S5). See Fig. S6 for predictor variable distributions.

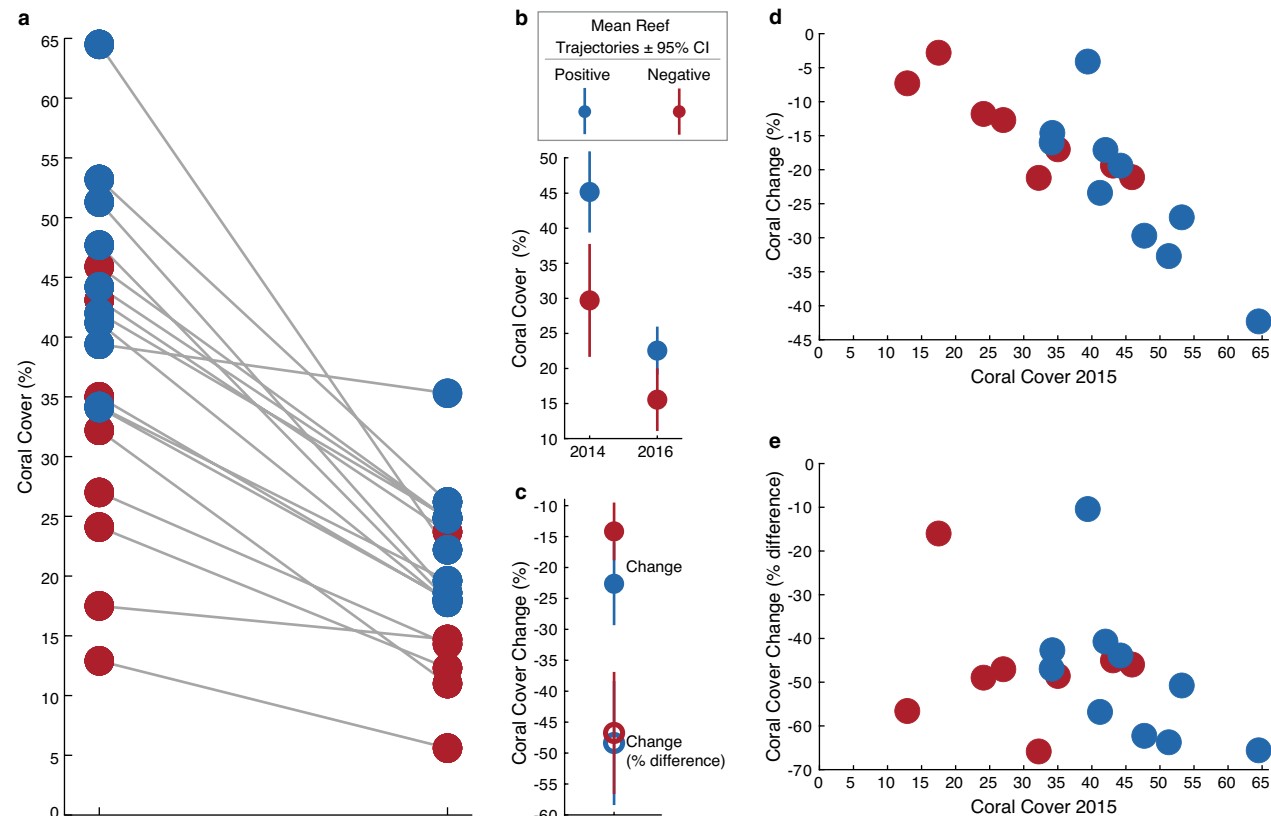

**Extended Data Fig. 5 | Coral cover change following the 2015 marine heatwave on positive versus negative coral cover trajectory reefs. a**, Coral cover on positive (blue; *n* = 10) and negative (red; *n* = 8) trajectory reefs surveyed (see Fig. 2b in main manuscript) prior to (2014) and 1-year following (2016) the marine heatwave. **b**, Positive trajectory reefs have a higher mean coral cover both prior to, and to a lesser extent, following the marine heatwave compared to negative trajectory reefs. **c**, Positive trajectory reefs experience increased absolute coral cover loss following the marine heatwave (underlying relationship shown in panel **d**), but this difference is largely absent once starting coral cover condition is accounted for (underlying relationship shown in panel **e**).

# Extended Data Table 1 | Local land-sea human impacts and environmental factors considered for our analyses

| Impact or Factor | Metric | Units | Spatial Resolution | Temporal Resolution | Temporal Range | Data Source | Justification |
|---|---|---|---|---|---|---|---|
| Fish Biomass | Total Biomass, Herbivores, Grazers, Browsers, Scrapers | kg ha$^{-1}$ | 25 m | 1–6 surveys, per site, per year | 2003–2019 | See Supplemental Information | Abundant reef fish populations support reef-scale ecosystem functions such as predation and nutrient release[29]. Herbivorous fishes support ecosystem resilience and mitigate the negative effects fleshy algae have on coral survival[108]. |
| Human Population | Population Density | people/15 km | 1 km | Annual | 2000–2019 | NASA GPWv4 | Human population density is a widely used proxy for local human impacts[25,109,110]. |
| Wastewater Pollution | Total Effluent | L ha$^{-1}$ | 100 m | Annual | 2000–2019 | Modified from ref[67] | High concentrations of toxins (e.g., endocrine disruptors, pathogenic bacteria and viruses, pharmaceuticals, and heavy metals) are found in wastewater pollution[111]. These toxins can drive a higher incidence of coral disease, reduced coral growth and reproduction, increased cover of fleshy algae, and increased coral bleaching and subsequent mortality[42]. |
| Nutrient Input | Total Nitrogen | kg ha$^{-1}$ | 100 m | Annual | 2000–2019 | See Supplemental Information | Human-derived nutrient input can promote rapid algal growth, outcompeting corals and disrupting ecosystem function[57]. |
| Urban Runoff | Impervious Surfaces | m$^2$ ha$^{-1}$ | 100 m | Annual | 2000–2019 | See Supplemental Information | Runoff can deliver a broad spectrum of land-based contaminants (e.g., heavy metals and household chemicals) that degrades nearshore water quality, with cascading effects on coral health[41], including the natural defence abilities of corals and increase the likelihood of mortality from heat stress[40]. |
| Sediment Input | Sediment | kg ha$^{-1}$ | 100 m | Annual | 2000–2019 | See Supplemental Information | Sediment input can impede the photosynthetic capacity of corals and reduce growth by burying coral colonies[41]. |
| Rainfall | Annual & Peak Rainfall | m$^3$ ha$^{-1}$ | 100 m | Annual | 2000–2019 | Modified from refs[75,76] | Large pulses of freshwater from storm events can cause localised die-off of nearshore corals, fish, and other reef-associated organisms[112]. Rain events can also mobilise high levels of nutrients, sediment, and land-based debris that impact nearshore water quality and coral reef health[113]. |
| Wave Exposure | Wave Power | kW m$^{-1}$ | 50 m | Hourly | 2000–2019 | See Supplemental Information | Gradients in wave exposure and associated flow produce varying levels of disturbance that can play a major role in determining coral reef community patterns[84,114,115]. |
| Phytoplankton Biomass | Chlorophyll-$a$ | mg m$^{-3}$ | 750 m & 4 km | Daily & 8-day | 2002–2014 | NOAA's Coral Reef Watch), and ref[80] | Chlorophyll-a is a widely used indicator for changes in phytoplankton production[116] that propagates through the food-web[117]. Corals can supplement nutritional requirements through heterotrophic feeding on zooplankton[39]. High levels of chlorophyll-a are also indicative of poor water quality that can have negative outcomes for corals[40]. |
| Sea-surface Temperature (SST) | Summertime Mean & Standard Deviation | °C | 5 km | Daily | 2000–2019 | NOAA's Coral Reef Watch | The mean and variability in summertime ocean temperature is a widely used metric for coral reef resilience[57]. |
| Heat Stress | Degree Heating Week | °C-weeks | 5 km | Daily | 2000–2018 | NOAA's Coral Reef Watch | Degree heating week is the accumulation of heat stress above the coral bleaching threshold over a 12-week period as is the dominant metric in coral reef research to quantify heat stress on corals (e.g.,[22]). |
| Irradiance | Photosynthetically Active Radiation | Einstein m$^{-2}$ d$^{-1}$ | 750 m | Daily | 2015–2019 | NOAA's Coral Reef Watch | Excessive irradiance can cause light stress that exacerbates coral bleaching[118,119]. |
| Fishing Gear Restrictions | Gear Rank | Categorical | NA | NA | 2000–2019 | See Supplemental Information | We use fishing gear restrictions as a metric to represent spatial fisheries management[120]. |
| Depth | Depth | Metres | NA | NA | 2003–2019 | Reef Surveys | Deeper reefs are often less impacted by heat stress compared to those located in shallower depths[2,21]. |

See 'Local land-sea human impacts and environmental factors' section in Supplemental Information for detailed information on calculating each impact or factor, including data collection methods, data sources and ancillary data sets, and specific tools or software utilized[108–120].

**Extended Data Table 2 | Summary of generalized additive mixed effects models (GAMM) relating coral response to the 2015 marine heatwave with local land-sea human impacts and environmental factors**

| Significant Land-Sea and Environmental Factors | Log Likelihood | AICc | ΔAICc | Adjusted R² |
|---|---|---|---|---|
| Model 1. Phytoplankton Biomass, Urban Runoff, Sediment Input, Scraper Biomass, Total Fish Biomass | -302.98 | 638.34 | 0 | 0.79 |
| Model 2. Urban Runoff, Sediment Input, Peak Rainfall, Scraper Biomass, Total Fish Biomass | -303.93 | 640.25 | 1.91 | 0.78 |

The top two candidate models are shown. AICc, Akaike's information criterion corrected for small sample size; ΔAICc, change in AICc across the candidate models (ΔAICc ≤ 2 of the top model are shown); Adjusted R², proportion of variation in the response variable explained by the candidate model.

**Extended Data Table 3 | Summary of ordinal logistic regression (OLR) models relating the per cent cover of reef-builders (hard coral + crustose coralline algae) four years following the 2015 marine heatwave to local land-sea human impacts and environmental factors**

| | Model Output | | | | | AICc | ΔAICc | McFadden's pseudo-$R^2$ |
|---|---|---|---|---|---|---|---|---|
| **Model 1** | Predictors | **Scraper Biomass** | **Wastewater Pollution** | Sediment Input | Peak Rainfall | 103.28 | 0 | 0.22 |
| | Coefficients | 0.689 | -0.867 | 0.159 | 0.547 | | | |
| | p value | 0.031 | 0.019 | 0.647 | 0.130 | | | |
| **Model 2** | Predictors | **Scraper Biomass** | Wastewater Pollution | Urban Runoff | Sediment Input | 104.72 | 1.44 | 0.21 |
| | Coefficients | 0.705 | -0.567 | -0.348 | 0.330 | | | |
| | p value | 0.025 | 0.115 | 0.282 | 0.303 | | | |
| **Model 3** | Predictors | Depth | Scraper Biomass | **Wastewater Pollution** | **Peak Rainfall** | 104.73 | 1.45 | 0.21 |
| | Coefficients | -0.508 | 0.540 | -0.802 | 0.728 | | | |
| | p value | 0.109 | 0.088 | 0.022 | 0.038 | | | |
| **Model 4** | Predictors | **Scraper Biomass** | **Urban Runoff** | Sediment Input | Peak Rainfall | 105.19 | 1.91 | 0.21 |
| | Coefficients | 0.673 | -0.655 | -0.076 | 0.493 | | | |
| | p value | 0.033 | 0.039 | 0.813 | 0.162 | | | |

AICc, Akaike's information criterion corrected for small sample size; ΔAICc, change in AICc across the candidate models (ΔAICc ≤ 2 of the top model are shown); McFadden's pseudo-$R^2$, proportion of variation explained by candidate model (0.2–0.4 represent an excellent fit)[98]. Significant predictors at p < 0.05 are in bold.

# Reporting Summary

## Statistics

For all statistical analyses, confirm that the following items are present in the figure legend, table legend, main text, or Methods section.

| n/a | Confirmed | |
|---|---|---|
| ☐ | ☒ | The exact sample size (*n*) for each experimental group/condition, given as a discrete number and unit of measurement |
| ☐ | ☒ | A statement on whether measurements were taken from distinct samples or whether the same sample was measured repeatedly |
| ☐ | ☒ | The statistical test(s) used AND whether they are one- or two-sided<br>*Only common tests should be described solely by name; describe more complex techniques in the Methods section.* |
| ☐ | ☒ | A description of all covariates tested |
| ☐ | ☒ | A description of any assumptions or corrections, such as tests of normality and adjustment for multiple comparisons |
| ☐ | ☒ | A full description of the statistical parameters including central tendency (e.g. means) or other basic estimates (e.g. regression coefficient) AND variation (e.g. standard deviation) or associated estimates of uncertainty (e.g. confidence intervals) |
| ☐ | ☒ | For null hypothesis testing, the test statistic (e.g. *F*, *t*, *r*) with confidence intervals, effect sizes, degrees of freedom and *P* value noted<br>*Give P values as exact values whenever suitable.* |
| ☒ | ☐ | For Bayesian analysis, information on the choice of priors and Markov chain Monte Carlo settings |
| ☐ | ☒ | For hierarchical and complex designs, identification of the appropriate level for tests and full reporting of outcomes |
| ☐ | ☒ | Estimates of effect sizes (e.g. Cohen's *d*, Pearson's *r*), indicating how they were calculated |

*Our web collection on statistics for biologists contains articles on many of the points above.*

## Software and code

Policy information about availability of computer code

| | |
|---|---|
| Data collection | All data and code that support the findings of this study are available at https://github.com/jamisongove/Coral-Reef-Persistence. |
| Data analysis | Statistical analyses were performed using the software packages R (www.r-project.org) version 4.0.2 (using libraries gamm4, MuMIn, foreach, doMC, ggplot2, gmt, tidyverse, zoo, lubridate) (ref. 1), Matlab (www.matthworks.com) using v2021a (using Statistics and Machine Learning toolbox), ArcGIS Desktop (www.esri.com) v10.6 with Advanced licensing and extensions Spatial Analyst and Geostatistical Analyst, InVEST Sediment Delivery Ratio model (https://naturalcapitalproject.stanford.edu/software/invest), and the PERMANOVA+ (ref. 2) add-on for Primer version 7 (ref. 3).<br><br>1 Team, R. C. in R Roundation for Statistical Computing (2021).<br>2 Anderson, M., Gorley, R. N. & Clarke, R. K. Permanova+ for primer: Guide to software and statistical methods (2008).<br>3 Clarke, K. & Gorley, R. Getting started with PRIMER v7. PRIMER-E: Plymouth, Plymouth Marine Laboratory (2015). |

For manuscripts utilizing custom algorithms or software that are central to the research but not yet described in published literature, software must be made available to editors and reviewers. We strongly encourage code deposition in a community repository (e.g. GitHub). See the Nature Portfolio guidelines for submitting code & software for further information.

## Data

Policy information about availability of data

All manuscripts must include a data availability statement. This statement should provide the following information, where applicable:

- Accession codes, unique identifiers, or web links for publicly available datasets
- A description of any restrictions on data availability
- For clinical datasets or third party data, please ensure that the statement adheres to our policy

---

All data that support the findings of this study are available at https://github.com/jamisongove/Coral-Reef-Persistence. Reef fish length-weight parameters were obtained from FishBase (https://fishbase.org) and ref. 1, human population data from NASA Gridded Population of the World v4 (https://sedac.ciesin.columbia.edu/data/set/gpw-v4-population-count-rev11), land use and land cover data from the NOAA Coastal Change Analysis Program (https://www.coast.noaa.gov/htdata/raster1/landcover/bulkdownload/), soils data from USDA Gridded Soil Survey Geographic Database (gSSURGO; https://www.nrcs.usda.gov/resources/data-and-reports/gridded-soil-survey-geographic-gssurgo-database), sub-watershed catchment data from USGS Stream Stats (https://water.usgs.gov/GIS/metadata/usgswrd/XML/ds680_archydrohucs.xml) (ref. 2), watershed and digital elevation model data from USGS National Hydrography Dataset (https://www.usgs.gov/national-hydrography/national-hydrography-dataset), rainfall data from refs. 3,4, Landsat 8 satellite image from USGS (https://earthexplorer.usgs.gov/), Landsat 7 and 8 cloud-free composites derived using Google Earth Engine (https://earthengine.google.com/), individual wastewater systems for Hawai'i from refs. 5,6, marine managed area designation from ref. 80 and downloadable from the State of Hawai'i (https://planning.hawaii.gov/gis), fishing regulations from the State of Hawai'i (https://dlnr.hawaii.gov/dar/fishing/fishing-regulations/), sea surface temperature and degree heating week data from NOAA Coral Reef Watch (https://coralreefwatch.noaa.gov/product/5km), ocean color (chlorophyll-a and irradiance) data from NOAA Coral Reef Watch (https://coralreefwatch.noaa.gov/product/oc/index.php) and ref. 7. See Methods and Supplemental Information for more detailed information on the data used to support the findings of this study.

1 Donovan, M. K. et al. Combining fish and benthic communities into multiple regimes reveals complex reef dynamics. Sci. Rep. 8, 16943 (2018).
2 Rea, A. & Skinner, K. D. Geospatial datasets for watershed delineation and characterization used in the Hawai'i StreamStats web application. US Geol. Surv. Data Ser. 680, 12 (2012).
3 Longman, R. J., Newman, A. J., Giambelluca, T. W. & Lucas, M. Characterizing the Uncertainty and Assessing the Value of Gap-Filled Daily Rainfall Data in Hawaii. J. Appl. Met. Clim. 59, 1261-1276 (2020).
4 Longman, R. J. et al. Compilation of climate data from heterogeneous networks across the Hawaiian Islands. Scientific Data 5, 180012 (2018).
5 DOH. Individual Wastewater System Database. Hawaii Dept. of Health (2017).
6 DOH. Underground Injection Control Permit application files. Hawaii Dept. of Health (2017).
7 Wedding, L. M. et al. Advancing the integration of spatial data to map human and natural drivers on coral reefs. PLoS ONE 13 (2018).

---

## Human research participants

Policy information about studies involving human research participants and Sex and Gender in Research.

| | |
|---|---|
| Reporting on sex and gender | NOT APPLICABLE |
| Population characteristics | NOT APPLICABLE |
| Recruitment | NOT APPLICABLE |
| Ethics oversight | NOT APPLICABLE |

Note that full information on the approval of the study protocol must also be provided in the manuscript.

# Field-specific reporting

Please select the one below that is the best fit for your research. If you are not sure, read the appropriate sections before making your selection.

☐ Life sciences       ☐ Behavioural & social sciences       ☒ Ecological, evolutionary & environmental sciences

For a reference copy of the document with all sections, see nature.com/documents/nr-reporting-summary-flat.pdf

# Ecological, evolutionary & environmental sciences study design

All studies must disclose on these points even when the disclosure is negative.

| | |
|---|---|
| Study description | The study tested the hypothesis that mitigating local human impacts facilitates coral reef persistence in the face of climate change-induced disturbance, specifically mass coral bleaching. Our goal was to move beyond commonly used proxies of local human impacts and generate spatially resolved data on specific land-sea human activities to identify actionable outcomes. This further allowed us to quantify the effects mitigating either land- or sea-based human impacts in isolation or simultaneously had on the ability of key reef-building organisms to recover post-disturbance. We achieved this by combining recurring in-water SCUBA surveys of coral reef benthic and fish communities with a 20-year time series of land-sea human impacts and other environmental factors thought to drive |

coral reef ecosystem processes. Our study included reefs across a broad range of ecological states, large spatiotemporal gradients in land-sea human impacts and environmental factors, and which experienced the most severe marine heatwave on record in the Hawaiian Islands.

**Research sample**

We quantified changes in the per cent cover of major reef-building benthic groups (hard coral, crustose coralline algae) and related these to concurrent changes in numerous land-sea human impacts, including urban runoff, wastewater pollution, nutrient loading, sediment input, and local restrictions on fishing gear types. Environmental factors included peak and annual rainfall, wave exposure, variability in ocean temperatures and heat stress, irradiance, and phytoplankton biomass. We also incorporated multiple fish biomass metrics that represent the critical role reef fish play in maintaining coral reef ecosystem dynamics. All human impacts and environmental factors were chosen based on prior evidence in the literature that they represent key drivers of reef ecosystem processes and were quantified using a variety of modelled and satellite-derived data sources.

**Sampling strategy**

Underwater visual surveys of shallow-water benthic and reef-fish assemblages were collated from the following three coral reef ecosystem monitoring agencies to maximise spatial and temporal replication across the study region: State of Hawai'i Division of Aquatic Resources, National Park Service, and The Nature Conservancy. Each program conducted surveys using similar data collection methods (see below) in shallow-water (<30 m) depths over hard-bottom substrate.

**Data collection**

All coral reef surveys used a traditional 25 m belt-transect method. Benthic surveys used permanently marked pins to ensure the same area of reef was surveyed over time. High resolution photographs were collected via photoquadrats at 1 m intervals along 25 m belt-transects (N = 26 photographs per transect). Thirty to fifty random points were overlaid on each photograph and the benthic component under each point was identified to the lowest possible taxonomic level. Per cent cover of the major functional groups were used in this analysis, namely hard coral, crustose coralline algae, macroalgae, and turf algae. All data were averaged among each transect and then among all transects for each site (1 – 4 transects per site, per year, depending on the monitoring program). Surveys of reef-fish assemblages were performed along the same permanently marked 25 m transects concurrently with benthic surveys. In all surveys, fishes were identified to species, sized, and enumerated. To account for differences among programs in how researchers surveyed reef fish, counts were calibrated using species and method specific adjustments previously developed for the region.

**Timing and spatial scale**

Underwater visual surveys of benthic assemblages were collated from three monitoring programs for the following years (number of reefs surveyed are in parentheses): 2003 (23), 2007 (23), 2011 (23), 2014 (40), 2015 (40), 2016 (80), 2017 (80), 2018 (15), 2019 (55). All benthic surveys used permanently marked pins to ensure the same area of reef was surveyed over time. High resolution photographs were collected via photoquadrats at 1 m intervals along 25 m belt-transects (N = 26 photographs per transect). Thirty to fifty random points were overlaid on each photograph and the benthic component under each point was identified to the lowest possible taxonomic level. Per cent cover of the major functional groups at each reef were used in this analysis, namely hard coral and crustose coralline algae. Surveys of reef-fish assemblages were performed along the same permanently marked 25 m transects concurrently with benthic surveys. However, reef fish surveys were performed more frequently (1 – 6 times per year from 2003 – 2019) than benthic surveys, depending on the reef location and monitoring program performing the surveys. In all surveys, fishes were identified to species, sized, and enumerated. To account for differences among programs in how researchers surveyed reef fish, counts were calibrated using species and method specific adjustments. The survey region spanned roughly 200 km of coastline on the island of Hawai'i.

**Data exclusions**

Fish species were excluded from fish biomass calculations according to life history characteristics that are not well captured with visual surveys, including cryptic benthic species, nocturnal species, pelagic schooling species, and manta rays. We also accounted for extreme observations of schooling species, which were defined by calculating the upper 99.9% of all individual observations, resulting in 26 observations out of over 0.5 million, comprised of 11 species. The distribution of individual counts in the entire database for those 11 species was then used to identify observations that fell above the 99.0% quantile of counts for each species individually. These observations were adjusted to the 99.0% quantile for analysis.

Other data exclusions include outliers in predictor variables (the local human impacts and environmental factors). Within the section "Coral reef trajectories pre-disturbance", prior to calculating per cent difference, we identified and removed outliers that fell outside a threshold of ± 2 standard deviations of the median. Within the section "Coral response to the 2015 Marine Heatwave", prior to model fitting, we identified the presence of outliers in our predictor variables as any point that fell outside a threshold of ± 2 standard deviations of the median. We then applied an additional step to retain any point above this threshold that was within 25% of the maximum predictor value below the threshold. This ensured that no data points were unnecessarily discarded from our formal model-fitting process because of applying an arbitrary threshold cutoff for data inclusion. We used the exact same process to identify and remove outliers within the section "Coral reefs four years post-disturbance" prior to formal model fitting.

**Reproducibility**

A description of the methodologies used is provided in the Methods and expanded on substantially for several of the human impact and environmental factors in the Supplementary Information. The data and full code necessary to reproduce the findings are available at https://github.com/jamisongove/Coral-Reef-Persistence

**Randomization**

Survey sites were either randomly or haphazardly chosen by the various monitoring agencies involved in data collection. Sites were separated by a minimum distance (250 m) and transects within sites were also separated by a minimum distance (5 - 10 m). To minimise observer bias of fish counts, sizing calibration dives were conducted using fish models of known size at the beginning of each field season. Observer crossover training was done using two observers side by side when possible. Benthic cover estimates were quantified by randomly assigning 20 points to each image using post-hoc image analysis programs (Photogrid or Coral Point Count with Excel Extensions) and identifying the benthic group to the lowest taxonomic rank under each point.

**Blinding**

All in situ benthic and reef fish surveys were conducted prior to this research question being conceived. The divers carried out the surveys for the most part without prior knowledge of the local human impacts and environmental factors for their respective survey locations – we later quantified these for each reef location and time of survey, thus blinding in this respect was achieved. In some cases, divers were aware of any local fishing restrictions in effect, but this was unavoidable as many of them specifically survey inside and outside of these zones

Did the study involve field work?  ☒ Yes  ☐ No

## Field work, collection and transport

| | |
|---|---|
| Field conditions | Because of the nature of collecting underwater benthic information, field conditions must be relatively calm (i.e., low wind and wave activity) with relatively good underwater visibility (i.e., > 5 m). |
| Location | Our study site was Hawaiʻi Island (19.55°N, 155.66°W), USA, which is the southeastern most island of the Hawaiian Archipelago, located in the northern central Pacific. The western section has roughly 200 km of coastline that is predominately oriented north to south. The coastline contains the longest contiguous reef ecosystem in the main Hawaiian Islands and large gradients in human population, local land-sea impacts, and environmental factors. The region represents an ideal study location for resolving the interacting land-sea human impacts driving reef ecosystem change and coral trajectories following acute climate-driven disturbance. All reefs included in this study were in shallow-water (depth < 30 m). |
| Access & import/export | All survey data were collected with the knowledge and consent of the State of Hawaiʻi, which has legal jurisdiction of all waters from 0 – 3 nm of the shoreline. The director of the State of Hawaiʻi's Division of Aquatic Resources, which is the managing agency of State waters, contributed survey data and both a collaborator and coauthor on this manuscript. |
| Disturbance | All surveys were performed by professional scientific divers that aim to minimise contact and disturbance of the reef. No coral reef benthic or fish species were removed from their habitat as part of this effort |

# Reporting for specific materials, systems and methods

We require information from authors about some types of materials, experimental systems and methods used in many studies. Here, indicate whether each material, system or method listed is relevant to your study. If you are not sure if a list item applies to your research, read the appropriate section before selecting a response.

## Materials & experimental systems

| n/a | Involved in the study |
|---|---|
| ☒ ☐ | Antibodies |
| ☒ ☐ | Eukaryotic cell lines |
| ☒ ☐ | Palaeontology and archaeology |
| ☒ ☐ | Animals and other organisms |
| ☒ ☐ | Clinical data |
| ☒ ☐ | Dual use research of concern |

## Methods

| n/a | Involved in the study |
|---|---|
| ☒ ☐ | ChIP-seq |
| ☒ ☐ | Flow cytometry |
| ☒ ☐ | MRI-based neuroimaging |

