## [Peer Review File · Nature]

Manuscript Title: Coral reefs benefit from reduced land-sea impacts under ocean warming

Reviewer Comments & Author Rebuttals

Reviewer Reports on the Initial Version:

Referees' comments:

Referee #1 (Remarks to the Author):

I think this is an important paper. Although very local in scale, the paper leverages a fantastic time series and very high resolution data, which are often missed in larger-scale studies. I think the paper may warrant publication in *Nature* for two reasons. First, it demonstrates the importance of integrated land and sea management. This has been a central feature of coral reef conservation since the late 1980s and early 1990s (e.g. Integrated Coastal Zone Management), but the proof of its efficacy has been wanting. Here, this paper delivers some much needed support for dismantling the conventional terrestrial and marine silos. Second, the paper highlights how local action can help reduce key impacts from climate change on coral reefs. It is a much needed message of hope in a field of study broadly dominated by doomism. However, there are some substantive modifications required before the paper can be published.

I liked the broad concept of using model outputs to develop scenarios of land-sea management, but the whole section lacked some clarity as to what they were doing and I am not convinced that moderate and low scenarios are actually that informative as they stand- you don't necessarily want to stay in low or medium, but rather move toward high. First, I think the whole section needs a sentence or two which simply explains that they used model results to generate scenarios of different land-sea management combinations and how these would influence the probability of being in a low, medium, or high reef builder category. Second, it is worth rethinking the output from this section. Instead of showing the probability of staying in categories that are not widely desirable, you really want to show how to move from less desirable to more desirable. One option would be to have 3 alluvial plots (one for each scenario; 1) increasing scrapers, 2) reducing wastewater, 3) both), akin to Fig. 3 in Cinner et al. 2020 Meeting fisheries, ecosystem function, and biodiversity goals in a human-dominated world. Science. In each alluvial plot, you could show how the proportion of reefs in the high, medium, low reef building categories change given that action. This may require some bootstrapping or running the analysis in a Bayesian framework and using the posteriors, but I think it would really make the message much, much clearer.

Relatedly, I am a little suspect about the decision to use 250kg/ha of scraper biomass in the scenarios. I understand that they have to pick a target and it will necessarily be somewhat arbitrary, but quite frankly, it seems like a lot of scrapers. It is unclear where this sits on the distribution of scraper biomass from this sample, but I wonder whether that is an unrealistic goal and one which present management is capable of achieving. The 2008 CRED data from Hawai'i (Big Island) that we have been using in some publications had just over 500kg/ha of all fish biomass as the average, and as the authors note, scrapers are a bit rare. I would like to have seen a distribution of scraper biomass (see below, I'd actually like to see it for all predictor variables, but especially this one), and some reassurance that the scenarios are realistic (i.e. that increasing scraper biomass to 250kg/ha is possible). To me, the most sensible approach here would be to bound the increase in scraper biomass used in the scenario by the effects of management (given the section on combined land and sea management). There is clear evidence from other studies (Russ, Babcock, McClanahan, etc.) using time series data that there will be more fish if you stop

killing fish, but I think the authors need to demonstrate for this dataset whether management actually does improve scraper biomass and can do so to the levels utilised in the scenario analysis (or change the levels used in the scenario analysis). There is no point in having a scenario which requires a bump of 235kg/ha, when the maximum management can provide is only 50 or whatever. Given the mosaic of protection types, there is a great opportunity to demonstrate which types of protection provide enough of a boon to fishes to aid in coral recovery. One option would be to model fish biomass given the range of predictor variables, then present the marginal effects of management (i.e. removing the effects of all of the other covariates- sampling, environment, etc.). Doing so might help provide a realistic bound for increasing fish biomass, rather than the seemingly arbitrary and potentially unrealistic 250kg/ha, or maybe it will justify that.

Given that this is a very local study, I think it is worth putting the degree heating weeks into a global context. Can you show the distribution of degree heating weeks for all reefs either globally or regionally (i.e. the Pacific) during that global bleaching event and show where the ~12 DHW from your study sits. I'd also like to see this study discussed in the context of the recent study by McWhorter et al. <https://doi.org/10.1111/gcb.16323> which suggests that refugia only exist in when global warming remains below 3C. I think this is important because while the present study clearly shows that local actions can help, but perhaps the capacity for local action becomes swamped above a certain level of heating.

The paragraph encompassing lines 82-89 is unclear. At present it essentially is a clumsy description of what is actually a clear key concept in human geography known as proximate and distal drivers. This paragraph should be re-written to integrate the social science literature on this concept, which would provide greater conceptual clarity and better explain what the actual issue is. See for example:

Geist, H. J., & Lambin, E. F. (2002). Proximate Causes and Underlying Driving Forces of Tropical Deforestation Tropical forests are disappearing as the result of many pressures, both local and regional, acting in various combinations in different geographical locations. *BioScience*, 52(2), 143-150.

Lambin, E. et al. (2001). The causes of land-use and land-cover change: moving beyond the myths. *Global environmental change*, 11(4), 261-269.

Cinner, J. E., & Kittinger, J. N. (2015). 22 Linkages between social systems and coral reefs. In C. Mora (ed) *Ecology of Fishes on Coral Reefs*, 215.

Hughes, T. et al. (2017). Coral reefs in the Anthropocene. *Nature*, 546(7656), 82-90.

Minor points

The paper needs a details oriented person to go over it. The last author's address seems incorrect. The main text refers to supplemental items that are mixed up. For example "and exceeded degree heating weeks 161 (Fig. S2), ..." This should be Fig S1.

Ln 136 "Reefs with positive trajectories had 93% greater human population density" not clear how this was measured- can you specify (e.g. .. greater human population density within the surrounding xx km²)

".. to depth, fishing gear, sediment input, annual rainfall," do you mean discarded fishing gear? Fishing gear allowed? Be clear

Reading comprehension would be improved if the date ranges for the different analyses were

included. Some span the full 20 years of available data, but the analyses with fish can't go beyond the 18 years which fish were sampled, thus there is some confusion about what was done.

I think that for each predictor, it would be helpful to have 3 supplemental plots: 1) a histogram of initial conditions (i.e. plot the value in each location/pixel 2000, 2003), 2) the delta over the time series (i.e. 2001-2019, 2003-2019), and variability (i.e. how much does each site/pixel change). This would also be helpful for coral cover and would help make some of the decision-making behind fig 2a,b more transparent.

One unanswered question that links the two parts of the study is how the places that were on a positive trajectory BEFORE the bleaching fared, likewise with those on a negative trajectory. Formally including this in the model might not be possible, given that the response is associated with many of the predictors and there might be collinearity there, but could you explore that qualitatively (i.e. even using red and blue "carpet" under the delta coral cover histogram).

One of the unresolved issues with this paper is that the authors do not have a compelling metric of fishing pressure. They use legally allowable gear, but compliance is the elephant in the room. I understand that they can't go back in time and collect objective metrics of fishing effort or compliance (e.g. discarded fishing gears), but this really does need to be discussed, especially since they authors make a big deal about the use of proxies, instead of appropriate spatially resolved data.

Referee #2 (Remarks to the Author):

The authors describe an unusually diverse set of coral reef trajectories that span a pre and post heatwave. They assign potential drivers to the fate of corals including both biotic and abiotic variables and make a case for local interventions aiding both the resistance and recovery from environmental stress.

This is a nice paper but I have mixed feelings on the way it's presented. On the positive, it's a great dataset and I'd like to see that published. On the negative, I feel the pitching of the paper - addressing a limitation of empirical data on benefits of local interventions - is not compelling. I also have some questions about the data analysis and generality of the findings. However, I believe these issues can be dealt with by reframing and clarifying several aspects of the analysis.

1) Framing:

I don't find the argument that there's a paucity of empirical evidence supporting the benefits of local interventions compelling. The authors' cite four papers that question the value of such interventions in a climate change context yet 3/4 papers are by John Bruno. Bruno is well known for views but he generally represents an extreme and certainly cannot be considered representative of the field. Part of the problem is that his arguments are either irrelevant or unsupported by data. Now I know that this process is not intended to be a review of Bruno (!) but let me point to a couple of examples. The data-rich paper cited (Baumann et al 2021) firstly tests for whether remoteness of sites (i.e., little human impact) reduces the impact of bleaching and major disturbances. They find no evidence of this but that's hardly surprising. Other than some work on corallivory by snails in the Caribbean I don't know of any concrete theories as to why remoteness should mitigate the impacts of heatwaves or cyclones etc. It's essentially a false straw man. No one advocates MPAs as a means of reducing climate stress, for example. So that finding tells us little about the value of managing human impacts on resistance to bleaching. They then look to see whether coral recovery is faster in remote areas and once again find no significant effect. Yet, recent studies have shown that such analyses tend to have extraordinarily low power, quite easily overlooking an improvement of 10-20% in coral cover (Mumby et al 2021). Thus, their conclusions aren't surprising and a better study would quantify the sensitivity of their approach to

detecting any trends.

I make these points merely to highlight that the apparent controversy this paper addresses is weaker than implied. Moreover, there is a wealth of studies that have documented impacts of improved water quality or fish biomass on processes of reef recovery / recovery potential - see the papers by SV Smith (Hawaii eutrophication reversal), Bob Richmond, Katharina Fabricius, Tom Tomascik, Mark Hay, Pete Mumby, Bob Steneck, Nick Graham, Donovan (also Hawaii).

To me, a more compelling framing concerns the difficulty in relating patterns to the specific drivers and you have an excellent opportunity within a contained reef system to do this.

2) I have several questions on the analysis but the most significant is that it wasn't clear how the sites varied in their exposure to thermal stress. Can the authors' reliably show that the coral outcome (either mortality from bleaching or recovery rate) occurred despite differences in exposure to thermal stress among sites? This needs to be highly transparent. A related issue concerns the conclusions regarding how different combinations of water quality and fish biomass would impact future responses to heatwaves. These implicitly assume that the distribution of heatstress among sites remains the same for each putative bleaching event. But what would happen if the areas with relatively good biological conditions (say) were exposed to the highest thermal stress?

3) Many of the variable seem to be strongly correlated (fig 1 and 3). How was this dealt with explicitly? Note, for example, that the pattern of less coral mortality with greater depth is well known (as stated by authors) but so too is the negative association between depth and herbivorous fish biomass. So the apparent correlation between bleaching mortality and herbivorous fish biomass could be spurious and driven by depth. These are the sorts of issues that need to be made more compelling.

4) Lastly, why were the reef states discretised into high, med, low rather than use analyses based on continuous scales? That usually loses data and wouldn't be ideal.

Referee #3 (Remarks to the Author):

This is an impressive study. The importance of local drivers in mitigating the impacts of climate disturbance on coral reefs is an important research front, which as the authors point out has been addressed with broad proxy data in the past. The level of detail in quantifying specific local drivers of coral reef trajectories and responses is really unique and makes this a powerful analysis of how anthropogenic, environmental and ecological factors influence coral reef persistence to global climate disturbance. The manuscript is very well written and reasoned, the figures are powerful, and the inferences important. The detail in the drivers allows for some really tangible management actions and outcomes.

There are various bits of information that are missing from the main text, which would help the reader, including some that raised red flags until I got to the methods or supplemental material. These things should be brought into the main text, which can be done with few words, leaving the details for the methods. These include:

- An obvious question is how correlated the various drivers are, and thus how independent they are. This is not clear from the main text or figures. There are nice figures assessing this collinearity in the supplement (Figs S2 and S4) and the legend explains that if correlations are >0.7 , one of the variables was excluded. This is appropriate, and should be explained briefly through the main text.

- Frequency, number of sites, and replication for underwater surveys. In fact this information is not even in the methods, and can only be found in the supplemental material. Replication will be a

key factor that readers wonder about. It looks strong from the supplemental material, but the reader shouldn't need to go that far to be sure of that.

- How were the reef trajectories (fig 2b) defined as positive, negative, or neutral. This information is in the methods, but half a sentence could explain that in the main text also.

The change in benthic cover through the heatwave (Fig 3b) seems to show that reefs with higher starting coral cover tended to lose the most coral cover. This begs the question of what the relationship was with coral composition for these changes through the heatwave. Were the sites with most loss of coral cover, also the ones with the highest cover of vulnerable coral taxa? How does this fit with your analyses / drivers.

Some of your drivers of change through the heat stress (Fig 3c) have very weak slopes (total fish biomass, grazer biomass, and to a less extent sediment load). While I see these factors were strongly supported by AICc model weights, the slopes are within the 80% CI's, so you should temper your language somewhat in the text regarding these drivers.

For the post-disturbance analysis, you bring in CCA with hard coral cover. I see the rationale for this, but it may be useful to have a supplemental figure /analysis that runs the same analysis (proportion low, medium and high, and how changes before versus after) for just hard coral cover, to tie the narrative through from the earlier figures.

The conclusion section is brief. That in itself is fine, however given you have different specific drivers as important in the pre (trajectories), during heat stress, and post disturbance recovery sections of the paper, this really needs to be unpacked a bit here. You need to consolidate this information and explain what it means in terms of management. For example, you have some nice tangible recommendations from the analysis in figure 4, but wastewater was not important in Fig 3, and urban runoff was important in figs 2 and 3 (among other variables). Some thought and recommendations across the disturbance periods and drivers would be useful here, perhaps in the context of ongoing anthropogenic drivers and heat extremes.

Related to the above point, while the 30 by 30 is a good hook, I wonder if protected areas are really what your findings point to. Perhaps in part, but the actions needed based on your detailed study can be more specific and thus useful for policy makers I think – such as waste water management, fisheries governance etc.

Other minor points:

You have some redundancy between the opening paragraph of the main text and the start of the abstract.

Another point to perhaps allude at the start is that in some well studied systems, land based and marine based drivers are not both strong. For example fishing pressure on the GBR is very light and doesn't target the herbivores. A strength of your study system is being able to disentangle these things in some detail, which isn't possible everywhere.

Line 137, it is not clear what you mean by 'historically less exposed' here. If you have used a lag function in assessing the drivers, that needs to be clear.

Referee #4 (Remarks to the Author):

A. Summary of key results: the authors show that benthic community dynamics on reefs located on the western coast of Hawai'i correlate with a variety of land- and sea-based variables that are indicative of local conditions and, for the most part, 'manageable.' First, they show that general

trends in coral cover (used throughout as the indicator of reef condition) prior to the mass bleaching event show distinct trends with regards to a range of variables, including herbivorous fish biomass, wave exposure, human population density, or wastewater pollution. They then highlight that a similar suite of variables appears to have mediated the condition of reefs after a severe bleaching event. Finally, they hone in on the most important drivers of reef condition in an ordinal framework, demonstrating that scraping herbivores and wastewater pollution combined appear to have the greatest effect on reef recovery. The authors use their results to suggest that local human management efforts can help ameliorate the effects of temperature-mediated bleaching on coral reefs.

B. Originality and significance: the quality of this paper is outstanding. It is well written, thoughtfully analyzed, beautifully visualized, and generally very well done. The consideration of so many high-resolution variables in a single framework is rather novel compared to existing work and produces some really interesting results that I believe are very valuable. However, with regards to overall originality and significance, I am not convinced that the advances made in the paper truly meet the expectations of the journal. I base this assessment on three elements, which include 1) the framing of the paper as a counterpoint to recent work claiming that local management does not help corals against bleaching, 2) the limited spatial scale and inherent context-dependency of the results, and 3) the fact that the major implications (managing fish populations and terrestrial runoff benefits coral reefs) are fairly well-established.

1) The paper is framed as a counterpoint to a set of papers that claim that local conditions do not significantly modify the resilience of reefs to coral bleaching (e.g. Hughes et al. 2017; Bruno et al. 2019; Bruno & Valdivia 2016). I have several issues with this and would argue that it's a bit of a strawman. For example, Bruno et al. 2019 specifically argue that they do not find an effect of MPAs (read: protection of herbivorous fishes) for coral reef *resilience* to stressors, but this is different from the recovery dynamics that are described in this paper. Similarly, Hughes et al. 2017 do argue that local management does not afford any protection from bleaching per se, but they notably do not say anything about recovery. Thus, both of these papers do not actually pose a counterpoint to the results of this paper as currently framed, and while I do think that some of the verbiage in these papers is unnecessarily unbalanced, I strongly believe that the vast majority of reef scientists and practitioners can see beyond this artificial dichotomy of local vs. global management of reefs. This means, in turn, that I think we ought to de-escalate this discussion rather than highlight it (as done in l. 74-80), but that is perhaps more of a personal preference. In any case, I would argue that there aren't a lot of scientists in this world who would refute the hypothesis that efficient management of local stressors benefits coral reefs in the face of global warming.

2) One of the strengths of the paper is the precise identification of two main drivers that have facilitated coral recover in the aftermath of the bleaching event—scraping parrotfishes and wastewater runoff. This is a great source of information for local management and provides tangible targets, but it does so (at this stage) almost exclusively for this particular island and possibly other Hawai'ian islands. Hawai'ian reefs are pretty unique in their geology, geographic positioning, climate, reef communities, and anthropogenic stressors and management, so at this point, I think we have to interpret these results as a very cool and useful phenomenological suite of findings that apply to Hawai'i, rather than extrapolating to coral reefs in general. In other words, an atoll in the Indian Ocean or Coral Sea is likely to benefit very little from human wastewater management or protection of herbivorous fishes (at least for the benthic community; Graham et al. 2020), but could probably benefit from other management actions (e.g., rat eradication, Graham et al. 2018; reef restoration, Lamont et al. 2022). Of course, the broader derivation of the results (that local management is good for reefs) is much more applicable to reefs worldwide, but this is also very well known and broadly documented (see 3) below).

3) Overall, I think there is overwhelming evidence that local management matters. In fact, especially with regards to fisheries management and pollution (the land- and sea-elements of the present paper), one of the co-authors published a paper not too long ago that made a very convincing case that globally, management of fish populations and pollution can benefit reefs in the face of bleaching events (Donovan et al. 2021). Beyond this recent paper, there is a plethora of work that shows these trends in isolation at more local scales of reef recovery after bleaching

(e.g., Mumby et al. 2021; Steneck et al. 2019 for fishing effects; Mellin et al. 2019; MacNeil et al. 2019 for water quality effects). Of course, this is related to the beneficial effects of restricting human impacts on ecosystems as a whole, which have been well established for coral reefs for several decades.

In saying this, I do not mean to diminish the presented results in any way. As stated initially, I think that this is an excellent paper that is worthy of being published in a good journal and that deserves attention. However, given that the instructions are to evaluate the novelty and impact of the paper, I felt compelled to provide my evaluation in light of the existing literature and prevailing knowledge base.

C. Data and methodology: the dataset is an extensive time series that covers a strong disturbance event and is thus well suited for gauging reef recovery, albeit at a limited spatial scale. The covariates the authors use are of higher resolution than commonly seen and permit a detailed assessment. As stated in the beginning, the overall quality of the manuscript is very high.

D. Use of statistics and treatment of uncertainties: overall, the analytical frameworks the authors used are well-suited for the task and quite elegant. However, I have some issues with the interpretation of the GAMMs, which the authors state show effects of various covariates (e.g. depth, herbivore biomass, fish biomass, sedimentation) on coral bleaching recovery (l. 178-215). Looking at Fig. 3 (which I understand shows model-averaged partial effects), I see reasonable effects of depth (an unequivocally un-manageable covariate) and urban runoff on the recovery dynamics of coral cover, but the other effects appear to be basically negligible. For example, total fish biomass over its entire range appears to shift coral recovery dynamics between approximately -12 and -10% with substantial uncertainty associated with these estimates. Similarly, sediment input appears to move coral recovery from -11% to perhaps -15% over its entire range, again showing substantial uncertainty. Are these really meaningful ecological effects? I would advise to be cautious with this, since GAMMs can be a bit quick to produce 'significant' effects and high R² values, especially in models with lots of parameters. I am rather skeptical as to whether there's a meaningful relationship in several of these covariates. On that note, it would also be great to see the raw data superimposed on the regression lines to get a better idea of model fit, which is impossible to see from the AIC values. I had a similar issue with the first analysis (the quantification of the pre-disturbance dynamics), which is essentially based on a categorical descriptor based on >3% change in coral cover over 10 years. While the authors do provide a rationale for that value based on the range of observed coral cover values (l. 500), I was rather surprised by the choice of such a small value. To me, a 3% change in coral cover does not symbolize a directional shift that merits a categorical assignment of positive or negative, so I would at least expect some kind of sensitivity analysis for this section that can support the categorical assignment with some larger values of change. With that being said, the OLR analysis is very powerful and elegant and really smart way of analyzing and visualizing the effects of covariates on reef builder cover. My only question here would be why the authors decided to expand this analysis to all reef-building organisms, rather than restricting it to live corals only as in the previous analysis. There wasn't really a convincing rationale in the paper for this decision, as far as I could tell.

E. Conclusions: as highlighted in my comments above, I think that some of the conclusions aren't as well supported by the presented effects sizes as the authors claim. This definitely merits reconsideration and a thoughtful ecological interpretation.

F. Suggested improvements: overall, I would suggest to step away from the artificial dichotomy between local vs. global stressors and management of coral reefs with the framing of the paper. Other than that, the few analytical/interpretational comments should provide some food for thought.

G. References: the paper appropriately references previous work and I am completely aware of the space restrictions that come with submission to Nature.

H. Clarity and context: as mentioned previously, the paper is well written but could be framed differently.

Author Rebuttals to Initial Comments:

Referee #1 (Remarks to the Author):

R1.1

I think this is an important paper. Although very local in scale, the paper leverages a fantastic time series and very high resolution data, which are often missed in larger-scale studies. I think the paper may warrant publication in Nature for two reasons. First, it demonstrates the importance of integrated land and sea management. This has been a central feature of coral reef conservation since the late 1980s and early 1990s (e.g. Integrated Coastal Zone Management), but the proof of its efficacy has been wanting. Here, this paper delivers some much needed support for dismantling the conventional terrestrial and marine silos. Second, the paper highlights how local action can help reduce key impacts from climate change on coral reefs. It is a much needed message of hope in a field of study broadly dominated by doomism. However, there are some substantive modifications required before the paper can be published.

We thank the reviewer for their positive comments about our paper.

R1.2

I liked the broad concept of using model outputs to develop scenarios of land-sea management, but the whole section lacked some clarity as to what they were doing and I am not convinced that moderate and low scenarios are actually that informative as they stand- you don't necessarily want to stay in low or medium, but rather move toward high. First, I think the whole section needs a sentence or two which simply explains that they used model results to generate scenarios of different land-sea management combinations and how these would influence the probability of being in a low, medium, or high reef builder category.

Thank you for bringing this to our attention. We have now re-structured this section, including adding language that better introduces and contextualises the results to ensure our goal and intention is more clearly explained upfront as you suggest.

Second, it is worth rethinking the output from this section. Instead of showing the probability of staying in categories that are not widely desirable, you really want to show how to move from less desirable to more desirable. One option would be to have 3 alluvial plots (one for each scenario; 1) increasing scrapers, 2) reducing wastewater, 3) both), akin to Fig. 3 in Cinner et al. 2020 Meeting fisheries, ecosystem function, and biodiversity goals in a human-dominated world. Science. In each alluvial plot, you could show how the proportion of reefs in the high, medium, low reef building categories change given that action. This may require some bootstrapping or running the analysis in a Bayesian framework and using the posteriors, but I think it would really make the message much, much clearer.

This was a fantastic suggestion by the reviewer. We completely agree that specifying the management goal of moving from *Low* towards *High* reef-builder cover is far more appropriate. Taking inspiration from the alluvial plot concept in Cinner et al. (2020) and the probability bi-plot in Graham et al. (2015), we re-extracted the probabilities from our model to create a single continuous probability surface showing the conditions under which one can move from a less desirable (*Low* reef-builder cover) to more desirable state (*Moderate* or *High* reef-builder cover). As such, we have removed the original 3-panels in Fig. 4 that showed the probability of remaining in a *Low*, *Moderate*, or *High* reef-builder cover category. Instead, these are now condensed into a single panel that shows the probability of moving between the three categories depending on the levels of wastewater pollution and scraper biomass. We then overlay our 'resource management scenarios' that demonstrate transitioning from *Low* (less desirable) to *High* (more desirable) reef-builder category requires simultaneously increasing scraper biomass and reducing mitigating wastewater pollution. We think the updated figure and edits to the manuscript provide increased clarity and far more impactful results. Again, we thank the reviewer for suggesting this improvement.

Cinner, J. E. *et al.* Meeting fisheries, ecosystem function, and biodiversity goals in a human-dominated world. *Science* **368**, 307-311 (2020).

Graham, N. A. J., Jennings, S., MacNeil, M. A., Mouillot, D. & Wilson, S. K. Predicting climate-driven regime shifts versus rebound potential in coral reefs. *Nature* **518**, 94-97 (2015)

R1.3

Relatedly, I am a little suspect about the decision to use 250kg/ha of scraper biomass in the scenarios. I understand that they have to pick a target and it will necessarily be somewhat arbitrary, but quite frankly, it seems like a lot of scrapers. It is unclear where this sits on the distribution of scraper biomass from this sample, but I wonder whether that is an unrealistic goal and one which present management is capable of achieving. The 2008 CRED data from Hawai'i (Big Island) that we have been using in some publications had just over 500kg/ha of all fish biomass as the average, and as the authors note, scrapers are a bit rare. I would like to have seen a distribution of scraper biomass (see below, I'd actually like to see it for all predictor variables, but especially this one), and some reassurance that the scenarios are realistic (i.e. that increasing scraper biomass to 250kg/ha is possible). To me, the most sensible approach here would be to bound the increase in scraper biomass used in the scenario by the effects of management (given the section on combined land and sea management). There is clear evidence from other studies (Russ, Babcock, McClanahan, etc.) using time series data that there will be more fish if you stop killing fish, but I think the authors need to demonstrate for this dataset whether management actually does improve scraper biomass and can do so to the levels utilised in the scenario analysis (or change the levels used in the scenario analysis). There is no point in having a scenario which requires a bump of 235kg/ha, when the maximum management can provide is only 50 or whatever. Given the mosaic of protection types, there is a great opportunity to demonstrate which types of protection provide enough of a boon to fishes to aid in coral recovery. One option would be to model fish biomass given the range of predictor variables, then present the marginal effects of management (i.e. removing the effects of all of the other covariates- sampling, environment, etc.). Doing so might help provide a realistic bound for increasing fish biomass, rather than the seemingly arbitrary and potentially unrealistic 250kg/ha, or maybe it will justify that.

The reviewer makes an excellent point with respect to providing justifications for the management scenarios presented in Fig. 4b. Previously, we provided no context for these values. With respect to scraper biomass, we choose 250 kg/ha as the management target given that it closely aligns with the biomass of scrapers within Kealakekua Bay, where no fishing has been allowed since 1969. Data from the permanent survey site within Kealakekua Bay indicate that the long-term mean (2003 – 2019; N = 17) in scraper biomass is 243 kg/ha. More recently (2016 – 2019; N = 4), mean scraper biomass is 302 kg/ha. Importantly though, Kealakekua Bay is exposed to numerous land-based stressors, including high levels of wastewater pollution (258,000 L/h in 2019). As such, our value of 250 kg/ha is aligned with the long-term mean to represent a more conservative estimate of scraper biomass on a reef in our study region with fisheries protection but with land-based stressors present. To provide further context, we compared our upper (250 kg/ha) and lower (30 kg/ha) scraper biomass values to the distribution of scraper biomass among all reefs (N = 80) in 2019, the most recent time point in which all reefs were surveyed within the same year (see below figure). The upper and lower biomass values represent the 88th and 36th percentile, respectively. We have now added a *Management Scenarios* subsection in the *Methods* of the main paper as well as in the Supplemental Information document that provides context and justification for the scraper biomass and wastewater pollution values chosen in our scenario analysis (i.e., Fig. 4b).

Figure. Distribution of scraper biomass in the most recent year (2019) in which all reefs ($N = 80$) were surveyed for reef fish. Scraper biomass bars are in 25 kg/ha intervals with the exception of the last bar that is the sum proportion of all reef surveys between 399 – 672 kg/ha, which is the maximum biomass value recorded. Inset figure represents the cumulative density estimate, where the height of each bar is equal to the cumulative relative number of observations in the bar and all previous bars. Scraper biomass from the monitoring site within Kealakekua Bay, a marine protected area established in 1969, is shown in red. The vertical dashed and solid red lines represent the long-term average (243 kg/ha; 2003 – 2019; $N = 17$) and recent average (302 kg/ha; 2016 – 2019; $N = 4$) scraper biomass. The horizontal dashed and solid red lines are the intersection of their respective biomass values along the y-axis, which is 0.92 and 0.93 for the long-term average and more recent average scraper biomass values.

R1.4

Given that this is a very local study, I think it is worth putting the degree heating weeks into a global context. Can you show the distribution of degree heating weeks for all reefs either globally or regionally (i.e. the Pacific) during that global bleaching event and show where the ~12 DHW from your study sites. I'd also like to see this study discussed in the context of the recent study by McWhorter et al.

<https://doi.org/10.1111/gcb.16323> which suggests that refugia only exist in when global warming remains below 3C. I think this is important because while the present study clearly shows that local actions can help, but perhaps the capacity for local action becomes swamped above a certain level of heating.

With regards to the first point, we agree we needed to provide more context for the thermal stress exposure experienced across our study region (a related comment was made in R2.3). We now plot the variation in DHW values across our study reefs (see Fig. 3b), highlighting how uniform and severe the thermal stress exposure was (mean = 12 DHW). We also place the 2015 marine heatwave into a longer historical context (Fig. S1), highlighting that this was the most severe marine heatwave in the region on record for the past 120 years (see new Fig. S1, also below).

With regards to the second point, we fully agree with the reviewer that we need to place our findings and conclusions in the context of the predicted future global increases in frequency and severity of marine heatwaves that trigger coral bleaching events (van Hoodonk et al. 2016, Dixon et al. 2022). In our *Conclusion* section we have added language that very frequent and severe mass bleaching predicted in the coming years to decades could overwhelm the positive effects of any local land-sea management efforts on reefs. McWhorter et al. (2022) is a very interesting paper, but is focused on the Great Barrier Reef. The two papers we have now included provide global predictions and are therefore perhaps more relevant for this particular point. Importantly though, we also make the following points in our *Conclusion* section:

- under reduced emissions scenarios there is substantial variation in the projected rates of ocean warming within and among countries and

- that supporting coral reef resilience to climate change locally alongside rapid reductions in global greenhouse gas emissions may buy reefs more time to adapt and persist into the future.

Dixon, A. M., Forster, P. M., Heron, S. F., Stoner, A. M. K. & Beger, M. Future loss of local-scale thermal refugia in coral reef ecosystems. *PLOS Climate* **1**, e0000004 (2022)

van Hooidonk, R. *et al.* Local-scale projections of coral reef futures and implications of the Paris Agreement. *Scientific Reports* **6**, 39666 (2016)

Figure. Long-term ocean temperature record averaged across the entire main Hawaiian Islands derived from monthly sea surface temperature (SST) from 1900 – 2020. Dashed lines represent +/- 2 standard deviations (SD) above the long-term mean. Red line is the 12-month moving average. The temperature in 2015 represents the most severe marine heatwave within this 120-year time series. Data is from NOAA’s Extended Reconstructed SST (<https://www.ncei.noaa.gov/products/land-based-station/noaa-global-temp>).

R1.5

The paragraph encompassing lines 82-89 is unclear. At present it essentially is a clumsy description of what is actually a clear key concept in human geography known as proximate and distal drivers. This paragraph should be re-written to integrate the social science literature on this concept, which would provide greater conceptual clarity and better explain what the actual issue is. See for example:

- Geist, H. J., & Lambin, E. F. (2002). Proximate Causes and Underlying Driving Forces of Tropical Deforestation. *Tropical forests are disappearing as the result of many pressures, both local and regional, acting in various combinations in different geographical locations. BioScience*, 52(2), 143-150.
- Lambin, E. *et al.* (2001). The causes of land-use and land-cover change: moving beyond the myths. *Global environmental change*, 11(4), 261-269.
- Cinner, J. E., & Kittinger, J. N. (2015). 22 Linkages between social systems and coral reefs. In C. Mora (ed) *Ecology of Fishes on Coral Reefs*, 215.
- Hughes, T. *et al.* (2017). Coral reefs in the Anthropocene. *Nature*, 546(7656), 82-90.

We appreciate the nudge from the reviewer here. We have now restructured our *Introduction* Paragraph 2 following comments from Reviewer 2 (R 2.2) and Reviewer 4 (R 4.3) about the overall framing of the knowledge gap. This restructure has meant a focus on direct (proximate) drivers in our *Introduction*, but we circle back to the importance of considering the interconnected complex nature of both proximate and

distal human drivers of reefs in our *Conclusion* and cite both Hughes et al. (2017) and Cinner & Kittinger (2015) as part of these discussions.

Cinner, J. E. & Kittinger, J. N. 22 Linkages between social systems and coral reefs. *Ecology of Fishes on Coral Reefs*, 215 (2015)

Hughes, T. P. *et al.* Coral reefs in the Anthropocene. *Nature* **546**, 82-90 (2017)

Minor points

R1.6

The paper needs a details oriented person to go over it. The last author's address seems incorrect. The main text refers to supplemental items that are mixed up. For example "and exceeded degree heating weeks 161 (Fig. S2), ..." This should be Fig S1.

We apologise for these oversights and have done a thorough review of the revised manuscript.

R1.7

Ln 136 "Reefs with positive trajectories had 93% greater human population density" not clear how this was measured- can you specify (e.g. .. greater human population density within the surrounding xx km²)"

We have now included greater specificity by adding that population density is the number of people within a 15 km radius of each reef in the main manuscript.

R1.8

".. to depth, fishing gear, sediment input, annual rainfall," do you mean discarded fishing gear? Fishing gear allowed? Be clear

Thank you for pointing out this lack of clarity. We are referring to the allowable fishing gear on a given reef. We have added this specific language to the manuscript.

R1.9

Reading comprehension would be improved if the date ranges for the different analyses were included. Some span the full 20 years of available data, but the analyses with fish can't go beyond the 18 years which fish were sampled, thus there is some confusion about what was done.

We agree this needed greater clarity. We have now clearly stated towards the end of the *Introduction* that we quantified drivers of coral reef benthic change at the scale of individual reefs over 12 years prior to disturbance (2003-2014), during and immediately following the marine heatwave (2014-2016), and four years post-disturbance (2016-2019). We have also now added these time windows to the legend on Fig. 1 (look for "Permanent Reef Survey Data Availability"), showing the specific date ranges for ecological data included in the three analytical phases (pre-disturbance, disturbance, post-disturbance). We think these additions help provide increased understanding of the differing time ranges between our reef survey and land-sea impact data presented in the manuscript.

R1.10

I think that for each predictor, it would be helpful to have 3 supplemental plots: 1) a histogram of initial conditions (i.e. plot the value in each location/pixel 2000, 2003), 2) the delta over the time series (i.e. 2001-2019, 2003-2019), and variability (i.e. how much does each site/pixel change). This would also be helpful for coral cover and would help make some of the decision-making behind fig 2a,b more transparent.

We thank the reviewer for this suggestion as these additional plots were definitely needed to provide more insight into the underlying distributions of our data. We have now included four supplemental figures that show the distribution for land-sea human impacts and environmental data across the various time ranges that are the focus of our paper. Fig. S3 (shown below) plots the distributions, change over time, and variability of each factor from 2000 – 2019. The distribution in mean values were calculated from the mean of the first five years (2000 – 2004) and the most recent five years (2015 – 2019). This accounted for year-to-year variability in the episodic nature of factors such as wave exposure, rainfall, and sediment input. Change over time ('delta') were the most recent five years minus the first five years. Variability was calculated as the standard deviation in annual data from 2000 – 2019. Distributions shown are based on a subset of data that were geographically constrained to within the northern and southern latitudinal extent of our reef surveys (Fig. 1b). We do not include metrics of fish biomass and phytoplankton biomass in this figure as these data are unavailable at the same temporal or spatial resolution.

The additional 3 supplemental figures plot the distributions of the land-sea human impacts and environmental factors that are shown in Fig. 2c (Fig. S4), and the predictor variables used in our coral response to the 2015 marine heatwave section (i.e., GAMM; Fig. S8) and coral reefs four years post-disturbance section (i.e., OLR; Fig. S11).

Figure. The distributions (left), change over time (middle), and variability (right) for land-sea human impacts and environmental factors. The distributions in mean values were calculated from the mean of the first five years (2000 – 2004) and the most recent five years (2015 – 2019). Change over time (‘delta’) was calculated based on the most recent five years minus the first five years. Variability was calculated as the standard deviation in annual data from 2000 – 2019. Data shown were geographically constrained to within the northern and southern latitudinal extent of our reef surveys (Fig. 1b). Metrics of fish biomass and phytoplankton biomass are not shown as these data are unavailable at the same temporal or spatial resolution. See Table S1 for summary information on local land-sea human impacts and environmental factors included in our analyses. See Supplementar Methods for detailed information on calculating each driver, including data collection methods, data sources and ancillary data sets, and specific tools or software utilised.

R1.11

One unanswered question that links the two parts of the study is how the places that were on a positive trajectory BEFORE the bleaching fared, likewise with those on a negative trajectory. Formally including this in the model might not be possible, given that the response is associated with many of the predictors and there might be collinearity there, but could you explore that qualitatively (i.e. even using red and blue “carpet” under the delta coral cover histogram).

This is an interesting point raised by the reviewer. They are correct that including this formally in our model fitting process is not possible, however we have now explored this in some detail. This becomes important with regards to a comment raised by Reviewer 3 about the possible effect variations in starting coral cover condition might have had on how reefs responded to thermal stress (see reviewer comment R3.3 below). In summary, reefs that were on a positive coral cover trajectory pre-disturbance (from 2003-2014) had higher mean coral cover just prior to the marine heatwave compared to those reefs that had been on a negative coral cover trajectory. As a consequence of the marine heatwave, positive trajectory reefs lost more coral cover (in absolute terms) than negative trajectory reefs, essentially because they had more to lose and further to fall (Côté et al. 2005). However, once variations in starting coral cover condition are accounted for (*sensu* Graham et al. 2008, *see our response to R3.3 below for details*), this relationship goes away, and positive and negative trajectory reefs are found to experience equivalent coral loss following the disturbance. We now include a supplemental figure to summarise this information (see Fig. S5, also below) and make reference to this in the *Methods (Statistical Analyses)* section of the main paper when we describe accounting for starting conditions in coral cover prior to modelling the drivers of coral loss following the marine heatwave.

Côté, I. M., Gill, J. A., Gardner, T. A. & Watkinson, A. R. Measuring coral reef decline through meta-analyses. *Philosophical Transactions of the Royal Society B: Biological Sciences* **360**, 385-395 (2005)

Graham, N. A. J. *et al.* Climate Warming, Marine Protected Areas and the Ocean-Scale Integrity of Coral Reef Ecosystems. *Plos One* **3**, e3039 (2008)

Figure. a, Coral cover on positive (blue) and negative (red) trajectory reefs surveyed (N=18), see Fig. 2b in main manuscript) prior to (2014) and 1-year following (2016) the marine heatwave. **b**, Positive trajectory reefs have a higher mean coral cover both prior to, and to a lesser extent, following the marine heatwave. **c**, Positive trajectory reefs experience increased absolute coral cover loss following the marine heatwave (underlying relationship shown in panel **d**), but this difference is removed once starting coral cover condition is accounted for (underlying relationship show in panel **e**).

R1.12

One of the unresolved issues with this paper is that the authors do not have a compelling metric of fishing pressure. They use legally allowable gear, but compliance is the elephant in the room. I understand that they can't go back in time and collect objective metrics of fishing effort or compliance (e.g. discarded fishing gears), but this really does need to be discussed, especially since they authors make a big deal about the use of proxies, instead of appropriate spatially resolved data.

This is a fair and important point raised by the reviewer. We use total fish biomass as an indicator of the overall state of the fish assemblage (McClanahan et al 2012), but also point out that fish biomass is reduced in areas that have increased fishing pressure (Cinner et al. 2016, 2018). In Hawai'i, non-commercial nearshore fisheries dominate, with people fishing for recreational, subsistence, and cultural purposes (Kittinger et al 2015; Grafeld et al 2017). However, the dominant harvesting modes and

magnitude of fishing activities are largely unknown at spatial or temporal scales relevant to this study (Delaney et al 2017). As such, we include total fish biomass in part to represent fishing effort on reefs but recognise its shortcomings in capturing reef- and species-specific differences in fishing pressure across our study region. We have now added language to the *Methods* section which highlights this more clearly.

McClanahan, T. R. et al. Prioritizing Key Resilience Indicators to Support Coral Reef Management in a Changing Climate. *Plos One* 7, e42884 (2012)

Cinner, J. E. et al. Bright spots among the world's coral reefs. *Nature* 535, 416-419 (2016).

Cinner, J. E. et al. Gravity of human impacts mediates coral reef conservation gains. *Proceedings of the National Academy of Sciences* 115, E6116-E6125 (2018).

Kittinger, J. N. et al. From Reef to Table: Social and Ecological Factors Affecting Coral Reef Fisheries, Artisanal Seafood Supply Chains, and Seafood Security. *Plos One* 10, e0123856 (2015).

Grafeld, S., Oleson, K. L. L., Teneva, L. & Kittinger, J. N. Follow that fish: Uncovering the hidden blue economy in coral reef fisheries. *Plos One* 12, e0182104 (2017).

Delaney, D. G. et al. Patterns in artisanal coral reef fisheries revealed through local monitoring efforts. *PeerJ* 5, e4089 (2017).

Referee #2 (Remarks to the Author):

R2.1

The authors describe an unusually diverse set of coral reef trajectories that span a pre and post heatwave. They assign potential drivers to the fate of corals including both biotic and abiotic variables and make a case for local interventions aiding both the resistance and recovery from environmental stress.

This is a nice paper but I have mixed feelings on the way it's presented. On the positive, it's a great dataset and I'd like to see that published. On the negative, I feel the pitching of the paper - addressing a limitation of empirical data on benefits of local interventions - is not compelling. I also have some questions about the data analysis and generality of the findings. However, I believe these issues can be dealt with by reframing and clarifying several aspects of the analysis.

We thank the reviewer for their positive comments and address their concern around the framing of the paper and questions about the data analysis and generality of the findings below as they expand on each of these comments.

R2.2

1) Framing:

I don't find the argument that there's a paucity of empirical evidence supporting the benefits of local interventions compelling. The authors' cite four papers that question the value of such interventions in a climate change context yet 3/4 papers are by John Bruno. Bruno is well known for views but he generally represents an extreme and certainly cannot be considered representative of the field. Part of the problem is that his arguments are either irrelevant or unsupported by data. Now I know that this process is not intended to be a review of Bruno (!) but let me point to a couple of examples. The data-rich paper cited (Baumann et al 2021) firstly tests for whether remoteness of sites (i.e., little human impact) reduces the impact of bleaching and major disturbances. They find no evidence of this but that's hardly surprising.

Other than some work on corallivory by snails in the Caribbean I don't know of any concrete theories as to why remoteness should mitigate the impacts of heatwaves or cyclones etc. It's essentially a false straw man. No one advocates MPAs as a means of reducing climate stress, for example. So that finding tells us little about the value of managing human impacts on resistance to bleaching. They then look to see whether coral recovery is faster in remote areas and once again find no significant effect. Yet, recent studies have shown that such analyses tend to have extraordinarily low power, quite easily overlooking an improvement of 10-20% in coral cover (Mumby et al 2021). Thus, their conclusions aren't surprising and a better study would quantify the sensitivity of their approach to detecting any trends.

I make these points merely to highlight that the apparent controversy this paper addresses is weaker than implied. Moreover, there is a wealth of studies that have documented impacts of improved water quality or fish biomass on processes of reef recovery / recovery potential - see the papers by SV Smith (Hawaii eutrophication reversal), Bob Richmond, Katharina Fabricius, Tom Tomascik, Mark Hay, Pete Mumby, Bob Steneck, Nick Graham, Donovan (also Hawaii).

To me, a more compelling framing concerns the difficulty in relating patterns to the specific drivers and you have an excellent opportunity within a contained reef system to do this.

We really appreciate the critique from Reviewer 2 and could not agree more. Reviewer 4 shared similar concerns with our framing of the paper (R 4.3). Reviewer 1's opening comments also highlighted that the novelty of our work lies in the high-resolution data on the proximate human drivers of reefs over time and our unique opportunity with these data to test whether integrated land-sea management benefits coral reefs under climate change. Reviewer 1 writes (in comment 1.1): “(the paper) *demonstrates the importance of integrated land and sea management. This has been a central feature of coral reef conservation since the late 1980s and early 1990s (e.g. Integrated Coastal Zone Management), but the proof of its efficacy has been wanting. Here, this paper delivers some much needed support for dismantling the conventional terrestrial and marine silos.*”

We have now re-written our *Introduction* Paragraph 2 to better highlight this knowledge gap, specifically the need to identify unambiguous targets on the combination of land-sea human impacts local resource managers should mitigate to support coral reef persistence under climate change. In doing so, we no longer need to cite the Bruno papers, or Baumann et al. (2021) as these are no longer suitable references for our statements. We also acknowledge several of the works documenting the connections between local conditions and reef integrity (including their response to acute disturbance) by the researchers that Reviewer 2 lists above (including Mumby et al. 2021). In doing so, we think the framing of our paper is now much more compelling and appropriate given our findings. This re-framing also led us to revise the *title* of our paper slightly, to focus more on the important take-home message that integrated land-sea management is key to promoting coral reef persistence under climate change.

Mumby, P. J., Steneck, R. S., Roff, G. & Paul, V. J. Marine reserves, fisheries ban, and 20 years of positive change in a coral reef ecosystem. *Conserv Biol* **35**, 1473-1483 (2021)

R2.3

2) I have several questions on the analysis but the most significant is that it wasn't clear how the sites varied in their exposure to thermal stress. Can the authors' reliably show that the coral outcome (either mortality from bleaching or recovery rate) occurred despite differences in exposure to thermal stress among sites? This needs to be highly transparent.

This is an important point and we thank the reviewer for prompting us here. Reviewer 1 made a similar comment (R 1.4), in the sense they asked for greater clarity on the DHW values experienced across our study region. We have now included a graph to summarise the variation in exposure to thermal stress

experienced across our study reefs within the main paper (please see new figure panel 'b' in Fig. 3). In summary, there was very little variation in exposure to accumulated thermal stress during the marine heatwave across our study region. DHWs varied from 9-14 weeks and the majority of reefs experienced >11 DHWs (Fig. 3b), far greater than the 8 DHW threshold expected to produce severe and widespread coral bleaching and mortality (Skirving et al. 2020). This is now summarised much more clearly and transparently in the results section of the main paper.

We also added DHW as a predictor in our generalised additive mixed-effects modelling framework (despite this adding to model complexity). DHW did not emerge as being important in explaining the variation in reef response to thermal stress. It did not feature in any of our top candidate models (please see Table S2) and had a very low overall relative importance score of 0.08 (defined as the sum of AICc model weights across all models containing each predictor, please see Table S3 for this score in the context of the importance scores for all other predictors).

We hope both of these efforts re-assure the reviewer that the reef response outcomes we document here were not a result of spatial variations in thermal stress exposure across our study region.

Skirving, W. *et al.* CoralTemp and the Coral Reef Watch Coral Bleaching Heat Stress Product Suite Version 3.1. *Remote Sensing* **12**, 3856 (2020)

A related issue concerns the conclusions regarding how different combinations of water quality and fish biomass would impact future responses to heatwaves. These implicitly assume that the distribution of heatstress among sites remains the same for each putative bleaching event. But what would happen if the areas with relatively good biological conditions (say) were exposed to the highest thermal stress?

Our results here document reef response to the most severe marine heatwave on record over the past 120 years in our study region (please see new Fig. S1, also under our response to R1.4). This led to intense and spatially homogeneous exposure of reefs to thermal stress (Fig. 3b). As such, our study rather fortuitously (from a scientific-enquiry point of view) represents a robust way to examine how variations in local land-sea human impacts and environmental factors correlate with reef response to thermal stress in the absence of any confounding effect of spatial variations in thermal stress exposure.

Climate models predict the frequency of very intense marine heatwaves, like the one documented in this study, will increase in the coming years/decades (e.g. van Hooidonk et al. 2016, Dixon et al. 2022). We have taken the reviewer's comment on board and, combined with the related comment raised by Reviewer 1 (R1.4), have updated our *Conclusion* section. We specifically discuss how these projected increases in future ocean temperatures and the frequency and severity of coral bleaching events could act to overwhelm the positive effects of any local management actions on coral reefs. Importantly though, we also now discuss the substantial variation in the projected rates of ocean warming predicted within and among countries under reduced emissions scenarios (van Hooidonk et al. 2016) and go on to state that actions that support coral reef persistence locally (like we identify here) alongside global reductions in greenhouse gas emissions may buy reefs more time to adapt and persist into the future.

Dixon, A. M., Forster, P. M., Heron, S. F., Stoner, A. M. K. & Beger, M. Future loss of local-scale thermal refugia in coral reef ecosystems. *PLOS Climate* **1**, e0000004 (2022)

van Hooidonk, R. *et al.* Local-scale projections of coral reef futures and implications of the Paris Agreement. *Scientific Reports* **6**, 39666 (2016)

R2.4

3) Many of the variables seem to be strongly correlated (fig 1 and 3). How was this dealt with explicitly? Note, for example, that the pattern of less coral mortality with greater depth is well known (as stated by authors) but so too is the negative association between depth and herbivorous fish biomass. So the apparent correlation between bleaching mortality and herbivorous fish biomass could be spurious and driven by depth. These are the sorts of issues that need to be made more compelling.

Reviewer 3 made a similar point below (R3.2) and we have now been much more explicit in the main paper *Methods* and figure captions about how we dealt with predictor variable correlations, clearly summarising which of each highly correlated pair ($r > 0.7$) was removed prior to model fitting. This helped to reduce the risk of model overfitting. The modelling framework we used also visualises the response-predictor relationships in a conditional manner, that is visualising any given response-predictor relationship while controlling for the other predictors in the model (by holding all of them at their mean).

The reviewer makes an additional point also, that concerns how the explanatory power of some of the key drivers in our GAMM models could simply reflect the factors they collineated with, regardless of whether or not these predictors had been removed prior to model fitting. Based on a modification to our response variable in response to a comment by Reviewer 3 (please see response to R3.3 for details), depth (as the Reviewer mentions above) no longer features in the top-ranking models. The five key factors in our GAMM models of coral loss are now: phytoplankton biomass, urban runoff, sedimentation, scraper biomass, and total fish biomass (please see Fig. 3). Both total fish biomass and scraper biomass correlated with total herbivore biomass, however the relationships between both total fish biomass and scraper biomass with coral loss had weak slopes (i.e., $< 5\%$ coral change) and we do not interpret them in any great detail in the paper (as suggested by Reviewers 3 and 4 in comments R3.4 and R4.8). Urban runoff did not correlate highly with any other predictor. Sediment correlates positively with annual rainfall, but we include peak rainfall as a metric and annual rainfall is the factor of course contributing to sedimentation (but we hypothesise it is sediment impacting the coral, the rainfall only impacts the corals indirectly via the sediment). Phytoplankton correlates positively with irradiance; irradiance of course fuels phytoplankton growth so this is not surprising. We now make it clear in the *Methods* section of the paper that when deciding which of each highly correlated pair of predictors to retain, that we retained the predictor for which we had a hypothesised direct link to coral loss (rather than an indirect link).

R2.5

4) Lastly, why were the reef states discretised into high, med, low rather than use analyses based on continuous scales? That usually loses data and wouldn't be ideal.

This is a good question from the Reviewer. It was a conscious decision for us to discretise the data in this way in order for the analysis and the results to speak more to the kinds of information local resource managers actually need for decision support. These needs have emerged through discussions between our research group and local resource managers across our study region over the past few years (some of whom are co-authors on this paper). Ultimately reef managers are striving to move from less to more desirable reef states (i.e., towards a higher cover of reef-building benthic organisms) and require unambiguous targets on the combination of land-sea human impacts they should mitigate to achieve this in the face of climate change. In response to a comment by Reviewer 1 (R1.2), we have modified aspects of our information delivery here in this section of the paper though. We re-extracted the needed probabilities from our model to show the conditions under which one can move from a less desirable (*Low* reef-builder cover) to more desirable state (*Moderate* or *High* reef-builder cover) over a single continuous probability surface (please see Fig. 4b). This focuses more attention on what is needed to move from *Low* to *High* reef-builder cover (rather than what to do to remain in either category as we showed previously). The ordinal logistic regression approach gave us the opportunity to create trade-off scenarios for land- and sea-based management (see new Fig. 4b panel) which could not be achieved

within the GAMM modelling framework. We were pleased to see this approach highlighted positively by Reviewer 4 (R4.10), who states “*the OLR analysis is very powerful and elegant and really smart way of analyzing and visualizing the effects of covariates on reef builder cover.*”

Referee #3 (Remarks to the Author):

R 3.1

This is an impressive study. The importance of local drivers in mitigating the impacts of climate disturbance on coral reefs is an important research front, which as the authors point out has been addressed with broad proxy data in the past. The level of detail in quantifying specific local drivers of coral reef trajectories and responses is really unique and makes this a powerful analysis of how anthropogenic, environmental and ecological factors influence coral reef persistence to global climate disturbance. The manuscript is very well written and reasoned, the figures are powerful, and the inferences important. The detail in the drivers allows for some really tangible management actions and outcomes.

We thank the reviewer for their positive comments on the novelty of our data set and analyses, the clear communication of our findings, and the potential for our findings to have societal impact.

R3.2

There are various bits of information that are missing from the main text, which would help the reader, including some that raised red flags until I got to the methods or supplemental material. These things should be brought into the main text, which can be done with few words, leaving the details for the methods.

We thank the reviewer for pointing this out. Reviewers 1 (R3.2) and Reviewer 2 (R 2.4) shared similar concerns.

These include:

- An obvious question is how correlated the various drivers are, and thus how independent they are. This is not clear from the main text or figures. There are nice figures assessing this collinearity in the supplement (Figs S2 and S4) and the legend explains that if correlations are >0.7 , one of the variables was excluded. This is appropriate, and should be explained briefly through the main text.

We have now been much more explicit in the main paper *Methods* and figure captions about how we dealt with predictor variable correlations, clearly summarising which of each highly correlated pair ($r>0.7$) was removed and why prior to model fitting.

- Frequency, number of sites, and replication for underwater surveys. In fact this information is not even in the methods, and can only be found in the supplemental material. Replication will be a key factor that readers wonder about. It looks strong from the supplemental material, but the reader shouldn't need to go that far to be sure of that.

We thank the reviewer for pointing out this omission. We have updated our *Methods* document to be more explicit about the number of reefs surveyed within each of the years included in our study. We have also included a clear delineation in data availability in the legend of Fig. 1 and provided greater detail on data inclusion in the figure captions.

- How were the reef trajectories (fig 2b) defined as positive, negative, or neutral. This information is in the methods, but half a sentence could explain that in the main text also.

A reef was considered to have a positive trajectory or negative trajectory if coral cover from the 2003 survey to the 2014 survey increased or decreased by greater than 3%, respectively (Fig. 1b). This cut off was based on the range in mean coral cover among all 23 reefs across the 12-year pre-disturbance period (range = 2.8%; min = 34.1%; max = 36.9%). We have now added this explanation to the Fig. 2 caption and this explanation has also now been added to the main paper *Methods*.

R3.3

The change in benthic cover through the heatwave (Fig 3b) seems to show that reefs with higher starting coral cover tended to lose the most coral cover. This begs the question of what the relationship was with coral composition for these changes through the heatwave. Were the sites with most loss of coral cover, also the ones with the highest cover of vulnerable coral taxa? How does this fit with your analyses / drivers.

The reviewer raises two important points: 1) what effect variations in starting coral cover across our survey reefs prior to mass bleaching might have had on the spatial variations in subsequent coral loss, and 2) how the amount of coral loss following bleaching might relate to coral community composition in the context of bleaching susceptibility.

In response to point 1, we had quantified changes in coral cover loss following bleaching and linked these to our land-sea human impacts and environmental factors in a qualitative manner, but have now extended this to more formally account for variations in starting condition in our GAMM models. This is important because reefs with higher initial coral cover have greater scope for loss (Côté et al. 2005). To overcome this and ensure comparability across reefs, we calculated coral cover change as:

$$\% \text{difference} = [(A_{a,i} - A_{b,i}) / A_{b,i}] \times 100$$

following (Graham et al. 2008), where A_b and A_a is the mean coral cover at each reef prior to bleaching in 2014 or 2015 and following bleaching in 2016, respectively. We then updated our GAMM models with this as our response variable metric of coral cover change. This modification resulted in some subtle changes to our results, specifically:

- It made the weak relationship between total fish biomass and coral loss even weaker and well within the bounds of the 80% CI (please see new Fig. S2).
- Instead of a weak relationship between grazer biomass and coral loss, the biomass of scrapers became a more important factor in our models (see Table S2). However, previously like grazers, the biomass of scrapers and coral loss had a weak slope that was well within the bounds of the 80% CI (please see new Fig. S2).
- It helped to clarify the relationship between sediment and coral loss.

We have updated Fig. 3 in the main paper and revised our results statements about the key factors explaining variations in coral loss following the marine heatwave accordingly.

Côté et al. (2005) Measuring coral reef decline through meta-analyses. *Philosophical Transactions of the Royal Society B: Biological Sciences* 360: 385-395

Graham, N. A. J. et al. Climate Warming, Marine Protected Areas and the Ocean-Scale Integrity of Coral Reef Ecosystems. *Plos One* 3, e3039 (2008)

In response to point 2, we do not have a specific measure of bleaching susceptibility by coral taxa. However, we do have the percentage cover of each coral taxa pre- and post-disturbance at a subset of

reefs (N=40; analysis data collected from the other 40 reef surveys did not include individual coral taxa). Below we show the relationship between the percentage cover of the six most dominant coral species (*Porites lobata*, *Porites compressa*, *Porites evermanni*, *Pocillopora meandrina*, *Montipora capitata* and *Porites rus*) pre-bleaching in 2014/2015 and absolute coral cover loss at those same sites post-bleaching in 2016. Please note that the linear fits should not be interpreted as a formal model-fitting process, they are to aid in interpretation here only.

The pre-bleaching cover of the most dominant species among these six (*P. lobata*, *P. compressa*, and *P. evermanni*) all show a positive relationship with overall coral cover loss post-bleaching. This again provides motivation for accounting for variations in coral cover starting condition prior to modelling the drivers of coral loss following bleaching. When starting condition is accounted for using the approach outlined above (*in response to Point 1*), these positive relationships no longer hold and the confounding effect of starting condition of the major coral taxa on coral cover change between pre- and post-bleaching goes away:

R3.4

Some of your drivers of change through the heat stress (Fig 3c) have very weak slopes (total fish biomass, grazer biomass, and to a less extent sediment load). While I see these factors were strongly supported by AICc model weights, the slopes are within the 80% CI's, so you should temper your language somewhat in the text regarding these drivers.

This is an important point and Reviewer 4 made a similar comment (R 4.8). We have done the following things in response to this:

- 1) Tempered our language with regards to discussing the role these drivers play in explaining variations in coral loss following the marine heatwave, noting the “weak slopes” directly in the main manuscript.
- 2) We have moved the plotted GAMM relationship between total fish biomass and coral loss, and the relationship between scraper biomass and coral loss (which both had weak slopes well within the bounds of the 80% CIs) out of our main Fig. 3 and into a Supplemental Figure (Fig. S2). We note this within the figure legend of Fig. 3 for the reader.

R3.5

For the post-disturbance analysis, you bring in CCA with hard coral cover. I see the rationale for this, but it may be useful to have a supplemental figure /analysis that runs the same analysis (proportion low, medium and high, and how changes before versus after) for just hard coral cover, to tie the narrative through from the earlier figures.

We appreciate the suggestion by the reviewer. We combined hard coral and CCA for our post-bleaching analysis as a way to assess the recovery of the dominant reef-building organisms on tropical coral reefs (Smith et al. 2016). This is important because certain species of CCA are key for: 1) fusing the reef framework together and promoting coral settlement and recruitment (Price 2010), and 2) suppressing the growth of competitive algae (Vermeij et al. 2011). CCA therefore serve a vital role in overall recovery of reef-building benthic organisms post-disturbance (as the Reviewer clearly appreciates). Given that the time window we have recovery data for is just four years, we felt it irresponsible to focus only on corals as their growth and development into larger adult colonies can take time (up to 10 years or more, Gilmour et al. 2013) and any analysis that focuses solely on them could be misleading. For example, mean coral cover in 2019 (4 years post-disturbance) was $17.5\% \pm 2.8$ (95% CI), meaning the threshold for “High” calcified cover from this distribution becomes 24% (25 and 75 percentile thresholds are 9% and 24% cover, respectively), which is not a target we think local resource managers should be aiming for.

We have taken care to explain the justification for this much more clearly in the main paper now.

Gilmour, J. P., L. D. Smith, A. J. Heyward, A. H. Baird & M. S. Pratchett. Recovery of an Isolated Coral Reef System Following Severe Disturbance. *Science*, 340, 69-71 (2013)

Price, N. Habitat selection, facilitation, and biotic settlement cues affect distribution and performance of coral recruits in French Polynesia. *Oecologia*, 163, 747-758 (2010)

Smith, J. E. *et al.* Re-evaluating the health of coral reef communities: baselines and evidence for human impacts across the central Pacific. *Proceedings of the Royal Society B: Biological Sciences* **283**, 20151985 (2016)

Vermeij, M. J. A., M. L. Dailer & C. M. Smith. Crustose coralline algae can suppress macroalgal growth and recruitment on Hawaiian coral reefs. *Marine Ecology Progress Series*, 422, 1-7 (2011)

R3.6

The conclusion section is brief. That in itself is fine, however given you have different specific drivers as important in the pre (trajectories), during heat stress, and post disturbance recovery sections of the paper, this really needs to be unpacked a bit here. You need to consolidate this information and explain what it means in terms of management. For example, you have some nice tangible recommendations from the analysis in figure 4, but wastewater was not important in Fig 3, and urban runoff was important in figs 2 and 3 (among other variables). Some thought and recommendations across the disturbance periods and drivers would be useful here, perhaps in the context of ongoing anthropogenic drivers and heat extremes.

This is a great suggestion by the reviewer. We have now completely expanded the *Conclusion*, including the first paragraph to explicitly discuss this idea. We use the relationships between herbivorous fishes and reef persistence over our different time windows pre- and post-disturbance as an example, stating that supporting reef persistence likely requires a diverse array of herbivores with contrasting feeding and behaviours that play key functional roles at different points in time (Bellwood et al. 2004, Chong-Seng et al. 2014). We then highlight the same is true for land-based stressors within our results and that achieving management outcomes will require mitigating the combination of local factors that support reef persistence across all successional stages pre-, during and post-climate driven disturbances.

Bellwood, D. R., T. P. Hughes, C. Folke & M. Nyström. Confronting the coral reef crisis. *Nature*, 429, 827-833 (2004)

Chong-Seng, K. M., K. L. Nash, D. R. Bellwood & N. A. J. Graham Macroalgal herbivory on recovering versus degrading coral reefs. *Coral Reefs*, 33, 409-419

R 3.7

Related to the above point, while the 30 by 30 is a good hook, I wonder if protected areas are really what your findings point to. Perhaps in part, but the actions needed based on your detailed study can be more specific and thus useful for policy makers I think – such as waste water management, fisheries governance etc.

We thank the Reviewer for this comment. We have modified our final paragraph of the *Conclusion* to better articulate the key take-home point from our findings. Specifically, we've added that in most coastal geographies, 30% land preservation is likely impractical given the high proportion of people living near the ocean. Instead, we suggest an integrated management approach that addresses land-based stressors like wastewater pollution, together with fisheries governance, is ultimately required to achieve successful ocean conservation outcomes.

Other minor points:

R3.8

You have some redundancy between the opening paragraph of the main text and the start of the abstract.

We thank the reviewer for bringing this to our attention. In our updated manuscript, we have strived to remove redundancy as much as possible, but noting that some redundancy is acceptable according to the latest journal guidelines.

R3.9

Another point to perhaps allude at the start is that in some well studied systems, land based and marine based drivers are not both strong. For example, fishing pressure on the GBR is very light and doesn't

target the herbivores. A strength of your study system is being able to disentangle these things in some detail, which isn't possible everywhere.

We thank the reviewer for their positive comment on the suitability of our study system for this question. Following suggestions from Reviewer 2 (R 2.2) and Reviewer 4 (R 4.3) we have changed the focus of the second paragraph in our *Introduction* to better highlight the novelties around our generated data sets and analyses. As such, the strength of our study system and approach are now better articulated here we think.

R3.10

Line 137, it is not clear what you mean by 'historically less exposed' here. If you have used a lag function in assessing the drivers, that needs to be clear.

We realize that this statement was unclear in our previous version of the manuscript. We now are more explicit with per cent differences of land-based stressors on negative trajectory reefs compared with positive trajectory reefs and avoid vague language such as that pointed out by the reviewer.

Referee #4 (Remarks to the Author):

R4.1

A. Summary of key results: the authors show that benthic community dynamics on reefs located on the western coast of Hawai'i correlate with a variety of land- and sea-based variables that are indicative of local conditions and, for the most part, 'manageable.' First, they show that general trends in coral cover (used throughout as the indicator of reef condition) prior to the mass bleaching event show distinct trends with regards to a range of variables, including herbivorous fish biomass, wave exposure, human population density, or wastewater pollution. They then highlight that a similar suite of variables appears to have mediated the condition of reefs after a severe bleaching event. Finally, they hone in on the most important drivers of reef condition in an ordinal framework, demonstrating that scraping herbivores and wastewater pollution combined appear to have the greatest effect on reef recovery. The authors use their results to suggest that local human management efforts can help ameliorate the effects of temperature-mediated bleaching on coral reefs.

We thank the reviewer for a nice succinct summary of our findings.

R4.2

B. Originality and significance: the quality of this paper is outstanding. It is well written, thoughtfully analyzed, beautifully visualized, and generally very well done. The consideration of so many high-resolution variables in a single framework is rather novel compared to existing work and produces some really interesting results that I believe are very valuable.

We appreciate the positive remarks by Reviewer 4 on the quality of our paper and the value of our data and findings.

However, with regards to overall originality and significance, I am not convinced that the advances made in the paper truly meet the expectations of the journal. I base this assessment on three elements, which include 1) the framing of the paper as a counterpoint to recent work claiming that local management does not help corals against bleaching, 2) the limited spatial scale and inherent context-dependency of the results, and 3) the fact that the major implications (managing fish populations and terrestrial runoff benefits coral reefs) are fairly well-established.

We respond to each of these comments as they are expanded on by Reviewer 4 below.

R4.3

1) *The paper is framed as a counterpoint to a set of papers that claim that local conditions do not significantly modify the resilience of reefs to coral bleaching (e.g. Hughes et al. 2017; Bruno et al. 2019; Bruno & Valdivia 2016). I have several issues with this and would argue that it's a bit of a strawman. For example, Bruno et al. 2019 specifically argue that they do not find an effect of MPAs (read: protection of herbivorous fishes) for coral reef *resilience* to stressors, but this is different from the recovery dynamics that are described in this paper. Similarly, Hughes et al. 2017 do argue that local management does not afford any protection from bleaching per se, but they notably do not say anything about recovery. Thus, both of these papers do not actually pose a counterpoint to the results of this paper as currently framed, and while I do think that some of the verbiage in these papers is unnecessarily unbalanced, I strongly believe that the vast majority of reef scientists and practitioners can see beyond this artificial dichotomy of local vs. global management of reefs. This means, in turn, that I think we ought to de-escalate this discussion rather than highlight it (as done in l. 74-80), but that is perhaps more of a personal preference. In any case, I would argue that there aren't a lot of scientists in this world who would refute the hypothesis that efficient management of local stressors benefits coral reefs in the face of global warming.*

We agree with the reviewer and similar concerns around the framing of our paper and knowledge gap were raised by Reviewer 2 (R 2.2). As a result, we have now re-written our *Introduction* Paragraph 2 to better highlight the knowledge gap addressed by our study, specifically the need to identify unambiguous targets on the combination of land-sea human impacts local resource managers should mitigate (and can actually mitigate) to support coral reef persistence under climate change (please also see our response to Reviewer 2 comment R2.2 above). In doing so, we think the framing of our paper is now much more compelling and accurate. This re-framing also led us to revise the *title* of our paper slightly, to focus more on the important take-home message that integrated land-sea management is key to promoting coral reef persistence under climate change.

R4.4

2) *One of the strengths of the paper is the precise identification of two main drivers that have facilitated coral recover in the aftermath of the bleaching event—scraping parrotfishes and wastewater runoff. This is a great source of information for local management and provides tangible targets, but it does so (at this stage) almost exclusively for this particular island and possibly other Hawaiian islands. Hawaiian reefs are pretty unique in their geology, geographic positioning, climate, reef communities, and anthropogenic stressors and management, so at this point, I think we have to interpret these results as a very cool and useful phenomenological suite of findings that apply to Hawai'i, rather than extrapolating to coral reefs in general.*

We appreciate that the reviewer sees the value in our data and findings, however we maintain our results are meaningful for other coral reefs beyond our study region for several reasons. First, the eight core processes supporting coral reef functioning recently identified by Brandl et al. (2019) (*CaCO₃ production, bioerosion, primary production, herbivory, secondary production, predation, nutrient release, and nutrient uptake*) are all present within our study system, like any other coral reef ecosystem. Second, we can learn a lot about how coral reefs work from more geographically-focused, region specific studies like this, as previous published examples in *Nature* have shown. For example, Graham et al. 2015 *Nature* focused on ~58 km of latitude in the Seychelles and Graham et al. 2019 *Nature* focused across <200 km of latitude in the Chagos Archipelago, which is a similar geographic scope to the ~180 km of latitude covered by our present study.

Graham, N. A. J., Jennings, S., MacNeil, M. A., Mouillot, D. & Wilson, S. K. Predicting climate-driven regime shifts versus rebound potential in coral reefs. *Nature* **518**, 94-97 (2015)

In other words, an atoll in the Indian Ocean or Coral Sea is likely to benefit very little from human wastewater management or protection of herbivorous fishes (at least for the benthic community; Graham et al. 2020), but could probably benefit from other management actions (e.g., rat eradication, Graham et al. 2018; reef restoration, Lamont et al. 2022). Of course, the broader derivation of the results (that local management is good for reefs) is much more applicable to reefs worldwide, but this is also very well known and broadly documented (see 3) below).

The reviewer makes a good point about the fact that the ecological response of coral reef communities to marine protected areas (MPAs) is changing over time due to the changing disturbance dynamics in the Anthropocene and cites Graham et al. (2020) *Nature Communications*. However, Graham et al. (2020) does not explicitly investigate this concept in terms of wastewater management and work by the same authors in the same Seychelles study system has shown the importance of local factors like nutrients and herbivore biomass in driving patterns of coral reef benthic community recovery following mass bleaching (Graham et al. 2015). It's therefore likely that within an inhabited Indian Ocean or Coral Sea system, that local human drivers do still play a role in governing reef dynamics following disturbance (if local human populations are present), but these effects may not always be quantifiable based on the data at hand.

Graham, N. A. J. *et al.* Changing role of coral reef marine reserves in a warming climate. *Nature Communications* **11**, 2000 (2020)

Graham, N. A. J., S. Jennings, M. A. MacNeil, D. Mouillot & S. K. Wilson. Predicting climate-driven regime shifts versus rebound potential in coral reefs. *Nature*, 518, 94-97 (2015)

R4.5

3) Overall, I think there is overwhelming evidence that local management matters. In fact, especially with regards to fisheries management and pollution (the land- and sea-elements of the present paper), one of the co-authors published a paper not too long ago that made a very convincing case that globally, management of fish populations and pollution can benefit reefs in the face of bleaching events (Donovan et al. 2021). Beyond this recent paper, there is a plethora of work that shows these trends in isolation at more local scales of reef recovery after bleaching (e.g., Mumby et al. 2021; Steneck et al. 2019 for fishing effects; Mellin et al. 2019; MacNeil et al. 2019 for water quality effects). Of course, this is related to the beneficial effects of restricting human impacts on ecosystems as a whole, which have been well established for coral reefs for several decades.

We agree with the Reviewer that numerous prior works have identified salient connections between local conditions and coral reef ecosystem structure and function, including their resistance to and recovery potential following mass bleaching (e.g., Graham et al. 2015, MacNeil et al. 2019, Asner et al. 2022, Donovan et al. 2021, Mumby et al. 2021). However, many of these past efforts have been forced to use proxies for direct local human impacts like human population density (e.g. Smith et al. 2016), the abundance of macroalgae in the system (e.g. Donovan et al. 2021), reef accessibility (e.g. Maire et al. 2016), or composite indices like 'water quality' (e.g. MacNeil et al. 2019) that can be affected by anything from deforestation (Maina et al. 2013) to aquaculture (Hozumi et al. 2018). Such proxies do not identify meaningful levers local resource managers can pull and are less likely to result in policy change or successful conservation outcomes. As Reviewer 1 notes (R1.1) the high-resolution human driver data we have generated are often missed in larger-scale studies. This sentiment is echoed by Reviewer 3 who states (R3.1) "*The importance of local drivers in mitigating the impacts of climate disturbance on coral reefs is [an] important research front, which as the authors point out has been addressed with broad proxy data in the past.*" Donovan et al. (2021) (which was a global study) provides important foundational work on how pollution can increase coral loss following marine heatwaves, but identifying the specific underlying local human impacts that contribution to coastal pollution is needed for management decision making. Many of these prior efforts are unable to zoom in to a scale relevant to

resource managers (i.e., the spatial scales in which resource management decisions are made). The data we have generated allow us to zoom in to these scales and identify the specific direct human drivers underpinning issues like ‘water quality’ and to what level they require mitigating in our study system in order to promote reef persistence under climate change. More importantly, these high-resolution data and our study system allow us to demonstrate that only by adopting an integrated management approach that addresses land-based stressors like wastewater pollution, together with fisheries governance, will we achieve successful ocean conservation outcomes for coral reefs. Of course, reducing human impacts at local scales to maintain coral reef integrity has been the guiding paradigm of coral reef conservation for decades (McLeod et al. 2019). Local resource managers have long aspired to an integrated land-sea approach (Marshall, Schuttenberg and West 2006), but the proof of its efficacy above either approach in isolation has been wanting as most terrestrial and ocean conservation efforts remain siloed (Taljaard et al. 2012). As Reviewer 1 states (R1.1), “*this paper delivers some much needed support for dismantling the conventional terrestrial and marine silos.*”

We hope this explanation emphasises more clearly to the Reviewer why our study is novel and timely. We previously failed to do this properly in the framing of the originally submitted version of the paper and hope our new framing, and in particular the revised Paragraph 2 of the *Introduction* (including making reference to Mumby et al. 2021 that the Reviewer cites above), does a better job of this.

Graham, N. A. J., Jennings, S., MacNeil, M. A., Mouillot, D. & Wilson, S. K. Predicting climate-driven regime shifts versus rebound potential in coral reefs. *Nature* **518**, 94-97 (2015)

MacNeil, M. A. *et al.* Water quality mediates resilience on the Great Barrier Reef. *Nature Ecology & Evolution* **3**, 620-627 (2019)

Donovan, M. K. *et al.* Local conditions magnify coral loss after marine heatwaves. *Science* **372**, 977-980 (2021)

Asner, G. P. *et al.* Mapped coral mortality and refugia in an archipelago-scale marine heat wave. *Proceedings of the National Academy of Sciences* **119**, e2123331119 (2022)

Smith, J. E. *et al.* Re-evaluating the health of coral reef communities: baselines and evidence for human impacts across the central Pacific. *Proceedings of the Royal Society B: Biological Sciences* **283**, 20151985 (2016)

Maire, E. *et al.* How accessible are coral reefs to people? A global assessment based on travel time. *Ecol Lett* **19**, 351-360 (2016)

Maina, J. *et al.* Human deforestation outweighs future climate change impacts of sedimentation on coral reefs. *Nature Communications* **4**, 1986 (2013)

McLeod, E. *et al.* The future of resilience-based management in coral reef ecosystems. *J Environ Manage* **233**, 291-301 (2019)

Marshall, P. A., Schuttenberg, H. Z. & West, J. M. A reef manager's guide to coral bleaching (2006)

Taljaard, S. *et al.* Implementing integrated coastal management in a sector-based governance system. *Ocean Coast Manage* **67**, 39-53 (2012)

R4.6

In saying this, I do not mean to diminish the presented results in any way. As stated initially, I think that this is an excellent paper that is worthy of being published in a good journal and that deserves attention. However, given that the instructions are to evaluate the novelty and impact of the paper, I felt compelled to provide my evaluation in light of the existing literature and prevailing knowledge base.

We appreciate the Reviewer's positive comments here and hope that our re-framing of the paper helps to better highlight the novelty of our paper in the context of past works.

R4.7

C. Data and methodology: the dataset is an extensive time series that covers a strong disturbance event and is thus well suited for gauging reef recovery, albeit at a limited spatial scale. The covariates the authors use are of higher resolution than commonly seen and permit a detailed assessment. As stated in the beginning, the overall quality of the manuscript is very high.

We thank the reviewer for their positive comments.

R4.8

D. Use of statistics and treatment of uncertainties: overall, the analytical frameworks the authors used are well-suited for the task and quite elegant. However, I have some issues with the interpretation of the GAMMs, which the authors state show effects of various covariates (e.g. depth, herbivore biomass, fish biomass, sedimentation) on coral bleaching recovery (l. 178-215). Looking at Fig. 3 (which I understand shows model-averaged partial effects), I see reasonable effects of depth (an unequivocally unmanageable covariate) and urban runoff on the recovery dynamics of coral cover, but the other effects appear to be basically negligible. For example, total fish biomass over its entire range appears to shift coral recovery dynamics between approximately -12 and -10% with substantial uncertainty associated with these estimates. Similarly, sediment input appears to move coral recovery from -11% to perhaps -15% over its entire range, again showing substantial uncertainty. Are these really meaningful ecological effects? I would advise to be cautious with this, since GAMMs can be a bit quick to produce 'significant' effects and high R2 values, especially in models with lots of parameters. I am rather skeptical as to whether there's a meaningful relationship in several of these covariates.

The Reviewer makes a good point and Reviewer 3 made a similar comment (R 3.4) with regards to toning down some of our interpretations of the response-predictor relationships in the GAMM outputs that had weaker slopes (within the bounds of the 80% CIs). We have done this now and actually removed those with weak slopes from the main Fig. 3 in the paper and into a Supplemental Figure (Fig. S2) to de-emphasise their interpretation. It is important to note here though that based on a modification to our response variable in our GAMM models in response to a comment by Reviewer 3 (please see response to R3.3 for details), two key things happened:

1. It made the weak relationship between total fish biomass and coral loss even weaker and well within the bounds of the 80% CI (please see new Fig. S2).
2. Instead of a weak relationship between grazer biomass and coral loss, the biomass of scrapers became a more important factor in our models (see Table S2). However, previously like grazers, the biomass of scrapers and coral loss had a weak slope that was well within the bounds of the 80% CI (please see new Fig. S2).

R4.9

On that note, it would also be great to see the raw data superimposed on the regression lines to get a better idea of model fit, which is impossible to see from the AIC values.

This is a very good suggestion. We have now added the underlying data points to the model fits (Fig. 3d and Fig. S2).

I had a similar issue with the first analysis (the quantification of the pre-disturbance dynamics), which is essentially based on a categorical descriptor based on >3% change in coral cover over 10 years. While the authors do provide a rationale for that value based on the range of observed coral cover values (l. 500), I was rather surprised by the choice of such a small value. To me, a 3% change in coral cover does not symbolize a directional shift that merits a categorical assignment of positive or negative, so I would at least expect some kind of sensitivity analysis for this section that can support the categorical assignment with some larger values of change.

We appreciate this enquiry from the reviewer. Below is a short recap of our approach and results, as well as a comparison between our threshold of 3% and 5% (see table below).

We chose 3% as the threshold for whether a reef had a positive trajectory (i.e., an increase >3% between 2003 – 2014) or negative trajectory (i.e., a decrease of > -3% between 2003 – 2014) based on the range in mean coral cover among all 23 reefs across the 12-year period (range = 2.8%; min = 34.1%; max = 36.9%; Fig. 2a).

With the 3% threshold, mean coral increase for positive trajectory reefs was 9.2% (min = 4.3%, max = 18%; N=10) and the mean coral loss for negative trajectory reefs was -12.6% (-6.6%, -16.8%; 8). To determine the per cent difference in local land-sea human impacts and environmental factors as either positive, negative, or no difference, we used a cut off of 0 ± 2 standard deviations from the median difference among all factors, calculated using bootstrap with replacement (10,000 iterations). The cutoff was 19.5%.

With the 5% threshold, mean coral increase for positive trajectory reefs was 11.2% (min = 6, max = 18; N=7). The summary statistics for mean coral loss for negative trajectory reefs were the same as the 3% threshold (12.6%, -6.6%, -16.8%; 8). The cutoff to determine the per cent difference as either positive, negative, or no difference was 16.5% (using the same method described above).

In the summary table below, that factors in blue were higher on positive trajectory reefs, grey is no difference, and red were higher on negative trajectory reefs. While per cent differences in some factors changed slightly, the overall patterns hold whether we use a 3% or 5% threshold. Increasing beyond a 5% threshold results in a decrease in the respective sample numbers used in the analysis. For example, an 8% threshold results in an N=5 for positive trajectory reefs. While this portion of our paper is meant to be a straightforward and general comparison of local conditions on reefs with divergent trajectories, we hesitate to increase the threshold as it would reduce the sample number and undermine the comparisons. As such, we have maintained a 3% threshold in our resubmitted manuscript.

Factor	Percent Difference (3% threshold)			Percent Difference (5% threshold)		
	Mean	Lower Bound	Upper Bound	Mean	Lower Bound	Upper Bound
Browsers	112.6	10.8	13.0	111.5	13.9	13.4
Scrapers	63.9	20.7	16.2	64.9	26.5	19.4
Human Population	63.4	16.2	44.4	59.6	22.0	48.3
Grazers	27.7	18.7	18.7	38.2	20.0	19.3
Wave Exposure	27.4	13.8	17.5	18.9	21.0	20.7
Herbivores	27.2	14.5	23.7	31.5	16.8	24.7
Total Biomass	24.3	10.9	21.1	17.6	16.6	23.3
Fishing Gear	12.8	9.2	4.6	15.9	9.2	5.5
Depth	4.8	3.6	4.9	0.8	5.3	5.4
Sediment Input	1.8	34.1	43.5	12.4	36.8	45.5
SST Mean	0.3	0.1	0.0	0.3	0.1	0.1
Annual Rainfall	-1.7	-23.6	-36.0	7.8	27.5	37.8
SST Variability	-8.2	-1.5	-1.0	-8.7	-1.8	-1.2
Peak Rainfall	-9.1	-21.6	-27.2	-5.3	-27.1	-27.9
Phytoplankton Biomass	-17.5	-5.5	-3.3	-16.1	-6.3	-4.4
Urban Runoff	-46.4	-33.3	-46.5	-52.0	-22.7	-50.7
Nutrient Loading	-74.7	-47.0	-43.3	-53.8	-56.4	-50.4
Wastewater Pollution	-79.8	-31.7	-49.5	-64.4	-37.4	-56.4

R4.10

With that being said, the OLR analysis is very powerful and elegant and really smart way of analyzing and visualizing the effects of covariates on reef builder cover. My only question here would be why the authors decided to expand this analysis to all reef-building organisms, rather than restricting it to live corals only as in the previous analysis. There wasn't really a convincing rationale in the paper for this decision, as far as I could tell.

We did not explain this well enough in the paper and apologise for this oversight. Reviewer 3 made a similar comment (R3.5) and we have pasted our response again below for convenience.

We combined hard coral and CCA for our post-bleaching analysis as a way to assess the recovery of the dominant reef-building organisms on tropical coral reefs, not just corals (Smith et al. 2016). This is important because certain species of CCA are key for: 1) fusing the reef framework together and promoting coral settlement and recruitment (Price 2010), and 2) suppressing the growth of competitive algae (Vermeij et al. 2011). CCA therefore serve a vital role in overall recovery of reef-building benthic organisms post-disturbance (as the Reviewer clearly appreciates). Given that the time window we have recovery data for is 4 years, we felt it irresponsible to focus only on corals as their growth and development into larger adult colonies can take time (up to 10 years or more, Gilmour et al. 2013) and any analysis that focuses solely on them could be misleading. For example, mean coral cover in 2019 (4 years post-disturbance) was 17.5% ± 2.8 (95%CI), meaning the threshold for “High” calcified cover from this distribution becomes 24% (25 and 75 percentile thresholds are 9% and 24% cover, respectively), which is not a target we think local resource managers should be aiming for.

We have taken care to explain the justification for this much more clearly in the main paper now.

Gilmour, J. P., L. D. Smith, A. J. Heyward, A. H. Baird & M. S. Pratchett (2013) Recovery of an Isolated Coral Reef System Following Severe Disturbance. *Science*, 340, 69-71

Price, N. (2010) Habitat selection, facilitation, and biotic settlement cues affect distribution and performance of coral recruits in French Polynesia. *Oecologia*, 163, 747-758

Smith, J. E., R. Brainard, A. Carter, S. Grillo, C. Edwards, J. Harris, L. Lewis, D. Obura, F. Rohwer, E. Sala, P. S. Vroom & S. Sandin (2016) Re-evaluating the health of coral reef communities: baselines and evidence for human impacts across the central Pacific. *Proceedings of the Royal Society B: Biological Sciences*, 283, 20151985

Vermeij, M. J. A., M. L. Dailer & C. M. Smith (2011) Crustose coralline algae can suppress macroalgal growth and recruitment on Hawaiian coral reefs. *Marine Ecology Progress Series*, 422, 1-7

R4.11

E. Conclusions: as highlighted in my comments above, I think that some of the conclusions aren't as well supported by the presented effects sizes as the authors claim. This definitely merits reconsideration and a thoughtful ecological interpretation.

We agree and based on some similar comments from the other reviewers (e.g., R1.4, R1.5, R2.3, R3.6, R3.7) we have revised our *Conclusion* section appropriately.

R4.12

F. Suggested improvements: overall, I would suggest to step away from the artificial dichotomy between local vs. global stressors and management of coral reefs with the framing of the paper. Other than that, the few analytical/interpretational comments should provide some food for thought.

We agree that our opening pitch around local *versus* global stressors was not useful and have substantially revised Paragraph 2 in the *Introduction* to better reflect the knowledge gaps and novelty of our work (please also see response to Reviewer comment R2.2 who shared similar concerns).

R4.13

G. References: the paper appropriately references previous work and I am completely aware of the space restrictions that come with submission to Nature.

We appreciate the comment by the Reviewer here.

R4.14

H. Clarity and context: as mentioned previously, the paper is well written but could be framed differently.

Please see above response to R4.12

Reviewer Reports on the First Revision:

Referees' comments:

Referee #1 (Remarks to the Author):

The authors did a solid job of responding to most of my comments and those of the other reviewers. This has really helped improve the transparency of what was done, and helped to highlight the novelty a bit more. I think the more clearly articulated focus on the original contribution here being the quantification of how integrated land-sea management is more than the sum of its parts (rather than showing that local condition matter, which I think the initial draft was bogged down in). I suggest adding land-sea to the title if space (i.e. integrated land-sea management). I support publication of the article pending some minor adjustments, which I think could help the authors better navigate Reviewer 4's points 2 (R4.4 in the response to reviewers).

Specifically, I disagree with reviewer 4 that the locality of the study should preclude publication, that the findings need to be applicable to every single reef on the planet (reviewer 4 observes that reducing wastewater is unlikely to make a difference in a remote, uninhabited atoll that doesn't wastewater because there are no people, but nearly 60% of the world's reefs are within 30 minutes of a human settlement and so this study's findings would indeed matter for the majority of the world's coral reefs), or that Hawaii is too much of an anomaly to warrant publication. On this latter point, the authors could illustrate where Hawaii fits in the global/regional context which would help demonstrate the relevance of this case study- or at the very least allow them to discuss key aspects that might not be generalisable. Reviewer 4 states that Hawaii has such unique "geology, geographic positioning, climate, reef communities, and anthropogenic stressors and management" that lessons simply aren't applicable to other places. While every place on the planet is by definition unique, I am dubious of the reviewer's claim that Hawaii is so anomalous that lessons are not applicable to other locations- but that is just my opinion. I suggest that the authors use some global studies to demonstrate where Hawaii fits into the global distribution for several of these issues (ones which are readily available). This actually builds of my earlier suggestion, which the authors did not take up, which is to contextualise the marine heatwave in Hawaii relative to the global or regional distribution. I think perhaps I wasn't totally clear, I suggest you plot the regional or global distribution of several key issues that the reviewer points out- say climate, anthropogenic stressors, management, etc. - and then mark where Hawaii sits on that distribution using a carpet plot at the bottom. This would allow you to quantify how unique or general Hawaii is- for example, you could take the distribution of global coral cover (even broken down by functional type) and the distribution of management (i.e. you could look at the proportion openly fished, restricted, and in marine reserves) from Emily Darling's 2019 Nature Ecology and Evolution paper and compare where Hawaii sits in the global distribution. You could look at some key human stressors (e.g. gravity, wastewater) in the 2022 A global map of human pressures on tropical coral reefs paper <<https://conbio.onlinelibrary.wiley.com/doi/full/10.1111/conl.12858>> and demonstrate where Hawaii sits. You could look at climate (e.g., mean SST, DHW) throughout the Pacific (or globally) from the 2016 heatwave and highlight where Hawaii sits. Mean SST, Etc. If Hawaii sits 2 SD outside of the global distribution on all of these factors simultaneously, then the reviewer has a point. However, if Hawaii falls within the 2SD of the global distribution for most of these, then I would argue that you have reasonably demonstrated that Hawaii isn't so anomalous after all. If it falls outside for one or two, then I think it is reasonable to discuss this and any implications this may have on the generalisability of the findings. Quite simply, demonstrate, don't argue...

Referee #2 (Remarks to the Author):

I appreciate the efforts the authors' have gone to in revising their paper to all four referees. I'm happy with the revision pending one minor clarification. I see the revision how cites Mumby et al

2021 *Cons Biology* in the context of local drivers of reef response to protection of herbivory after disturbance. That's an appropriate citation and it's good to see the paper link to other studies that have provided such evidence.

However, my original comment regarded the challenge of finding evidence of drivers of reef recovery - Mumby et al 2021 *Conservation Letters*. This is a different paper and I think it's relevant for the context here - the main strength of the authors' paper is that it has the power to articulate management drivers of reef dynamics post-bleaching. Many studies do not as was shown in the *Conservation Letters* paper.

Referee #4 (Remarks to the Author):

The authors have improved the manuscript following my initial comments, especially with regards to the overall framing. As indicated throughout my initial review, I think the manuscript is of high quality and provides interesting findings derived from a high resolution temporal dataset. Nevertheless, I am still not convinced that the paper offers the strong, unequivocal advances and notes of novelty one would expect from a manuscript published in *Nature*. The work is now fully centered on highlighting the role of integrative management in ameliorating land-based and marine stressors to help coral reefs maintain or bolster populations of reef-building organisms through various phases of disturbance (pre, during, and post). Yet, the evidence provided to support these claims is still relatively weak (despite the high quality of the underlying dataset), not only due to the factors I mentioned in my original review (extremely limited spatial scale, high context-dependency of coral reef dynamics and management actions), but also based on the statistical outcomes.

For the first analysis (pre-disturbance trajectories), there are strong correlations between several metrics of coral reef fish populations and reefs having a positive trajectory for coral cover, but it's impossible to infer causation from these results. Indeed, the strongest correlation appears to be for browsing, herbivorous reef fishes, which, according to the authors represented <10% of the total herbivore biomass on reefs. Thus, it is at least questionable whether these fishes could actively influence the trajectory of a reef. In turn, higher coral cover and resulting habitat complexity may simply support more fish, which would explain the positive relationships with fish-related variables across the board (cf. Russ et al. 2021; Darling et al. 2017). The negative correlations for the land-based stressors are intuitive, and directionality of the effect is not an issue.

During the heatwave, the authors provide somewhat compelling evidence that urban runoff and sediment input exacerbate the response of corals to bleaching while phytoplankton concentration appears to have a positive effect. While these relationships are interesting (especially for the phytoplankton), the latter is also decidedly unmanageable when it comes to marine policy, so it doesn't really support the main message of the manuscript ("Integrated management promotes coral reef persistence under climate change"). In contrast, the variables that can be targeted by marine policy decisions (fish biomass and herbivorous fishes, in particular) showed no meaningful relationship with coral cover throughout the disturbance whatsoever.

Finally, the post-disturbance recovery analysis is, in my opinion, the strongest part of the manuscript, as it actually shows the potential synergistic effects of managing a land-based stressor and (possibly) fish populations on coral reefs. Of course, the directionality of the correlation between scraper biomass and reef-builder cover is subject to the same uncertainty as mentioned above (i.e., bottom up effects of reef structure and architecture rather than top-down effects of scrapers on the reef itself). For example, DeMartini et al. (2013, *J. Coastal Conservation*) showed that parrotfishes preferentially recruit to branching corals, notably in the same location as the present study, with a strong effect of sedimentation on both corals and parrotfishes.

With these comments, I am not insinuating that the authors' conclusions are invalid. As I mentioned in my original review, I believe that the findings of the manuscript are intuitive based on what we know about coral reefs and especially their response to land-based stressors. However, I do believe that publication in Nature calls for extraordinary findings backed by extraordinarily strong evidence. I do not wish to make myself the authority and gatekeeper of what does (and doesn't) qualify as such, and I certainly support one of the other reviewers' opinion that the field of coral reef ecology can do with results that highlight the ability of societies to manage reef dynamics. Nevertheless, I am not 100% convinced that the manuscript provides unequivocal evidence for the consequences of integrative land-sea based management and since the reviewer instructions call for assessments of novelty and impact, I feel compelled to present my evaluation of manuscript with regards to these criteria.

Thus, overall, I stand by my original opinion, which is that the manuscript is of high quality with no methodological concerns, but that the evidence presented within the paper does not necessarily hit the high notes of strength of evidence, novelty and impact that I usually associate with the journal.

Author Rebuttals to First Revision:

Referees' comments:

Referee #1 (Remarks to the Author):

The authors did a solid job of responding to most of my comments and those of the other reviewers. This has really helped improve the transparency of what was done, and helped to highlight the novelty a bit more. I think the more clearly articulated focus on the original contribution here being the quantification of how integrated land-sea management is more than the sum of its parts (rather than showing that local condition matter, which I think the initial draft was bogged down in). I suggest adding land-sea to the title if space (i.e. integrated land-sea management).

We thank the Reviewer for their continued encouragement and thoughtful critique of our paper. We have drafted a revised title of “*Integrated land-sea management promotes coral reef persistence under climate change*”, but it is 83 characters (with spaces) and therefore exceeds *Nature*'s required title length of 75 characters (with spaces). However, we agree with the Reviewer that “land-sea” forms a critical part of the message here. There are examples of recent *Nature* paper titles that exceed the 75 character limit (for example: <https://www.nature.com/articles/s41586-022-05640-x>), so perhaps there is some flexibility here that the Editor could comment on. Below we suggest our preferred title and an alternative shorter version for consideration:

Preferred: *Integrated land-sea management promotes coral reef persistence under climate change* (83 characters)

Alternative: *Integrated land-sea management promotes coral reefs under climate change* (72 characters)

I support publication of the article pending some minor adjustments, which I think could help the authors better navigate Reviewer 4's points 2 (R4.4 in the response to reviewers). Specifically, I disagree with reviewer 4 that the locality of the study should preclude publication, that the findings need to be applicable to every single reef on the planet (reviewer 4 observes that reducing wastewater is unlikely to make a difference in a remote, uninhabited atoll that doesn't wastewater because there are no people, but nearly 60% of the world's reefs are within 30 minutes of a human settlement and so this study's findings would indeed matter for the majority of the world's coral reefs), or that Hawaii is too much of an anomaly to warrant publication. On this latter point, the authors could illustrate where Hawaii fits in the global/regional context which would help demonstrate the relevance of this case study- or at the very least allow them to discuss key aspects that might not be generalisable. Reviewer 4 states that Hawaii has such unique "geology, geographic positioning, climate, reef communities, and anthropogenic stressors and management" that lessons simply aren't applicable to other places. While every place on the planet is by definition unique, I am dubious of the reviewer's claim that Hawaii is so anomalous that lessons are not applicable to other locations- but that is just my opinion. I suggest that the authors use some global studies to demonstrate where Hawaii fits into the global distribution for several of these issues (ones which are readily available). This actually builds of my earlier suggestion, which the authors did not take up, which is to contextualise the marine heatwave in Hawaii relative to the global or regional distribution. I think perhaps I wasn't totally clear, I suggest you plot the regional or global distribution of several key issues that the reviewer points out- say climate, anthropogenic stressors, management, etc. – and then mark where Hawaii sits on that distribution using a carpet plot at the bottom. This would allow you to quantify how unique or general Hawaii is- for example, you could take the distribution of global coral cover (even broken down by functional type) and the distribution of management (i.e. you could look at the proportion openly fished, restricted, and in marine reserves) from Emily Darling's 2019 Nature Ecology and Evolution paper and compare where Hawaii sits in the global distribution. You could look at some key human stressors (e.g. gravity, wastewater) in the 2022 A global map of human pressures on tropical coral reefs paper < <https://conbio.onlinelibrary.wiley.com/doi/full/10.1111/conl.12858>> and demonstrate where Hawaii sits. You could look at climate (e.g., mean SST, DHW) throughout the Pacific (or globally) from the 2016 heatwave and highlight where Hawaii sits. Mean SST, Etc. If Hawaii sits 2 SD outside of the global distribution on all of these factors simultaneously, then the reviewer has a point. However, if Hawaii falls within the 2SD of the global distribution for most of these, then I would argue that you have reasonably demonstrated that Hawaii isn't so anomalous after all. If it falls outside for one or two, then I think it is reasonable to discuss this and any implications this may have on the generalisability of the findings. Quite simply, demonstrate, don't argue...

We would like to thank the Reviewer for their very thoughtful and extremely helpful guidance here. We have followed their suggestion and now include a new figure (Extended Data Fig. 1) in

the revised version of the manuscript. The new figure places the Main Hawaiian Islands within the context of coral reefs globally. Specifically, the new figure compares human, environmental, and climate factors for coral reefs in Hawai‘i with those for coral reefs globally using the data from Darling et al. (2019), Tuholske et al. (2021), and Andrello et al. (2022). In summary, the mean for the Main Hawaiian Islands falls well within two standard deviations of the global mean for all factors, highlighting that indeed the reefs in Hawai‘i are generalisable to coral reefs globally. We have updated the manuscript (L101-104), added Extended Data Fig. 1, and modified the Materials and Methods section within the “*Study Site*” paragraph (L463-466) to capture this point as follows:

(L101-104): Our study reefs spanned large spatiotemporal gradients in land-sea human impacts and environmental factors (Fig. 1c) that are comparable to coral reef ecosystems globally (Extended Data Fig. 1), and which experienced the most severe marine heatwave on record in the Hawaiian Islands (Extended Data Fig. 2).

(L463-466): The coastline contains the longest contiguous reef ecosystem in the main Hawaiian Islands⁸² and large gradients in human population, local land-sea impacts, and environmental factors that are comparable to reef ecosystems globally (Extended Data Fig. 1).

Darling, E. S. et al. Social–environmental drivers inform strategic management of coral reefs in the Anthropocene. *Nature Ecology & Evolution* **3**, 1341-1350 (2019).

Tuholske, C. et al. Mapping global inputs and impacts from human sewage in coastal ecosystems. *Plos One* **16**, e0258898 (2021)

Andrello, M. et al. A global map of human pressures on tropical coral reefs. *Conserv Lett* **15**, e12858 (2022).

Referee #2 (Remarks to the Author):

I appreciate the efforts the authors' have gone to in revising their paper to all four referees. I'm happy with the revision pending one minor clarification. I see the revision now cites Mumby et al 2021 Cons Biology in the context of local drivers of reef response to protection of herbivory after disturbance. That's an appropriate citation and it's good to see the paper link to other studies that have provided such evidence.

However, my original comment regarded the challenge of finding evidence of drivers of reef recovery - Mumby et al 2021 Conservation Letters. This is a different paper and I think it's relevant for the context here - the main strength of the authors' paper is that it has the power to articulate management drivers of reef dynamics post-bleaching. Many studies do not as was shown in the Conservation Letters paper.

We appreciate the Reviewer pointing us towards the Mumby et al., (2021) *Conservation Letters* reference which highlights the striking challenge in detecting important conservation benefits in highly dynamic ecosystems like coral reefs. We have adjusted the second paragraph in the

revised manuscript to accommodate this reference (L80-83) as follows (note this paper was actually published in 2022):

(L80-83): Detecting conservation benefits in highly dynamic ecosystems is challenging (Mumby et al., 2022), but recent studies have identified salient connections between local conditions and coral reef resistance to and recovery potential following mass bleaching^{2,12,17-19}.

We note, however, that this (alongside previous Reviewer suggestions to cite additional literature) does move us over the recommended number of references within the main text by four references. We did previously remove some of our original citations to account for this, but cannot see further opportunity to do this without compromising the underpinning evidence of our statements. Perhaps there is some flexibility here that the Editor could comment on.

Mumby, P. J., Chaloupka, M., Bozec, Y.-M., Steneck, R. S. & Montero-Serra, I. Revisiting the evidentiary basis for ecological cascades with conservation impacts. *Conserv Lett* **15**, e12847 (2022).

Referee #4 (Remarks to the Author):

The authors have improved the manuscript following my initial comments, especially with regards to the overall framing. As indicated throughout my initial review, I think the manuscript is of high quality and provides interesting findings derived from a high resolution temporal dataset. Nevertheless, I am still not convinced that the paper offers the strong, unequivocal advances and notes of novelty one would expect from a manuscript published in Nature. The work is now fully centered on highlighting the role of integrative management in ameliorating land-based and marine stressors to help coral reefs maintain or bolster populations of reef-building organisms through various phases of disturbance (pre, during, and post). Yet, the evidence provided to support these claims is still relatively weak (despite the high quality of the underlying dataset), not only due to the factors I mentioned in my original review (extremely limited spatial scale, high context-dependency of coral reef dynamics and management actions), but also based on the statistical outcomes.

We thank the Reviewer for their positive comments on the improved framing and quality of the revised manuscript. The Reviewer also makes a number of comments that we have addressed with our updated submission. Specific changes and responses to these are below.

For the first analysis (pre-disturbance trajectories), there are strong correlations between several metrics of coral reef fish populations and reefs having a positive trajectory for coral cover, but it's impossible to infer causation from these results. Indeed, the strongest correlation appears to be for browsing, herbivorous reef fishes, which, according to the authors represented <10% of the total herbivore biomass on reefs. Thus, it is at least questionable whether these fishes could actively influence the trajectory of a reef. In turn, higher coral cover and resulting habitat complexity may simply support more fish, which would explain the positive relationships with fish-related variables across the board (cf. Russ et al. 2021; Darling et al. 2017). The negative correlations for the land-based stressors are intuitive, and directionality of the effect is not an issue.

The Reviewer raises an interesting and important point about Browsers that resulted in us re-visiting our presentation of the data in Fig. 2 (originally Fig. 2c, now Fig. 2d). In our previous submission, we showed the percentage difference in drop-one jackknife means for all land-sea human impact and environmental factors. Doing so facilitated the ability to show the difference between positive and negative trajectory reefs in all factors on a single panel. However, there were some downsides to this visualisation approach, which the Reviewer has astutely highlighted with respect to Browser biomass.

Our updated submission now shows the difference in absolute terms (Fig. 2d). Specifically, we plot the difference in the mean drop-one jackknife value for each factor in our comparison of conditions on positive and negative trajectory reefs. The difference in browser biomass is now much lower along the y-axis relative to total fish biomass, herbivore biomass, and scraper biomass, representing a more accurate reflection of the underlying patterns in the data. The percentage difference statements are still very useful for summarising the core trends in the data to the reader and so we have left these in the main text but moved our original Fig. 2c plot to Extended Data Fig. 3.

In addition, we have now removed the red, blue and grey colours for the plotted Jackknife means and range in our new Fig. 2d and Extended Data Fig. 3. We realised that this implied some form of formal statistical association with either positive or negative trajectory reefs and was potentially misleading. We have now opted to simply shade the background of the plot to indicate how the mean difference in values for each factor is associated with either positive or negative trajectories in coral cover.

The Reviewer also comments on the likely colineation between the local land-sea human impact and environmental factors. Our original goal here was simply to show the data and not undertake a formal model-fitting exercise. This was because we felt (and still feel) the lower replication here did not lend well to the types of model-fitting approaches we undertook in the subsequent two sections of the paper (*Coral response to the 2015 marine heatwave*, and, *Coral reefs four years post-disturbance*). However, we have now added a more formal analysis to this section that embraces the colineation between the local land-sea human impact and environmental factors and their respective values (see new Fig. S2) among positive and negative trajectory reefs. We have used a permutational multivariate analysis of variance (PERMANOVA) (Anderson 2017) to formally test for a difference in the human-environmental conditions of positive *versus* negative trajectory reefs. The results indicated that positive and negative trajectory reefs do indeed have distinct human-environmental conditions ($Pseudo-F_{1,17}=3.38$, $p=0.001$). We have now visualised this distinctiveness along a single multivariate axis using a canonical analysis of principal coordinates analysis (CAP, Anderson & Willis 2003) (Fig. 2c). We then calculated the cross-validation allocation success from the leave-one-out procedure of the CAP analysis. This gave a further measure of relative group distinctness for positive and negative trajectory reefs in terms of their human-environmental factors. Positive and negative trajectory reefs had allocation success values of 90% and 87.5%, respectively (>50% indicates an increasingly more distinct set of attributes than expected by chance alone - this threshold comes from the possibility of each individual observation having a 50% chance of being placed into one of the two groups). We then contextualise these differences in Fig. 2d for the reader, and highlight those variables that

showed the greatest differences (i.e., the best discriminators) between positive and negative trajectory reefs in the main text.

These additional analyses provide quantitative evidence that the local land-sea human impact and environmental factors differ significantly between positive and negative trajectory reefs, while accounting specifically for the variable colineation and multidimensional nature of the data. We hope this helps to alleviate some of the Reviewer's concerns. Obviously, inferring causation from any observational study such as this is not appropriate, but we can talk in terms of statistical evidence and inference and we feel these additional analyses have improved the rigour of this section of the paper. We thank the Reviewer for prompting these changes.

Finally, the Reviewer comments on the bi-directional nature of an age-old challenge in observational fish-benthos data collected on coral reefs that is also present in other subtidal systems such as kelp forests. In fact, it is of course likely that a positive feedback exists over these longer time periods, whereby increasing coral cover promotes habitat suitability for reef fishes, with herbivores fishes then in turn facilitating coral growth by reducing competitive exclusion by fleshy algae (Bozec et al. 2013). We have added the following sentence to this effect within this part of the paper (*Reef trajectories pre-disturbance*, L141-144) and thank the Reviewer for prompting this:

(L141-144): These patterns likely reflect positive feedbacks, whereby increasing coral cover promotes habitat suitability for reef fishes, with herbivorous fishes then facilitating coral growth by reducing competitive exclusion by fleshy algae²⁸.

Anderson, M. J., Walsh, D. C. I., Robert Clarke, K., Gorley, R. N. & Guerra-Castro, E. Some solutions to the multivariate Behrens–Fisher problem for dissimilarity-based analyses. *Australian & New Zealand Journal of Statistics* **59**, 57-79 (2017).

Anderson, M. J. & Willis, T. J. Canonical Analysis of Principal Coordinates: A Useful Method of Constrained Ordination for Ecology. *Ecology* **84**, 511-525 (2003).

Bozec, Y.-M., Yakob, L., Bejarano, S. & Mumby, P. J. Reciprocal facilitation and non-linearity maintain habitat engineering on coral reefs. *Oikos* **122**, 428-440 (2013).

During the heatwave, the authors provide somewhat compelling evidence that urban runoff and sediment input exacerbate the response of corals to bleaching while phytoplankton concentration appears to have a positive effect. While these relationships are interesting (especially for the phytoplankton), the latter is also decidedly unmanageable when it comes to marine policy, so it doesn't really support the main message of the manuscript ("Integrated management promotes coral reef persistence under climate change"). In contrast, the variables that can be targeted by marine policy decisions (fish biomass and herbivorous fishes, in particular) showed no meaningful relationship with coral cover throughout the disturbance whatsoever.

We appreciate the Reviewer's comments on the relevancy of our findings, namely whether phytoplankton biomass and land-based pollution are helpful for supporting management decision making and implementing marine policy.

In regard to phytoplankton biomass: natural biophysical gradients, such as wave forcing, temperature, and phytoplankton biomass, drive coral reef ecosystem structure and function across multiple spatial and temporal scales (Williams et al. 2019). The relative influence of natural drivers can vary by geography and is often context-specific. While natural drivers are inherently unmanageable, understanding how natural drivers combine with human impacts is key for setting locally relevant and place-based management targets. This is because resource managers must know whether and how existing marine management will be effective, and if not, whether the implementation of adaptive management is required. We have now added the following sentence to the manuscript (L206-209) helping to explain to the reader the importance and marine management relevance of this result:

(L206-209): Working towards locally relevant management strategies requires understanding how human impacts superimpose on natural biophysical drivers, like phytoplankton biomass²⁰, to influence reef ecosystem response to acute disturbance.

Williams, G. J. et al. Coral reef ecology in the Anthropocene. *Funct Ecol* **33**, 1014-1022 (2019).

Regarding the Reviewer's comment about whether marine ecosystem management can be informed by understanding the impacts of land-based pollution on coral reef response to severe marine heatwaves – the answer is most definitely *yes*. Throughout this study we have worked with and included both Federal and State resource managers in the study design, write-up and interpretation. This scientist-manager partnership has been essential for ensuring our manuscript investigates questions that are both relevant and applicable to real-world management decision making. To that end, we felt that our manager co-authors (Brian Neilson and Gerald Davis) were the most appropriate to provide an informed response to this Reviewer comment and their respective summaries are below in *italics*.

Brian Neilson, Administrator of the Division of Aquatic Resources, Department of Land and Natural Resources, State of Hawai'i:

Given the scale and magnitude of urban runoff and sedimentation impacts on coral reefs it may seem like a daunting challenge, but these stressors are certainly manageable from a marine policy standpoint; especially at the local scale, if the political will and capital are available. In the United States, urban runoff and sediment input can be managed through Federal, State, and County laws and restoration activities. For example, Federal and State water quality laws can be used to promote compliance with water quality standards. In addition, managers can restore watersheds by revegetation efforts, ungulate control, and wastewater management. These are longer-term actions but can increase reef resilience to stressors related to climate change. Below are a examples of existing policy measures and management actions that address runoff and sedimentation impacts to coral reefs. Note that these policy measures are often limited by the capacity to monitor and enforce:

Examples of existing U.S. Federal, State, and County policy measures:

- *Environmental Laws requiring Environmental Assessments or Environmental Impact Statements*

- *Water Quality Standards: Clean Water Act (CWA), State Standards for CWA*
- *National Pollution Discharge Elimination System (NPDES)*
- *County zoning and land use regulatory system*
- *County building permits*
- *County shoreline ordinances, rules, and setbacks*
- *State laws governing land use*
- *State coastal zone management*
- *State laws governing coastal development, construction, and maintenance*
- *State laws governing soil erosion and sediment control*
- *State water pollution laws*
- *State wastewater laws*

Examples of management actions that reduce land-based pollution, like urban runoff and sediment input, and help mitigate the associated impacts to coral reefs:

- *Watershed management*
- *Stream restoration*
- *Wetland/estuarine restoration*
- *Storm water management*
- *Revegetation*
- *Ungulate control*
- *Invasive species control*
- *Riparian zone management*
- *Engineered sediment traps/deposition zones*
- *Sewage treatment infrastructure*

Gerald Davis, Director of the Habitat Conservation Division, Pacific Islands Regional Office, National Oceanic and Atmospheric Administration:

The National Marine Fisheries Service (NMFS) under the Magnuson-Stevens Fisheries Conservation and Management Act (MSA; 16 U.S. Code § 1881c) has the authority to address land-based pollution as it pertains to fisheries management. For example, NMFS performs consultations that evaluate impacts to federally managed fisheries under the MSA regulatory requirement for impacts to designated Essential Fish Habitat (EFH). EFH in the Hawaiian Islands extends from the shoreline to the outer edge of the exclusive economic zone (i.e., 200 nm from shore). EFH is designated based on a management unit species habitat needed to spawn, reproduce, grow, feed and mature. EFH also includes the substrata and water column. Science that demonstrates linkages between land-based sources of freshwater and associated transport of particulates (e.g., sediment, chemicals, toxins, etc.) to fisheries condition is fundamental to managing fishery dependent habitats such as coral reefs. These linkages include, but are not limited to: water temperature, suspended particulates, introduced toxins, alien invasive species, nutrient levels, types of resource extraction and associated impacts to habitat condition, and shoreline discharge and associated impacts to water quality. Fishery and EFH conditions are the ultimate measures of ecosystem condition, especially when they can be linked to management intervention as is the case in this study.

These responses by resource managers on how existing regulatory policy can be utilised to address land-based stressors on coral reefs hopefully alleviates any concerns the Reviewer might have on the relevancy of our findings to marine management. In addition, a recent paper in the *Proceedings of the National Academy of Sciences* by Carlson et al., (2022) *Untapped policy avenues to protect coral reef ecosystems* provides a very thorough summary of U.S. and international policies that provide the regulatory framework for addressing land-based pollution on coral reefs. We have now added the following language in the manuscript that summarises this important point (L220-224):

(L220-224): Existing but underutilised local and national policies like the Clean Water Act in the United States provide actionable pathways for marine management interventions of land-based stressors (Carlson et al. 2022). Management strategies that leverage such policies to help mitigate coastal runoff, particularly in urban areas, may support increased coral survival during severe marine heatwaves.

Carlson, R. R., Foo, S. A., Burns, J. H. R. & Asner, G. P. Untapped policy avenues to protect coral reef ecosystems. *Proceedings of the National Academy of Sciences* **119**, e2117562119 (2022).

Finally, the post-disturbance recovery analysis is, in my opinion, the strongest part of the manuscript, as it actually shows the potential synergistic effects of managing a land-based stressor and (possibly) fish populations on coral reefs. Of course, the directionality of the correlation between scraper biomass and reef-builder cover is subject to the same uncertainty as mentioned above (i.e., bottom up effects of reef structure and architecture rather than top-down effects of scrapers on the reef itself). For example, DeMartini et al. (2013, J. Coastal Conservation) showed that parrotfishes preferentially recruit to branching corals, notably in the same location as the present study, with a strong effect of sedimentation on both corals and parrotfishes.

The Reviewer again raises the very important point of the bi-directional nature of fish-benthos relationships on coral reefs. In this section of the paper (*Coral reefs four years post-disturbance*), we use a short observational window (i.e., four years) to focus on the top-down effects of resident herbivorous fishes and minimise the longer-term effects of increasing habitat suitability promoting fish recruitment. This was done by our temporal pairing of the fish and benthos data whereby we constrain the temporal window to an ecologically relevant time period. In this case, scraper biomass estimates were derived from multiple observations across several time points following the marine heatwave, rather than a single snapshot estimate. Furthermore, most herbivorous reef fishes are strongly site attached, usually with small home ranges as both adults and juveniles (Mumby & Wabnitz 2002, Bellwood et al. 2016). This includes parrotfish, which have a home range of < 1 km in our study region (Meyer et al. 2010). It is therefore unlikely that fish from other locations migrated into areas of high reef-builder cover post-disturbance. We have now added the following language in the manuscript that summarises this important point (L267 - 276) while also noting the possible influence of bottom-up effects as the Reviewer advises:

(L267 - 276): *Beyond these top-down effects on benthic condition, bottom-up effects of improved habitat quality could be contributing to the positive relationship we observed between scraper biomass and higher reef-builder cover. Parrotfish are the dominant scrapers in Hawai‘i, and typically have home ranges of less than 1 km⁴⁹. Furthermore, our scraper biomass estimates were derived from multiple observations across several time points following the marine heatwave, rather than a single snapshot estimate. Such strong site-based fidelity, combined with our recurring surveys, suggests that resident scrapers played a key role in promoting higher reef-builder cover rather than the association driven purely by an influx of individuals seeking more favourable habitat post-disturbance.*

Mumby, P. J. & Wabnitz, C. C. C. Spatial Patterns of Aggression, Territory Size, and Harem Size in Five Sympatric Caribbean Parrotfish Species. *Environ Biol Fish* **63**, 265-279 (2002).

Bellwood, D. R., Goatley, C. H. R., Khan, J. A. & Tebbett, S. B. Site fidelity and homing in juvenile rabbitfishes (Siganidae). *Coral Reefs* **35**, 1151-1155 (2016).

Meyer, C. G., Papastamatiou, Y. P. & Clark, T. B. Differential movement patterns and site fidelity among trophic groups of reef fishes in a Hawaiian marine protected area. *Mar Biol* **157**, 1499-1511 (2010).

With these comments, I am not insinuating that the authors' conclusions are invalid. As I mentioned in my original review, I believe that the findings of the manuscript are intuitive based on what we know about coral reefs and especially their response to land-based stressors. However, I do believe that publication in Nature calls for extraordinary findings backed by extraordinarily strong evidence. I do not wish to make myself the authority and gatekeeper of what does (and doesn't) qualify as such, and I certainly support one of the other reviewers' opinion that the field of coral reef ecology can do with results that highlight the ability of societies to manage reef dynamics. Nevertheless, I am not 100% convinced that the manuscript provides unequivocal evidence for the consequences of integrative land-sea based management and since the reviewer instructions call for assessments of novelty and impact, I feel compelled to present my evaluation of manuscript with regards to these criteria.

Thus, overall, I stand by my original opinion, which is that the manuscript is of high quality with no methodological concerns, but that the evidence presented within the paper does not necessarily hit the high notes of strength of evidence, novelty and impact that I usually associate with the journal.

We thank the Reviewer for once again reiterating that they find our manuscript to be high quality in its technical execution and hope that we have further improved the rigor of our data presentation and analyses with the amendments outlined above.

Reviewer Reports on the Second Revision:

Referees' comments:

Referee #1 (Remarks to the Author):

I am satisfied with how the authors have addressed the points raised by myself and the other reviewers.

Referee #4 (Remarks to the Author):

I have re-assessed the paper by Gove et al. following their resubmission. I appreciate the revised visualization and additional multivariate analysis in Fig. 2, which I believe improves that paper. Their reasoning about the inference of directionality regarding herbivorous fishes and coral cover is acceptable, although it remains impossible to resolve cause and effect (but that's the case for all observational studies of this nature).

Regarding my second comment, I would like to point out that I did not question the usefulness of understanding effects of land-based pollution on coral reefs from a management perspective. I apologize if my phrasing wasn't entirely clear to the authors, but I stated that: "During the heatwave, the authors provide somewhat compelling evidence that urban runoff and sediment input exacerbate the response of corals to bleaching, while phytoplankton concentration appears to have a positive effect. While these relationships are interesting (especially for the phytoplankton), the latter is also decidedly unmanageable when it comes to marine policy, so it doesn't really support the main message of the manuscript ("Integrated management promotes coral reef persistence under climate change"). Thus, I am specifically only referring to the phytoplankton when I say that it is not a manageable variable, and thus doesn't really support the main message of the paper. I fully agree that understanding baseline environmental settings and their effect on reef configurations and dynamics is critical, but to me, that is a different kettle of fish from the central plank of the paper (which is that combined land-sea management bolsters coral populations). I could not agree more that understanding effects of land-based pollution on reefs is a critical and extremely valuable endeavor.

That being said, I still cannot help flagging that the effects displayed in Fig. 3 (and EDF 4) are really quite weak and full of uncertainty (as highlighted by the 'somewhat compelling' in my previous comments). For sediment input, for example, the upper bounds of the uncertainty estimate at the extreme end of the predictor (10,000kg ha⁻¹) exceeds the mean (?) estimated fit at 0kg ha⁻¹). The authors show that sediment input is the highest ranked variable in the AIC framework, but they rightly state that this is *relative* importance in a likelihood-based framework and thus has no bearing on the magnitude of the biological effect in nature. These analyses are the cornerstone of a paper that is entitled "Integrated land-sea management promotes coral reef persistence under climate change", but the actual effect sizes are not particularly strong or unambiguous, and one of the strongest relationships is with a variable that cannot be managed (phytoplankton, see comment above). I recognize that there is some subjectivity in the assessment of what is (or isn't) a strong or meaningful effect beyond statistical significance, but I don't think the results as presented in this part of the manuscript are as strong as the rest of the paper tries to paint them.

Therefore, I am still somewhat uncomfortable recommending publication in Nature, due to the reservations I have expressed above and throughout the review process. However, this is simply what the data say and it won't change with additional rounds of review or analyses, so if the editor and all other reviewers believe that the results are strong enough to merit publication in Nature,

then I might just be the one who is being overly critical or pedantic. As mentioned on multiple occasions, the quality of the manuscript per se is excellent, so it all just boils down to how impactful and ground-breaking one gauges the findings to be.

Author Rebuttals to Second Revision:

Referees' comments:

Referee #1 (Remarks to the Author):

I am satisfied with how the authors have addressed the points raised by myself and the other reviewers.

We thank the Reviewer for their continued support of our paper.

Referee #4 (Remarks to the Author):

I have re-assessed the paper by Gove et al. following their resubmission. I appreciate the revised visualization and additional multivariate analysis in Fig. 2, which I believe improves that paper. Their reasoning about the inference of directionality regarding herbivorous fishes and coral cover is acceptable, although it remains impossible to resolve cause and effect (but that's the case for all observational studies of this nature).

We are glad the Reviewer found these changes to be acceptable and to improve the paper.

Regarding my second comment, I would like to point out that I did not question the usefulness of understanding effects of land-based pollution on coral reefs from a management perspective. I apologize if my phrasing wasn't entirely clear to the authors, but I stated that: "During the heatwave, the authors provide somewhat compelling evidence that urban runoff and sediment input exacerbate the response of corals to bleaching, while phytoplankton concentration appears to have a positive effect. While these relationships are interesting (especially for the phytoplankton), the latter is also decidedly unmanageable when it comes to marine policy, so it doesn't really support the main message of the manuscript ("Integrated management promotes coral reef persistence under climate change"). Thus, I am specifically only referring to the phytoplankton when I say that it is not a manageable variable, and thus doesn't really support the main message of the paper. I fully agree that understanding baseline environmental settings and their effect on reef configurations and dynamics is critical, but to me, that is a different kettle of fish from the central plank of the paper (which is that combined land-sea management bolsters coral populations). I could not agree more that understanding effects of land-based pollution on reefs is a critical and extremely valuable endeavor.

Thank you for clarifying. Our goal with the title is to communicate the most salient results from our study. We agree that phytoplankton is not a directly manageable variable, however the positive effect of increased nearshore phytoplankton on coral survival during the marine heatwave is just one of many results across our study. In all three phases of our analyses (1. Reef trajectories pre-disturbance, 2. Coral response to the marine heatwave, and 3. Coral reefs four years post-disturbance) some combination of a reduction of land-sea human impacts correlated with increased coral persistence over time. The combination of impacts changed depending on the temporal period in question, as we point out in the first paragraph of our Conclusion. However, arguably our most critical finding – *that the simultaneous reduction in land-sea human impacts resulted in a 3- to 6-fold greater probability of a reef having high reef-builder cover four years post-disturbance than if either occurred in isolation* – directly supports the paper title. The core result from our pre-disturbance analysis – *reefs with increased herbivorous fish populations and reduced land-based impacts, like*

wastewater pollution and urban runoff, had positive coral cover trajectories pre-disturbance – also directly supports the paper title. Finally, a core result from our coral response to the marine heatwave analysis – *that reefs with reduced urban runoff and sediment input experienced a modest reduction in coral mortality following severe heat stress* – directly supports the paper title.

In summary, the overwhelming majority of our results pertaining to pre-, during, and post-disturbance are directly applicable to, and can be influenced by, resource management actions and therefore support our proposed title of the manuscript.

*That being said, I still cannot help flagging that the effects displayed in Fig. 3 (and EDF 4) are really quite weak and full of uncertainty (as highlighted by the ‘somewhat compelling’ in my previous comments). For sediment input, for example, the upper bounds of the uncertainty estimate at the extreme end of the predictor (10,000kg ha⁻¹) exceeds the mean (?) estimated fit at 0kg ha⁻¹). The authors show that sediment input is the highest ranked variable in the AIC framework, but they rightly state that this is *relative* importance in a likelihood-based framework and thus has no bearing on the magnitude of the biological effect in nature. These analyses are the cornerstone of a paper that is entitled “Integrated land-sea management promotes coral reef persistence under climate change”, but the actual effect sizes are not particularly strong or unambiguous, and one of the strongest relationships is with a variable that cannot be managed (phytoplankton, see comment above). I recognize that there is some subjectivity in the assessment of what is (or isn’t) a strong or meaningful effect beyond statistical significance, but I don’t think the results as presented in this part of the manuscript are as strong as the rest of the paper tries to paint them.*

Therefore, I am still somewhat uncomfortable recommending publication in Nature, due to the reservations I have expressed above and throughout the review process. However, this is simply what the data say and it won’t change with additional rounds of review or analyses, so if the editor and all other reviewers believe that the results are strong enough to merit publication in Nature, then I might just be the one who is being overly critical or pedantic. As mentioned on multiple occasions, the quality of the manuscript per se is excellent, so it all just boils down to how impactful and ground-breaking one gauges the findings to be.

We appreciate this additional prompt by Reviewer 4 to temper our statements with respect to the correlation of coastal runoff and coral mortality within the second phase of our analyses (*Coral response to the marine heatwave*). We recognise the Reviewer’s concern and have modified our language within both the abstract and main text when communicating this result. Specifically, we have made the following changes to the text to better articulate the more modest effect these variables had on coral loss during the marine heatwave:

Abstract (L54-57) Reefs with increased herbivorous fish populations and reduced land-based impacts, like wastewater pollution and urban runoff, had positive coral cover trajectories pre-disturbance and experienced a modest reduction in coral mortality following severe heat stress.

Main Text (178-180) We found that reefs exposed to the lowest levels of urban runoff, and to a lesser extent sediment input, experienced a modest reduction in coral mortality from the marine heatwave (Fig. 3d).